# Glycogenesis and glyconeogenesis from glutamine, lactate and glycerol support human macrophage functions

Najia Jeroundi[1,10], Charlotte Roy [ID][1,10], Laetitia Basset [ID][1], Pascale Pignon[1], Laurence Preisser[1], Simon Blanchard [ID][1,2], Cinzia Bocca[3,4], Cyril Abadie[5], Julie Lalande[5], Naïg Gueguen[3,4], Guillaume Mabilleau [ID][6,7], Guy Lenaers[3,4], Aurélie Moreau[8], Marie-Christine Copin[1,7], Guillaume Tcherkez[5,9], Yves Delneste[1,2], Dominique Couez[1,11] & Pascale Jeannin [ID][1,2,11] ✉

## Abstract

**Macrophages fight infection and ensure tissue repair, often operating at nutrient-poor wound sites. We investigated the ability of human macrophages to metabolize glycogen. We observed that the cytokines GM-CSF and M-CSF plus IL-4 induced glycogenesis and the accumulation of glycogen by monocyte-derived macrophages. Glyconeogenesis occurs in cells cultured in the presence of the inflammatory cytokines GM-CSF and IFNγ (M1 cells), via phosphoenolpyruvate carboxykinase 2 (PCK2) and fructose-1,6-bisphosphatase 1 (FBP1). Enzyme inhibition with drugs or gene silencing techniques and $^{13}$C-tracing demonstrate that glutamine (metabolized by the TCA cycle), lactic acid, and glycerol were substrates of glyconeogenesis only in M1 cells. Tumor-associated macrophages (TAMs) also store glycogen and can perform glyconeogenesis. Finally, macrophage glycogenolysis and the pentose phosphate pathway (PPP) support cytokine secretion and phagocytosis regardless of the availability of extracellular glucose. Thus, glycogen metabolism supports the functions of human M1 and M2 cells, with inflammatory M1 cells displaying a possible dependence on glyconeogenesis.**

**Keywords** Macrophages; Glyconeogenesis; Glycogenolysis; Cytokine secretion; Phagocytosis
**Subject Categories** Immunology; Metabolism

## Introduction

Macrophages are myeloid cells found in most tissues that control many processes, including wound healing and tissue repair, inflammatory responses and their resolution, anti-infectious responses, and tumor development. They are often nicknamed "firefighter cells" due to their ability to survive in damaged and/or infected tissues and to orchestrate the local recruitment, proliferation, and/or functions of other cells, to enable a return to homeostasis (Okabe and Medzhitov, 2016). When tissue-resident macrophages are overwhelmed by damage or infection, circulating monocytes are recruited and differentiate locally into macrophages. Macrophages continuously screen their microenvironment and adapt their functions to the demands of the tissue. They are characterized by considerable phenotypic and functional plasticity that renders their classification difficult (Locati et al, 2020; Xue et al, 2014; Okabe and Medzhitov, 2016). A simplistic binary classification of macrophages into the M1 and M2 groups, corresponding to the ends of a continuum, is still widely used (Mosser and Edwards, 2008; Locati et al, 2020). M1 cells have inflammatory, antimicrobial, and antitumor properties, whereas M2 cells maintain tissue homeostasis and facilitate tissue healing and repair. Murine M1 and M2 cells also differ in terms of metabolism, with M1 cells principally making use of glycolysis, whereas M2 cells are dependent on glutamine (Gln) metabolism and fatty-acid oxidation (FAO) (Izquierdo et al, 2015; Van den Bossche et al, 2017). There are marked differences in phenotype and metabolism between human and murine macrophages (Monnier et al, 2022).

Various parameters, including oxygen availability, nutrient availability, and extracellular pH, influence macrophage metabolism and function (Geeraerts et al, 2017). Inflammatory cytokine production by myeloid cells is thought to require a shift to a glycolytic flux dependent on glucose availability (Van den Bossche et al, 2017; Tannahill et al, 2013). However, glucose availability is often low in infected or injured tissues and in solid tumor microenvironments (Hirayama et al, 2009;

[1]Univ Angers, Nantes Université, Inserm, CNRS, CRCI2NA, SFR ICAT, LabEx IGO, F-49000 Angers, France. [2]Immunology and Allergology laboratory, University Hospital, Angers, France. [3]Univ Angers, Inserm, CNRS, MitoVasc, SFR ICAT, F-49000 Angers, France. [4]Department of Genetics and Biochemistry, University Hospital, Angers, France. [5]Univ Angers, INRAe, IRHS, SFR QUASAV, F-49000 Angers, France. [6]Univ Angers, Nantes Université, Inserm, Oniris, RMeS, SFR ICAT, F-49000 Angers, France. [7]Department of Cell and Tissue Pathology, University Hospital, Angers, France. [8]Inserm, Nantes Université, University Hospital of Nantes, Centre de Recherche Translationnelle en Transplantation et Immunologie, Nantes, France. [9]Research School of Biology, ANU College of Science, Australian National University, Canberra, ACT 2601, Australia. [10]These authors contributed equally: Najia Jeroundi, Charlotte Roy. [11]These authors jointly supervised this work: Dominique Couez, Pascale Jeannin. ✉E-mail: pascale.jeannin@univ-angers.fr

Tucey et al, 2019). For example, glucose concentrations in gastric and colon cancers range from 0.1 to 0.4 mM, values markedly lower than the mean concentration of glucose in blood (Hirayama et al, 2009). The metabolic strategies used by human macrophages to perform their functions in conditions in which the nutrient supply is limited remain largely unknown.

Glycogen is a large branched-chain glucose polymer. It is stored principally in the liver and skeletal muscle, where it constitutes a reserve of carbohydrate fuel to meet the metabolic and energy demands of the cell. Glycogen synthesis (glycogenesis) and degradation (glycogenolysis) are controlled by tissue-specific enzymes: glycogen synthase (GYS) and glycogen phosphorylase (PYG), respectively (Adeva-Andany et al, 2016). Glycogen stores are usually generated from extracellular glucose through glycogenesis. Hepatocytes and, to a lesser extent, kidney cells can also use non-carbohydrate substrates, such as pyruvate, lactate, glycerol and glucogenic amino acids, to generate glucose-6-phosphate (G6P) in a process known as gluconeogenesis. In these cells, G6P can be used to generate glycogen for storage or can be hydrolyzed by glucose-6-phosphatase (G6PC1) to generate glucose. Thus, G6PC1 maintains euglycemia in the fasting state by catalyzing the terminal step in the hepatic and renal gluconeogenesis and glycogenolysis pathways (Hutton and O'Brien, 2009). The G6P used for glycogenolysis in muscle fuels ATP production via the TCA cycle, which is essential for cell contraction.

Pioneering studies reported the presence of glycogen stores in human and rodent myeloid cells (Eichner et al, 1983; Scott, 1968; Stossel et al, 1970; Wulff, 1962). However, the presence and role of these reserves in myeloid cells have recently been re-examined. In mice, M1 cells differ from M2 cells in that they synthesize glycogen and are dependent on glycogenolysis for the secretion of inflammatory cytokines (Ma et al, 2020). Dendritic cells also store glycogen and use glycogenolysis to drive cytokine production and mitochondrial respiration (Thwe et al, 2017). Neutrophils generate glycogen stores through glyconeogenesis and glycogenesis and use these stores to ensure their own survival and to produce inflammatory cytokines (Sadiku et al, 2021).

By contrast, glycogen metabolism has never been evaluated in human macrophages. Using human monocyte-derived macrophages and tumor-associated macrophages (TAMs), we show here that a combination of the inflammatory cytokines GM-CSF and IFNγ triggers glyconeogenesis from glutamine, lactic acid, and glycerol in macrophages, whereas treatment with GM-CSF or M-CSF plus IL-4 promotes glycogenesis. We found that glycogenolysis fueled at least in part the pentose phosphate pathway (PPP) and supported human macrophage functions (cytokine secretion and phagocytosis), regardless of extracellular glucose availability. These results highlight the metabolic plasticity of human macrophages and shed light on the ways in which they act to restore homeostasis in nutrient-depleted injured tissues.

# Results

## GM-CSF or M-CSF plus IL-4 triggers human macrophage glycogenesis

Two cytokines, GM-CSF and M-CSF, promote human monocyte survival and differentiation into macrophages (Mφ): GM-CSF-Mφ (orange histogram) for those induced by GM-CSF and M-CSF-Mφ

(gray histogram) for those induced by M-CSF (Fig. 1A) (Jeannin et al, 2018). GM-CSF is present only at sites of inflammation, whereas M-CSF is constitutively and ubiquitously expressed by many tissues and cell types (Hamilton, 2019; Jeannin et al, 2018). Prototypic human M1 (green histogram) and M2 (purple histogram) cells are typically generated from monocytes in the presence of GM-CSF plus IFNγ and M-CSF plus IL-4, respectively, and exhibited conventional M1 (CD40[high] CD80[high] CD86[high] CD14[low] CD163[low] HLA-DR[low]) and M2 (CD40[low] CD80[low] CD86[low] CD14[high] CD163[high] HLA-DR[high]) profiles (Fig. EV1A,B). We therefore investigated the ability of these Mφ subtypes to synthesize and store glycogen.

Consistent with previous reports (Thwe et al, 2017), monocytes (Fig. 1A, white histogram) were found to store very little glycogen. Glycogen levels in GM-CSF-Mφ were slightly higher than those of monocytes on day 2 and much higher on day 5, and no modulation of glycogen levels was observed when IFNγ was added during the differentiation process (M1 cells) (Fig. 1A). Unlike GM-CSF-Mφ, M-CSF-Mφ had very low levels of glycogen on day 5, similar to those in monocytes, unless IL-4 was added (M2 cells) (Fig. 1A). The addition of IL-4 at day 2 to differentiating M-CSF-Mφ triggered a small but significant accumulation of glycogen after 24 h (D3) and a substantial accumulation after 3 days (D5) (Fig. 1B). This observation helps to explain the lack of glycogen detection in murine M2 cells generated with M-CSF but incubated for only 24 h with IL-4 (Ma et al, 2020). These results were confirmed by (i) periodic acid-Schiff (PAS) staining which revealed multiple PAS-positive cytoplasmic granules in M1 and M2 cells and (ii) by the addition of the fluorescent D-glucose analog, 2-NBDG (Louzao et al, 2008; Zhu et al, 2020), which incorporation resulted in the appearance of fluorescent granules in M1 and M2 cells (Fig. 1C, left panel). Image analysis showed that >95% of monocytes and day 5 M1 and M2 cells were positive for PAS staining or 2-NBDG internalization (Fig. 1C, left panels) and confirmed that M1 and M2 cells contain higher levels of glycogen than monocytes (Fig. 1C, right panels).

We then investigated whether IL-1β and IL-10, other cytokines that modulate human macrophages polarization (Pérez and Rius-Pérez, 2022), could modulate macrophage glycogen content. None of them affected glycogen content by GM-CSF-Mφ or M-CSF-Mφ, except IL-4 that significantly enhanced glycogen levels in GM-CSF-Mφ (Fig. 1D). Finally, unlike IL-4, IFNγ did not induce glycogen accumulation by M-CSF-Mφ (Fig. 1D). Thus, both GM-CSF and M-CSF plus IL-4 induce glycogen storage by human macrophages.

We then demonstrated that GM-CSF-Mφ and M2 cells use glucose (Glc) as a substrate for glycogenesis. We first checked that the differentiation of monocytes into GM-CSF-Mφ and M2 cells was accompanied by the consumption of extracellular Glc (Fig. 1E), as previously reported (Izquierdo et al, 2015; Steitz et al, 2020; Zhou et al, 2021). We found that GM-CSF-Mφ and M2 cells generated by replacing the culture medium (CM, containing 2 g/L Glc) with Glc[low] medium (hatched bars) on day 2, contained very small amounts of stored glycogen on day 5 (Fig. 1F). To confirm that extracellular glucose was used to synthesize glycogen, GM-CSF-Mφ and M2 cells were generated in the presence of $^{13}C_6$-Glc. At day 5, the intracellular glycogen pool was purified and then hydrolyzed to generate Glc monomers before quantification by GC-MS, as described (Gudewicz and Filkins, 1976). Results revealed the presence of $^{13}C$-labeled glycogen in both cell types (Figs. 1G and EV1C).

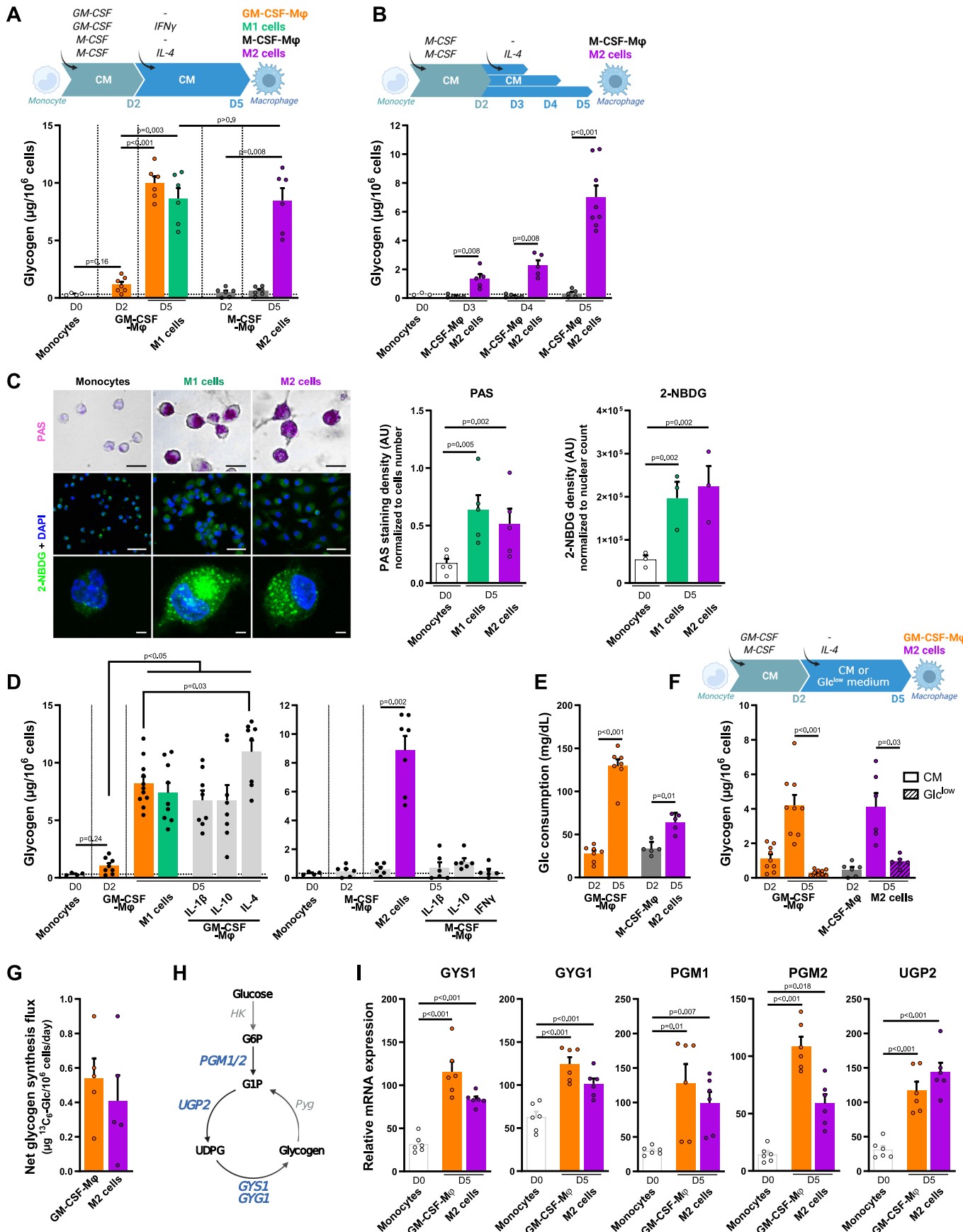

◀

**Figure 1. GM-CSF or M-CSF plus IL-4 triggers human macrophage glycogenesis.**

Monocytes were differentiated into GM-CSF-Mφ, M1 cells (GM-CSF plus IFNγ added at day 2), M-CSF-Mφ or M2 cells (M-CSF plus IL-4 added at day 2) in complete medium (CM) (A–E, I) or with a switch on day 2 to Glc$^{low}$ medium (F, G). Timelines summarize experimental procedures for Mφ generation. (A) Glycogen quantification in monocytes and in day 2 and day 5 Mφ ($n = 6$). (B) Day 2 M-CSF-Mφ were exposed to IL-4 (purple) or not (gray) for the last 3 days. Glycogen was quantified in monocytes (D0) and after 24 h (D3), 48 h (D4) and 72 h (D5) incubation without or with IL-4 ($n = 5$–8). (C) Left panels: monocytes and day 5 M1 and M2 cells were analyzed by light microscopy after PAS staining (upper panels; scale bar, 50 μm) or by confocal microscopy after 2 h incubation with 2-NBDG (middle and lower panels; scale bars, 40 and 2 μm). For fluorescent microscopy analysis, nuclei were counterstained with DAPI. Results are representative of 1 out of 3 (for 2-NBDG) or 5 (for PAS) different donors. Right panels: relative intensity of PAS staining and 2-NBDG fluorescence in monocytes and day 5 M1 and M2 cells (300 cells/donor were analyzed). (D) Day 2 GM-CSF- or M-CSF-Mφ were exposed or not to IFNγ, IL-1β, IL-10 or IL-4. Glycogen was quantified at day 5 ($n = 6$–10). (E) Glucose consumption in the day 2 and 5 supernatants of GM-CSF-Mφ and M2 cells ($n = 5$). (F) Glycogen was quantified in GM-CSF-Mφ and M2 cells cultured in CM or Glc$^{low}$ medium ($n = 6$–9). (G) Monocytes were differentiated into GM-CSF-Mφ and M2 cells in CM with a switch on day 2 to Glc$^{low}$ medium supplemented with $^{13}C_6$-Glc. Glycogen was purified at day 5 and hydrolyzed to monomers before GC-MS analysis ($n = 5$). (H) Overview of enzymes involved in glycogenesis. (I) Relative mRNA expression of GYS1, GYG1, PGM1/2, and UGP2 determined by RT-qPCR in monocytes and day 5 GM-CSF-Mφ and M2 cells ($n = 6$). Values are represented as the mean ± SEM, each dot represents a different donor. Statistical significance was determined by Welch's ANOVA test followed by Dunnett's multiple comparison post hoc test (A, D, F, I) or by two-tailed unpaired Welch $t$ test (B, C, E). *$P < 0.01$, **$P < 0.005$, ***$P < 0.001$, ****$P < 0.0001$. Source data are available online for this figure.

Finally, GM-CSF-Mφ and M2 cells contained larger amounts of mRNA and protein for glycogen synthase isoform 1 (GYS1), the key enzyme controlling glycogenesis, than monocytes (Figs. 1H,I and EV1D). The levels of mRNA for other enzymes involved in glycogenesis, such as phosphoglucomutase isoforms 1 and 2 (PGM1 and PGM2), UDP-glucose pyrophosphorylase 2 (UGP2), and skeletal muscle glycosyltransferase glycogenin isoforms (GYG1), were also highly upregulated in GM-CSF-Mφ and M2 cells relative to monocytes (Fig. 1H,I). The hepatic GYS2 and GYG2 isoforms remained undetectable.

Taken together, these results show that the presence of the inflammatory cytokine GM-CSF or of a combination of M-CSF plus IL-4 triggers glycogenesis in human macrophages.

## A combination of GM-CSF plus IFNγ promotes human macrophage glyconeogenesis

The liver and kidneys can synthesize G6P and glucose from non-carbohydrate substrates, such as glycerol, lactate, and the glycogenic amino acids glutamine (Gln) and alanine (Ala) (Adeva-Andany et al, 2016; Meyer et al, 2002; Stumvoll et al, 1998). Macrophages accumulate in damaged and inflamed tissues, in which glucose concentrations may be low (Geeraerts et al, 2017; Okabe and Medzhitov, 2016). In inflamed tissues, the recruited monocytes come into contact with GM-CSF and IFNγ (produced by activated leukocytes), promoting their differentiation into inflammatory M1 cells. Although GM-CSF and M-CSF plus IL-4 induce macrophage glycogenesis (Fig. 1F), we hypothesized that some macrophage subtypes, and especially M1 cells (generated in the presence of GM-CSF plus IFNγ), may perform glyconeogenesis in a nutrient-poor environment.

GM-CSF-Mφ (orange), M1 cells (green) and M2 cells (purple) were generated in CM, which was replaced on day 2 with Glc$^{low}$ Gln$^{low}$ medium, with or without supplementation with Glc, Gln, lactic acid, glycerol, Ala, or pyruvate (Fig. 2A); the culture of cells in Glc$^{low}$ Gln$^{low}$ medium did not impact the acquisition of M1 and M2 specific markers (Fig. EV1A,B). Intracellular glycogen levels were determined on days 2 and 5 in each set of conditions. On day 2, GM-CSF-Mφ and M-CSF-Mφ (gray) had a low glycogen content (Fig. 2A). As a control, all three macrophage subtypes were found to have accumulated glycogen on day 5 in Glc-supplemented medium but not in non-supplemented Glc$^{low}$ medium (none) (Fig. 2A).

By contrast to our findings for GM-CSF-Mφ and M2 cells, the glycogen content of day 5 M1 cells was found to be markedly higher if Gln, lactic acid, or glycerol was added to the medium (Fig. 2A), indicating that M1 cells can use non-carbohydrate substrates to generate glycogen stores. Unlike IFNγ, the cytokines IL-1β, IL-10, and IL-4 were unable to unlock GM-CSF-Mφ glyconeogenesis, although small amounts of glycogen were synthesized from Gln in the presence of IL-4 (Fig. EV2A). The addition of IFNγ during M2 cell differentiation (M-CSF + IL-4) did not lead to glyconeogenesis in these cells (Fig. EV2A).

We confirmed the occurrence of glyconeogenesis in M1 cells by generating these cells in Glc$^{low}$ Gln$^{low}$ medium in the presence of $^{13}C_5$-Gln or $^{13}C_3$-lactic acid before monitoring $^{13}$C-Glc content in glycogen by GC-MS. $^{13}$C-enriched glycogen was detected in cells cultured with Gln and lactic acid, evidencing the use of both molecules as alternative substrates for glyconeogenesis by M1 cells (Fig. 2B). The results showed that at day 5, the glycogen synthesis flux for $^{13}C_3$-lactic acid or $^{13}C_5$-Gln (Fig. 2B) were low, while glycogen levels were equivalent to those found in macrophages cultured under optimal conditions (i.e., in the presence of glucose) (Fig. 2A). Such differences may relate to isotopic dilution by pre-existing metabolic pools and simultaneous metabolic pathways, leading to a relatively low percentage of $^{13}$C in intermediates (explaining the apparent low allocation of $^{13}$C to glycogen).

Phosphoenolpyruvate carboxykinase (PCK) and fructose-1,6-bisphosphatase (FBP) are two key enzymes required for Gln-, lactic acid-, and glycerol-dependent glyconeogenesis (Fig. 2C). GM-CSF-Mφ and M1 cells overexpressed FBP1 relative to other Mφ subtypes, at both the mRNA and protein levels (Fig. 2D,E). FBP1 activity is dependent on its substrate, fructose-1,6-biphosphate, which is generated via the PCK/PEP pathway (Timson, 2019). Interestingly, mRNA and protein for the mitochondrial PCK2 were also present in much larger amounts in M1 cells than in GM-CSF-Mφ (Fig. 2D,E). IFNγ was the only cytokine tested that upregulated PCK2 in GM-CSF-Mφ in the presence of Glc or a non-carbohydrate substrate (Fig. EV2B). In M1 cells, fructose-1,6-biphosphate generated via the PCK2/PEP pathway in the presence of IFNγ probably activates FBP1. Finally, the mRNAs encoding FBP2 and the cytosolic PCK1 isozyme (expressed in the liver and kidney) were undetectable in these macrophage subsets (Fig. EV2C).

The use of inhibitors to block glyconeogenesis in M1 cells significantly decreased intracellular glycogen levels in these cells,

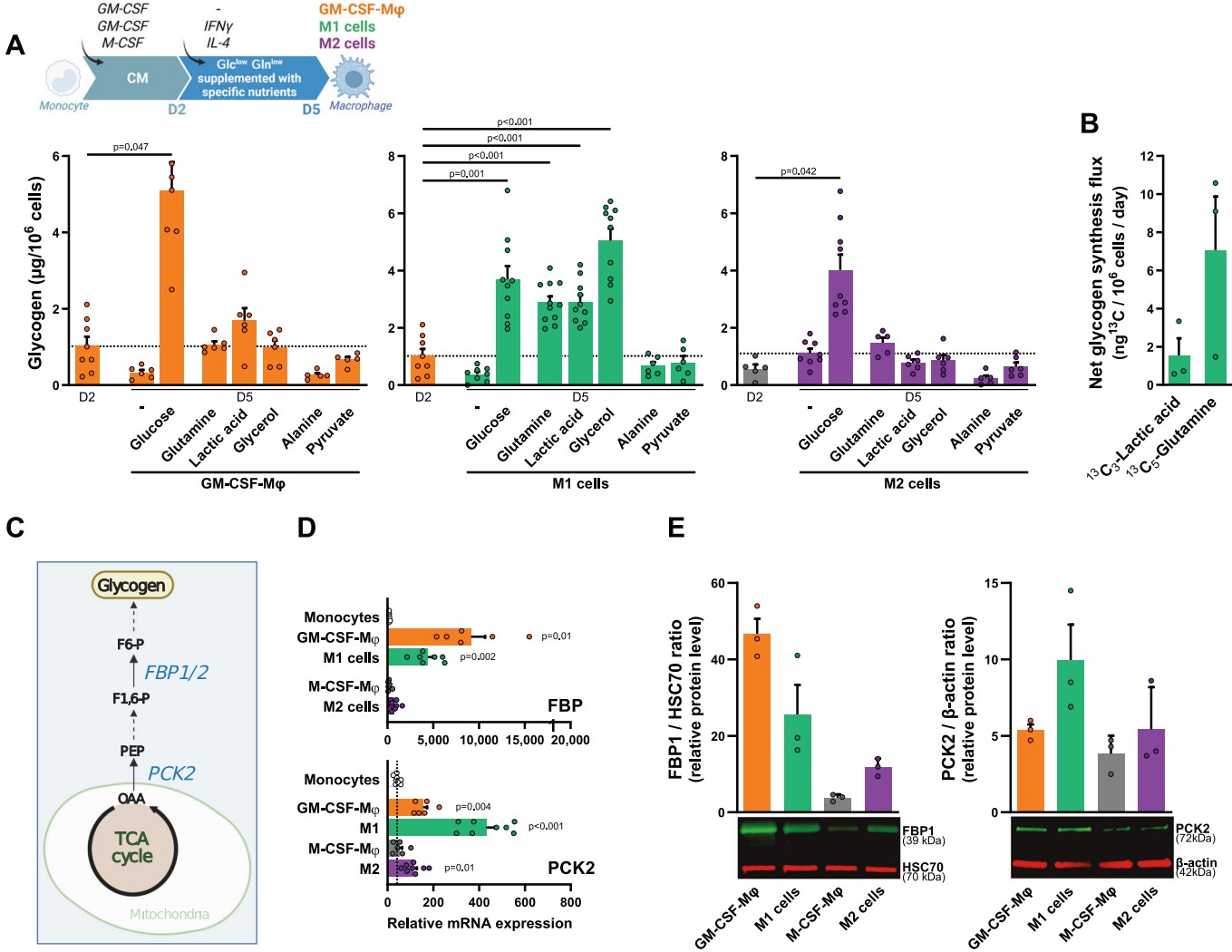

**Figure 2. The combination of GM-CSF plus IFNγ promotes human macrophage glyconeogenesis.**

Timelines summarize experimental procedures for macrophage generation. (**A**) Monocytes were differentiated into GM-CSF-Mφ, M1 or M2 cells in CM, with a switch on day 2 to Glc$^{low}$ Gln$^{low}$ medium supplemented or not with nutrients (Glc, Gln, lactic acid, glycerol, Ala, pyruvate). Glycogen was quantified on day 2 and day 5 Mφ ($n = 5$–10). (**B**) Monocytes were differentiated into M1 cells as in (**A**) in a medium supplemented with $^{13}C_3$-lactic acid or $^{13}C_5$-Gln. At day 5, glycogen was purified and hydrolyzed to Glc before GC-MS analysis ($n = 3$). (**C**) Schematic diagram of the enzymatic pathways involving PCK2 and FBP1 enzymes in glyconeogenesis. (**D**) PCK2 and FBP1 mRNA expression determined by RT-qPCR in monocytes and day 5 Mφ ($n = 5$–8). (**E**) Protein expression of PCK2 and FBP1 in day 5 Mφ. Results are expressed as FBP1/HSC70 and PCK2/β-actin band intensity ratios (representative western blots are shown from three independent experiments). Values are represented as the mean ± SEM ($n = 3$), each dot represents a different donor. Statistical significance was determined by Welch's ANOVA test followed by Dunnett's multiple comparison post hoc test. The P values in (**D**) represent the comparison between monocytes and macrophages. *$P < 0.01$, **$P < 0.005$, ***$P < 0.001$, ****$P < 0.0001$. Source data are available online for this figure.

regardless of the substrate (Gln, lactic acid or glycerol) (Fig. 3A,B). M1 cells displayed high levels of expression for genes encoding enzymes involved in TCA-dependent Gln metabolism, such as glutaminase (GLS), succinate dehydrogenase (SDH), and malate dehydrogenase 2 (MDH2) (Fig. EV3A,B). The inhibition of GLS, glutamic-oxaloacetic transaminase 1 (GOT1) and SDH by CB-839, GOT1i and NV-161, respectively, prevented Gln-dependent glycogen accumulation in M1 cells (Fig. 3A,B; Table 1). M1 cells strongly expressed genes encoding enzymes involved in lactate-dependent glyconeogenesis, such as lactate dehydrogenases A and B (LDHA/B) and pyruvate carboxylase (PC) (Fig. EV3A,B). The inhibition of the monocarboxylate transporters MCT and LDH by CHC and

GSK2837808, respectively, prevented lactic acid-dependent glycogen accumulation in M1 cells (Fig. 3A,B; Table 1). We used lactic acid rather than sodium lactate because MCT is a lactate-proton symporter requiring a low extracellular pH to function (Colegio et al, 2014; Paolini et al, 2020). Finally, M1 cells also strongly expressed enzymes involved in glycerol metabolism: aquaglyceroporin 9 (AQP9), which mediates glycerol uptake, and glycerol kinase (GK), which converts glycerol to P-glycerol (Fig. EV3A,B).

The blocking of M1 cell glyconeogenesis with PCK2 or FBP1 inhibitors (PEPCKi and MB05032, respectively) reduced glycogen content regardless of the substrate (Fig. 3A,B; Table 1). The use of siRNAs directed against PCK2 or FBP1 also decreased

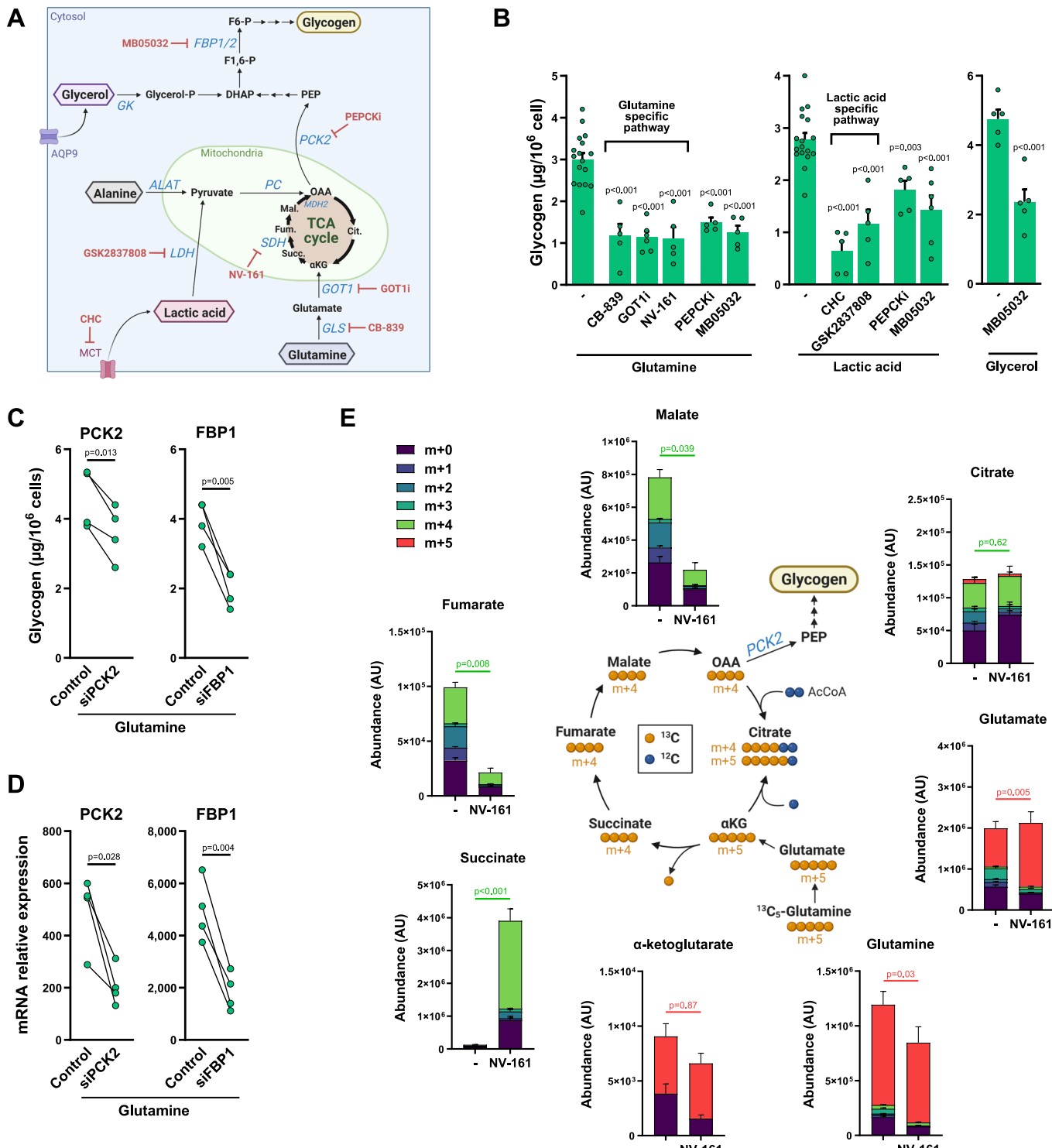

Gln-dependent glycogen stores in M1 cells (Fig. 3C,D). We then used $^{13}C_5$-Gln to investigate the metabolic pathways associated with Gln glyconeogenesis in M1 cells. Following incubation with $^{13}C_5$-Gln, an enrichment in $^{13}C$ (appearance of $^{13}C_4$ isotopologs) was noted for α-ketoglutarate, succinate, fumarate, and malate (Fig. 3E). $^{13}C$-succinate content was significantly increased in the presence of the succinate dehydrogenase (SDH, complex II) inhibitor NV-161,

whereas the levels of fumarate and malate were significantly decreased (Fig. 3E), indicating that M1 cells used Gln as an anapleurotic substrate for the TCA cycle. These findings indicate that the conventional TCA cycle is involved in Gln glyconeogenesis in M1 cells.

We then evaluated whether M1 cells can synthesize glycogen using concomitantly glycogenesis and glyconeogenesis. Day 5 M1 cells were

**Figure 3.   M1 cells glyconeogenesis pathways.**

(A) Main metabolites and enzymatic pathways involved in glyconeogenesis. Specific inhibitors used in this study are in red. (B) Glycogen content in day 5 M1 cells generated as described in Fig. 2A in culture medium supplemented with Gln, lactic acid or glycerol, in the absence or presence of enzyme inhibitors ($n = 5–16$). (C, D) Day 2 GM-CSF-Mφ were switched to $Glc^{low}$ $Gln^{low}$ medium containing GM-CSF, IFNγ and PCK2-targeting or FBP1-targeting siRNA, or a control siRNA. Gln was added 2 days later. Glycogen quantification (C) and relative mRNA expression (D) were measured at day 7 ($n = 4$). (E) Day 2 GM-CSF-Mφ were switched to $Glc^{low}$ $Gln^{low}$ medium containing GM-CSF and IFNγ. At day 3, cells were treated or not with the SDH inhibitor NV-161 and supplemented with $^{13}C_5$-Gln for 30 min. The relative abundance of $^{13}C$-metabolites was determined by LC-MS. Results are expressed as relative abundance of each metabolite generated after Gln metabolism (arbitrary unit or AU). Values are represented as the mean ± SEM ($n = 8$); each dot represents a different donor. Statistical significance was determined by Welch's ANOVA test followed by Dunnett's multiple comparison post hoc test (B), or by two-tailed paired $t$ test (C–E). The P values in (B) represent the comparison between each inhibitor and the control without inhibitor. *$P < 0.01$, **$P < 0.005$, ***$P < 0.001$, ****$P < 0.0001$. Source data are available online for this figure.

**Table 1.   Analysis of the effect of NV-161, CB-839, GOT1i, PEPCKi, MB05032, CHC, GSK2837808, CP-91149, and 6AN on cell viability.**

| Compound | Concentration | CAS number | Function | Cell viability (%) | Ref |
|---|---|---|---|---|---|
| NV-161 | 100 μM | N/A | SDH inhibitor | 102.2 ± 6.2 | Ehinger et al, 2016 |
| CB-839 | 200 nM | 1439399-58-2 | GLS inhibitor | 104.3 ± 5.8 | Zimmermann et al, 2019 |
| GOT1i | 20 μM | 732973-87-4 | GOT1 inhibitor | 101.0 ± 6.1 | Anglin et al, 2018; Yoshida et al, 2020 |
| PEPCKi | 10 μM | 628279-07-2 | PCK2 inhibitor | 112.1 ± 3.8 | Montal et al, 2019 |
| MB05032 | 50 μM | 261365-11-1 | FBP1/2 inhibitor | 99.6 ± 8.1 | Sadiku et al, 2021; Zhang et al, 2010 |
| CHC | 125 μM | 28166-41-8 | MCT inhibitor | 103.8 ± 11.6 | Paolini et al, 2020 |
| GSK2837808 | 15 μM | 1445879-21-9 | LDH inhibitor | 98.0 ± 10.3 | Paolini et al, 2020 |
| CP-91149 | 50 μM | 186392-40-5 | PYG inhibitor | 95.9 ± 11.1 | Sadiku et al, 2021; Thwe et al, 2017; Ma et al, 2020 |
| 6AN | 20 μM | 329-89-5 | G6PD inhibitor | 93.3 ± 5.9 | Ma et al, 2020 |

SDH succinate dehydrogenase, GLS glutaminase, GOT1 glutamic-oxaloacetic transaminase 1, PCK2 phosphoenolpyruvate carboxykinase 2, FBP1/2 fructose-1,6-bisphosphatase 1/2, MCT monocarboxylate transporters, LDH lactate dehydrogenase, PYG glycogen phosphorylase, G6PD glucose-6-phosphate dehydrogenase. Cell viability was assessed at day 5 by flow cytometry (MACSQuant analyzer, Miltenyi Biotec) after staining with 7-AAD to discriminate dead cells from living cells. Results are expressed as a percentage of viable cells normalized against non-treated condition for each donor (mean ± SD, $n = 5$).

generated in CM containing 2 g/L Glc and either 2 or 20 mM Gln. Glycogen stores were higher in cells cultured with 20 mM Gln compared to cells cultured with 2 mM Gln, suggesting that they can use Gln to synthesize glycogen even when Glc is available (Fig. EV2D). In addition, glycogen levels in M1 cells were lower in the presence of a PCK2 inhibitor, regardless of Gln concentration (Fig. EV2D), suggesting that M1 cells can use simultaneously Glc and non-carbohydrate substrates to synthesize glycogen.

Finally, supplementation with Ala and pyruvate did not lead to glyconeogenesis in M1 cells (Fig. 2A). For Ala, this lack of glyconeogenesis induction may be due to the low levels of alanine aminotransferase (ALAT), which converts alanine to pyruvate, in M1 cells (Fig. EV3A,B). Pyruvate import via MCT transporters compensates for the loss of carboxylate upon lactate release (Halestrap and Price, 1999). These experimental conditions ($Glc^{low}$ $Gln^{low}$ medium under ambient $O_2$ partial pressure) were not favorable for lactate generation, potentially accounting for the failure of M1 cells to metabolize pyruvate.

In conclusion, only the concomitant presence of the inflammatory cytokines GM-CSF and IFNγ appears to be able to unlock human macrophage glyconeogenesis from Gln, lactic acid, or glycerol.

## Tumor-associated macrophages store glycogen and are capable of glyconeogenesis

We investigated whether established tumor-associated macrophages (TAM) could also synthesize and store glycogen. We found that CD14$^+$ cells isolated from human ovarian cancer ascites, which

consist mostly of macrophages (Worzfeld et al, 2018; Steitz et al, 2020; Reinartz et al, 2016) had minimal glycogen stores (Fig. 4A). Consistent with this observation, ascites fluids had detectable levels of Glc ($< 5$ mM), and low levels of IL-4 and IFNγ ($< 10$ pg/mL), and GM-CSF ($< 100$ pg/mL), which are required for glycogenesis and glyconeogenesis by human macrophages. GM-CSF, IFNγ, and high concentrations of lactic acid are present in established solid tumors (Zhou et al, 2021). We observed that stimulation with GM-CSF plus IFNγ led to the synthesis and storage of glycogen from Glc or lactic acid in ascites fluid TAMs (Fig. 4A), together with an upregulation of PCK2 expression (Fig. 4B).

In ovarian tumors, high levels of mucins, responsible for amylase-resistant PAS staining, prevented the analysis of glycogen in tissue macrophages. We therefore evaluated the presence of glycogen in TAMs infiltrating lung adenocarcinomas, which contain only low levels of mucins. As expected, lung adenocarcinoma was massively infiltrated by CD163-positive macrophages (large cells with abundant, vacuolated cytoplasm and a round, regular nucleus containing a small nucleolus) (Fig. 4C). In the same tissue sections, PAS staining, which was abolished by α-amylase, revealed the presence of glycogen in TAM (Fig. 4C). Moreover CD163-positive cells were not stained with TTF-1, a marker of adenocarcinoma cells and type 2 pneumocytes that do not contain glycogen (Fig. EV4). The presence of glycogen in macrophages was confirmed by FTIR spectroscopic imaging in serial sections. FITR spectroscopy also revealed a colocalization of glycogen with lactic acid ($R^2 = 0.87$, $P < 0.0001$) (Fig. 4D). Given that GM-CSF and IFN-γ, along with elevated lactic acid, are present in solid tumors, this

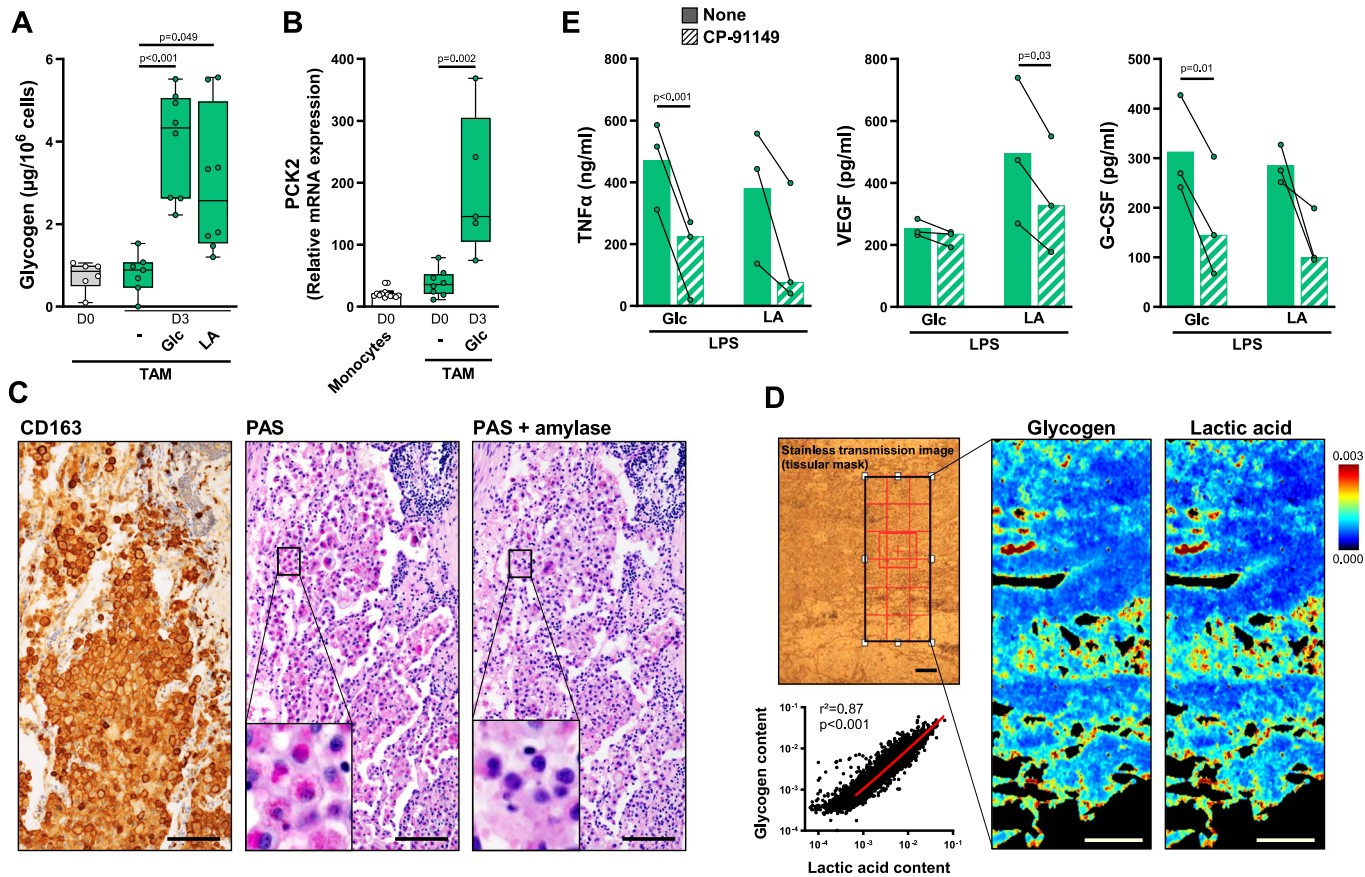

**Figure 4. Tumor-associated macrophages store glycogen and are capable of glyconeogenesis.**

(A, B) TAM isolated from ovarian cancer ascites were cultured in Glc$^{low}$ Gln$^{low}$ medium containing GM-CSF and IFNγ supplemented with Glc or lactic acid. Glycogen (**A**; $n = 6$–8) and relative mRNA expression of PCK2 (**B**; $n = 5$–13) were quantified in freshly purified TAM and after 3 days of culture. The boxplots display a median line, interquartile range (IQR) boxes, min to max whiskers. (**C**) Immunohistochemical analysis of CD163 expression in lung adenocarcinoma. On the same tissue section, glycogen was detected by PAS staining, after treatment or not with amylase; scale bar, 100 µm. (**D**) Lactic acid and glycogen quantification by FTIR spectroscopic imaging in lung adenocarcinoma. Left panel: unstained bright field image; right panel, glycogen and lactic acid maps; scale bar, 200 µm; linear correlation between lactic acid and glycogen contents in infiltrating lung adenocarcinoma was calculated (results are representative 1 out of 2 biologically independent experiments). (**E**) Day 3 TAM were treated or not with CP-91149 for 15 min before a 6 h stimulation with LPS. TNFα, VEGF and G-CSF were quantified by ELISA ($n = 3$). Values are represented as the mean ± SEM, each dot represents a different donor. Statistical significance was determined by Welch's ANOVA test followed by Dunnett's multiple comparison post hoc test (**A**), or by two-tailed unpaired Welch $t$ test (**B**) or by paired $t$ test (**E**). *$P < 0.01$, **$P < 0.005$, ***$P < 0.001$, ****$P < 0.0001$. Source data are available online for this figure.

finding suggest that TAMs can use lactic acid in vivo to generate glycogen (Fig. 4D). Thus, human TAMs synthesize and store glycogen and are capable of glycogenesis and glyconeogenesis.

### Activated M1 and M2 cells catabolize glycogen

We then investigated the possible association of glycogen metabolism with macrophage function. Human macrophages require stimulation, with TLR ligands such as LPS for example, to reveal their functions and to induce the production of cytokines (Izquierdo et al, 2015). As previously reported, the stimulation of day 5 M1 and M2 cells with LPS induced an increase in extracellular Glc consumption and of glycolytic activity (Huang et al, 2014; Covarrubias et al, 2016; Izquierdo et al, 2015; Na et al, 2016). However, Glc availability is generally low at the injured/inflamed sites at which macrophages function. We therefore compared glycogenolysis by stimulated Mφ in the presence and absence of extracellular Glc.

Day 5 M1 and M2 cells were stimulated with LPS in CM or Glc$^{low}$ medium, and glycogen was quantified 6 h later. LPS stimulation in CM resulted in a sharp decrease in glycogen content in M1 and M2 cells (Fig. 5A). This decrease was even more pronounced when LPS stimulation occurred in Glc$^{low}$ medium (Fig. 5A).

Glycogen storage results from a balance between synthesis and catabolism in favor of synthesis (Adeva-Andany et al, 2016). We therefore evaluated the impact on glycogen stores in stimulated M1 and M2 cells of glycogenolysis inhibition with a glycogen phosphorylase (PYG) inhibitor, CP-91149, at a non-toxic concentration (Fig. 5B; Table 1). The inhibition of glycogenolysis prevented the LPS-induced decrease in glycogen levels in M1 and M2 cells (Fig. 5A), demonstrating that LPS stimulation enhances glycogenolysis. Similarly, live *E. coli* also induced a decrease of glycogen stores in M1 and M2 cells that was prevented by PYG inhibition (Fig. EV5A). All Mφ subtypes contained both mRNA

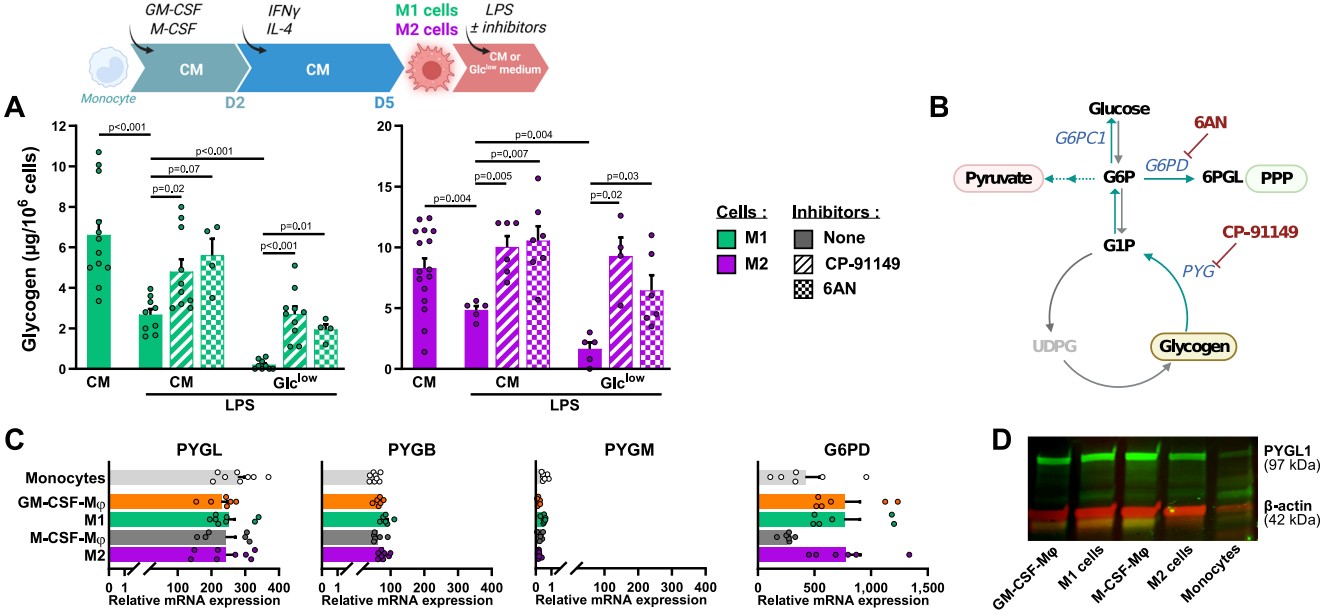

**Figure 5. Activated M1 and M2 cells catabolize glycogen.**

(A) Experimental procedure for LPS-stimulated macrophages. Glycogen was quantified in day 5 M1 and M2 cells treated or not with CP-91149 or 6AN, 15 min before 6 h LPS stimulation (n = 5–16). (B) Overview of the three glucose metabolic pathways (pyruvate synthesis, glycogen metabolism, pentose phosphate pathway or PPP). The targets of the inhibitors CP-91149 and 6AN are indicated. (C) PYGL, PYGB, PYGM and G6PD mRNA expression was determined by RT-qPCR in monocytes and day 5 Mφ (n = 5–8). (D) PYGL expression was analyzed by western blotting in monocytes and day 5 macrophages, with actin as loading control (lower, representative of one out of 3). Values are represented as the mean ± SEM (n = 3), each dot represents a different donor. Statistical significance was determined by Welch's ANOVA test followed by Dunnett's multiple comparison post hoc test. *P < 0.01, **P < 0.005, ***P < 0.001, ****P < 0.0001. Source data are available online for this figure.

and protein for glycogen phosphorylase (PYG), the key enzyme required for glycogen degradation, mostly as the liver isoform (PYGL) (Fig. 5C,D), but also, to a lesser extent, as the brain isoform (PYGB) (Fig. 5C). By contrast, the muscle isoform (PYGM) was not detected (Fig. 5C). Thus, LPS stimulation increases glycogenolysis in M1 and M2 cells, even in the presence of extracellular Glc, demonstrating a non-redundant role of extracellular glucose and glycogen stores.

## Glycogen sustains M1 and M2 cell functions

Murine dendritic and M1 cells have recently been shown to use glycogenolysis to support their functions (Ma et al, 2020; Thwe et al, 2017). We observed that PYG inhibition with chemical inhibitors or siRNA significantly decreased IL-1β, IL-6, IL-12, and TNFα production by LPS-stimulated human M1 cells (Fig. 6A–C), IL-10 and IL-6 production by LPS-stimulated M2 cells (Fig. 6D) and also IL-12 and IL-10 production by *E. coli*-stimulated M1 and M2 cells, respectively (Fig. EV5B). The inhibition of glycogenolysis also reduced cytokine production by M1 cells with glycogen stores generated from Gln, lactic acid or glycerol (Fig. EV5C) and by TAM with glycogen stores generated from Glc or lactic acid (Fig. 4E) and finally reduced polystyrene microspheres and of pHrodo *E. coli* phagocytosis by M2 cells (Figs. 6D and EV5D).

These effects were observed in both Glc^low^ medium and Glc^rich^ medium (CM) (Fig. 6A,D) demonstrating that the effector functions of M1 and M2 cells are dependent on glycogenolysis rather than extracellular glucose availability. These observations are

consistent with recent findings showing that inflammatory cytokine production by human monocytes and DCs and murine alveolar macrophages is not affected by Glc starvation (Woods et al, 2020; Otto et al, 2021; Thwe et al, 2017).

The glucose-6 phosphate (G6P) produced by glycogenolysis can fuel various metabolic pathways (PPP, pyruvate synthesis and Glc release) (Fig. 5B). Glycogenolysis-derived G6P was recently reported to sustain murine M1 cell functions, fueling oxidative PPP, which begins with the oxidation of G6P by glucose-6-phosphate dehydrogenase (G6PD) (Ma et al, 2020). Human M1 and M2 cells express G6PD (Fig. 5C). We therefore evaluated the impact of a non-toxic concentration of the G6PD inhibitor 6AN (Table 1) on cytokine production by M1 and M2 cells. Incubation with 6AN limited LPS-induced glycogenolysis in M1 and M2 cells (Fig. 5A), and decreased the production of IL-1β, IL-6, IL-12, and TNFα by M1 cells (Fig. 6A) and the production of IL-10, but not IL-6, by M2 cells (Fig. 6D). The use of siRNA directed against G6PD to block PPP also led to a decrease in IL-1β and TNFα production by LPS-stimulated human M1 cells (Fig. 6B,C). These results suggest that glycogenolysis enables human M1 and M2 cells to fulfill their functions, at least in part through PPP.

In the liver and kidney, the G6PC1 system catalyzes the dephosphorylation of G6P to Glc, a step required for the release of Glc from cells into the bloodstream (Hutton and O'Brien, 2009). Under our experimental conditions, no Glc release by M1 and M2 cells was detected. This absence of Glc release was corroborated by the absence of G6PC1 mRNA (Fig. EV5E). Only low levels of the mRNA encoding G6PC isoform 3 were detected; this ubiquitously

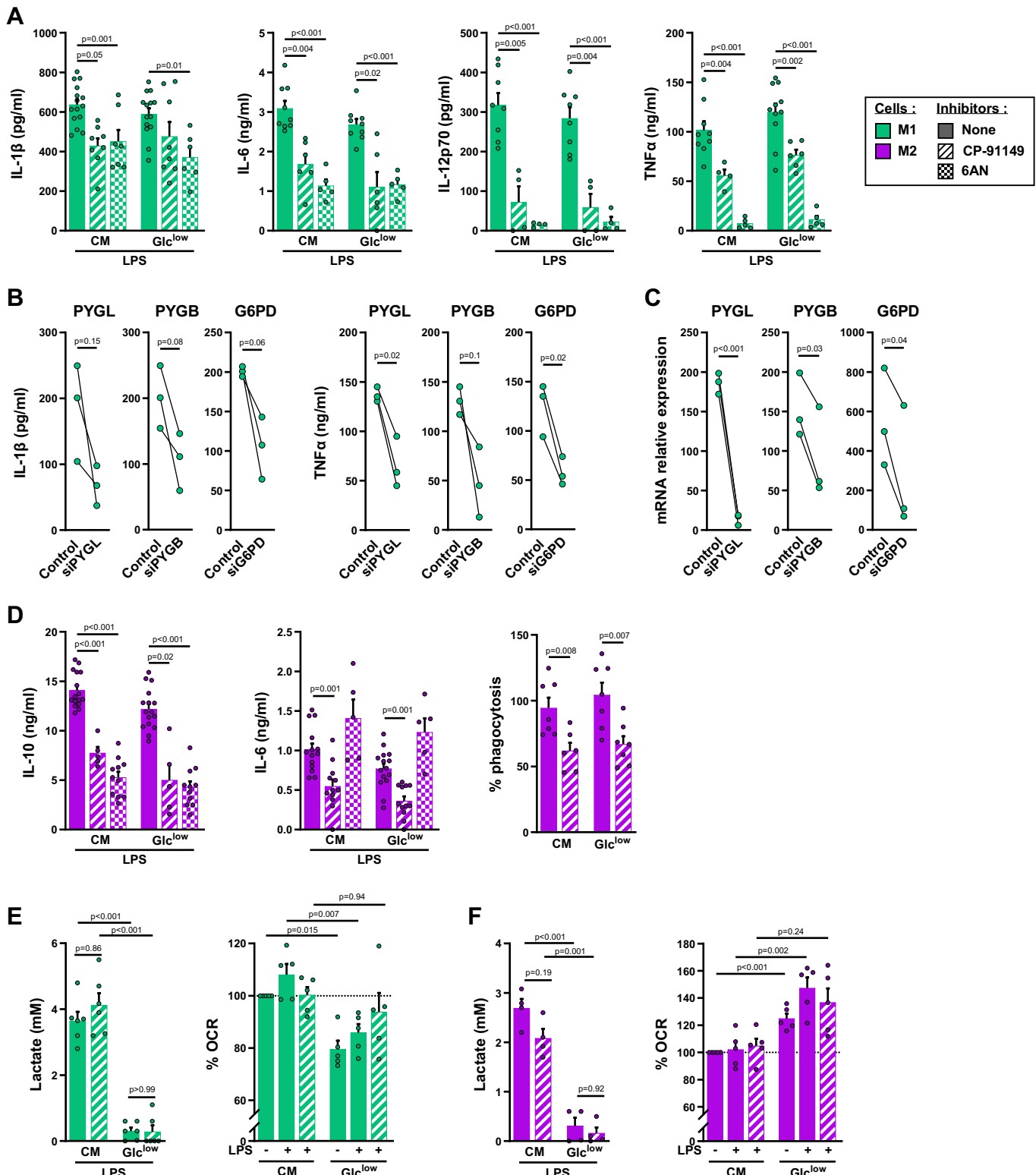

expressed isoform hydrolyzes phosphates of sugars other than G6P (Fig. EV5E) (Martin et al, 2002; Veiga-da-Cunha et al, 2019).

Finally, the inhibition of glycogenolysis had no effect on glycolysis (as assessed by quantifying lactate production) and mitochondrial respiration

(as assessed by measuring oxygen consumption) in LPS-stimulated M1 (Fig. 6E) and M2 cells (Fig. 6F), suggesting that macrophage respiratory metabolism is powered by extracellular Glc rather than glycogen degradation products.

**Figure 6. Glycogen sustains M1 and M2 cell functions.**

(A) Day 5 M1 cells were cultured in CM or in Glc^low medium and treated or not with CP-91149 or 6AN, 15 min before LPS stimulation. Cytokines were quantified by ELISA in the M1 cells culture supernatants after 2 h (TNFα and IL-6), 6 h (IL-1β) or 24 h (IL-12p70) activation with LPS ($n = 5$–14). (B, C) Day 2 GM-CSF-Mφ were incubated either with siRNA targeting PYGL, PYGB, G6PD or a control siRNA. Cells were then stimulated with LPS at day 5, and cytokines were quantified by ELISA in M1 cell culture supernatants after 2 h (TNFα) or 6 h (IL-1β) (B) PYGL, PYGB and G6PD mRNA expression were determined by RT-qPCR in day 5 macrophages ($n = 4$) (C). (D) Day 5 M2 cells were cultured in CM or in Glc^low medium and treated or not with CP-91149, 15 min before LPS stimulation. Cytokines were quantified by ELISA in the M2 cells culture supernatants after 2 h (IL-6), 6 h (IL-10) activation with LPS ($n = 5$–14). Phagocytosis was assessed by flow cytometry after 3 h LPS stimulation ($n = 6$). (E, F) Lactate was quantified in 6 h LPS-stimulated M1 (E) and M2 (F) cells culture supernatants and oxygen consumption rate (OCR) of M1 and M2 cells was monitored after 2 h ($n = 4$–6). Values are represented as the mean ± SEM, each dot represents a different donor. Statistical significance was determined by Welch's ANOVA test followed by Dunnett's multiple comparison post hoc test or two-tailed unpaired Welch $t$ test for phagocytosis assay in (D). *$P < 0.01$, **$P < 0.005$, ***$P < 0.001$, ****$P < 0.0001$. Source data are available online for this figure.

Thus, glycogen mobilization via glycogenolysis and PPP is a key metabolic pathway supporting the functions of human M1 and M2 cells regardless of the availability of extracellular glucose.

## Discussion

Macrophages restore tissue homeostasis following injury or infection by controlling tissue repair and inflammatory responses. Tissue damage is often associated with poor nutrient availability. We therefore investigated the strategies used by these firefighting cells to ensure the performance of their functions under difficult conditions. We investigated the ability of human macrophages to store glycogen, an intracellular carbon and energy reserve. We show here that the presence of GM-CSF or of a combination M-CSF plus IL-4 (M2 cells) triggers glycogenesis and glycogen storage by macrophages, whereas a combination of the inflammatory cytokines GM-CSF plus IFNγ (inflammatory M1 cells) induces glyconeogenesis from Gln, lactic acid and glycerol. Finally, we show that glycogenolysis is essential for macrophage functions, even in the presence of extracellular Glc (Fig. 7). This ability of human macrophages to metabolize glycogen and to resort to glyconeogenesis under inflammatory conditions sheds light on the mechanisms by which these cells manage to meet tissue demands even in nutrient-poor injured tissues.

Our results suggest that GM-CSF and the M-CSF plus IL-4 combination activate GYS, a key enzyme in glycogenesis, the activity of which is controlled by various protein kinases, including GSK3 (Mccorvie et al, 2022). By inactivating GSK3, the Akt kinase activates GYS, thereby enabling glycogenesis (Bultot et al, 2012). GM-CSF and IL-4 are known to activate Akt (Chang et al, 2012; Hercus et al, 2009; Li et al, 2020). Human dendritic cells generated from monocytes by incubation with GM-CSF and IL-4 can also perform glycogenesis (Thwe et al, 2017). Based on our observations, we suspect that the lack of glycogen reserves reported by others in murine M2 cells generated with M-CSF and IL-4 could be due to an insufficient incubation time with IL-4, which was limited to only 24 h (Ma et al, 2020).

In response to stimulation, many cell types can produce GM-CSF, but the expression of this cytokine is restricted to sites of inflammation (Hamilton, 2019; Jeannin et al, 2018). IFNγ is produced by activated infiltrating T and NK cells (Mah and Cooper, 2016). GM-CSF and IFNγ are, therefore, concomitantly expressed at sites of inflammation. We found that the combination of GM-CSF plus IFNγ triggered glycogen synthesis from Gln, lactate, and glycerol by M1 cells, whereas this capacity is generally attributed to

hepatocytes (Adeva-Andany et al, 2016). M1 cells strongly express the key glycogenesis enzymes PCK2 and FBP1. GM-CSF increases levels of FBP1, which is present primarily in gluconeogenic organs and displays activity dependent on its substrate, fructose-1,6-biphosphate, which is generated by the PCK/PEP pathway (Timson, 2019). GM-CSF and IFNγ combination increases the mitochondrial expression of PCK2, leading to an increase in fructose-1,6-biphosphate production. PCK2 has also been implicated in glyconeogenesis in neutrophils (Sadiku et al, 2021). In conclusion, the cytokines present at sites of inflammation can trigger the storage of glycogen in macrophages, via FBP1 and PCK2, despite the sparsity of nutrients in the environment.

We found that M1 cells strongly expressed AQP9 and GK and used glycerol as a glyconeogenic substrate. We also observed that the levels of glycogen generated from glycerol in M1 cells were higher than those generated from Gln or LA. One possible explanation for this finding is that glycerol may be converted to glycogen more rapidly, as this process requires fewer enzymatic reactions and involves the expression of only one key enzyme, FBP1, whereas glyconeogenesis from Gln or LA involves the Krebs cycle and the expression of PCK2 and FBP1. Glycerol consumption and AQP9 expression by macrophages and dendritic cells have been described elsewhere (De Santis et al, 2018; Liu et al, 2009; Holm et al, 2015). The low levels of GK in GM-CSF-Mφ may account for these cells not using glycerol as a glyconeogenic substrate, despite the high levels of FBP1. Thus, under fasting conditions, the glycerol released from triglycerides in adipocytes may enable inflammatory macrophages to store glycogen as a source of energy for their functions.

We found that human M1 cells displayed SDH activity and metabolized Gln via succinate and malate, with the subsequent conversion of OAA to PEP by PCK2. By contrast, in murine M1 cells, the TCA cycle is disrupted at SDH (Jha et al, 2015). This difference can be explained by the generation of murine macrophages in the presence of M-CSF, IFNγ and LPS. Finally, glyconeogenesis from Gln has also been reported to occur via PCK2 in some tumor cells and neutrophils and via PCK1 in murine memory CD8[+] T cells (Ma et al, 2018; Sadiku et al, 2021; Vincent et al, 2015).

We show here that lactic acid acts as a substrate for glyconeogenesis via PCK2 in M1 cells. Glyconeogenesis from lactic acid via PCK2 has been reported in human neutrophils and lung cancer cells (Leithner et al, 2015; Sadiku et al, 2021). Lactic acid concentrations are high in injured tissues and in the tumor microenvironment (Zhou et al, 2021). GM-CSF and IFNγ, which are present in the microenvironment of established tumors, could

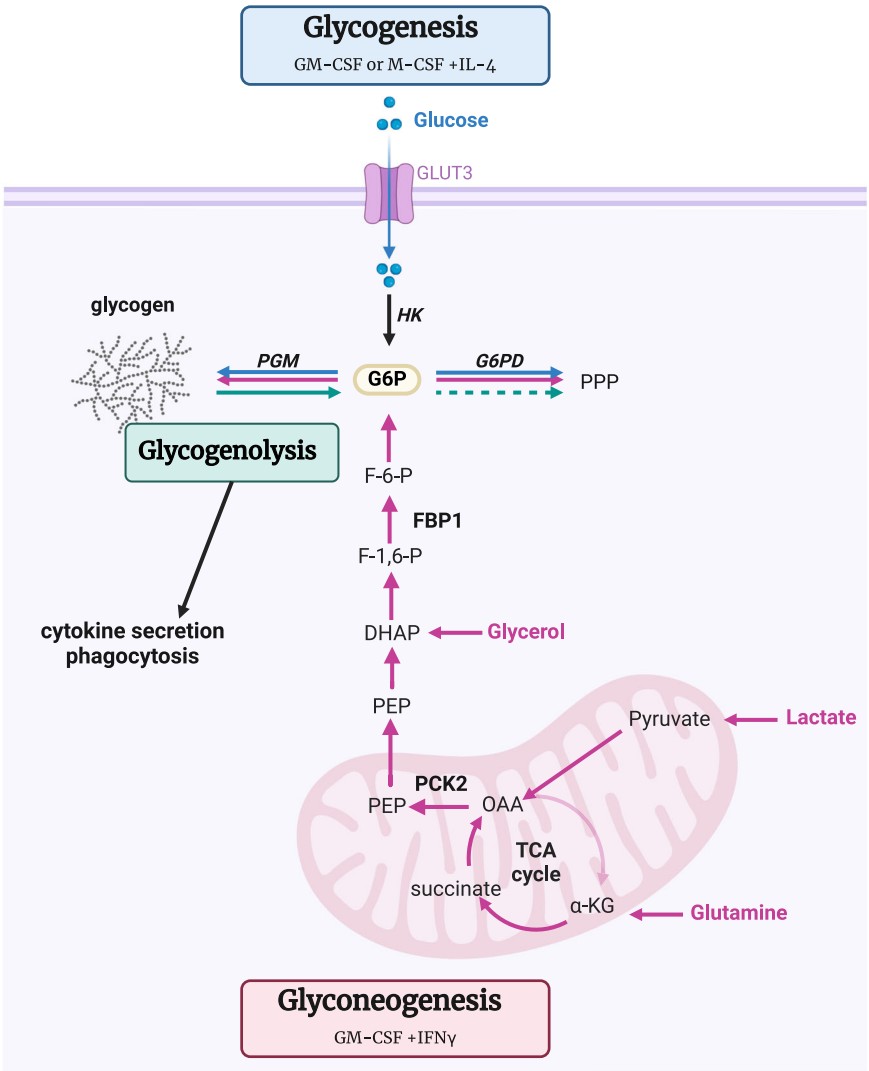

**Figure 7. Glycogen metabolism supports human macrophage functions.**

The combination of M-CSF + IL-4 (M2) and the presence of GM-CSF triggers glycogen synthesis from glucose (glycogenesis). The concomitant presence of the inflammatory cytokines GM-CSF and IFNγ appears unique to promote glycogen synthesis from Gln, LA or glycerol (glyconeogenesis). Glycogen stores are then channeled through glycogenolysis and further PPP to support cytokine secretion and phagocytosis by human macrophages.

therefore trigger glyconeogenesis from lactic acid by TAMs. We found that TAMs infiltrating lung adenocarcinomas contained glycogen and that a combination of GM-CSF and IFNγ triggered glycogen synthesis from lactic acid by TAMs from ovarian ascites. Finally, we and others have shown that lactic acid consumption by macrophages amplifies their tumor-promoting properties (Colegio et al, 2014; Paolini et al, 2020). Thus, by conferring a protumoral phenotype on TAMs and enabling them to store glycogen, lactic acid helps to maintain the detrimental effects of these cells in established tumors.

To conclude, our results suggest that in the absence of stimulation, macrophages synthesize and store glycogen from Glc. Under inflammatory conditions, macrophages acquire the ability to use additional substrates to synthesize glycogen.

We found that an absence of extracellular Glc did not affect cytokine production by stimulated human M1 and M2 cells.

Consistent with this finding, some myeloid cells, such as human monocytes and DCs, and murine alveolar macrophages, produce inflammatory cytokines despite Glc starvation, whereas murine bone marrow-derived macrophages do not (Murugina et al, 2020; Woods et al, 2020; Otto et al, 2021; Thwe et al, 2017; Raulien et al, 2017). Preliminary studies with the non-metabolizable Glc analog 2-DG have shown that Glc consumption by stimulated macrophages is required for inflammatory cytokine production (Tannahill et al, 2013; Van den Bossche et al, 2017; Zhao et al, 2017; Vijayan et al, 2019). However, it was subsequently reported that 2-DG impaired not only glycolysis but also OXPHOS activity, thereby inducing an unfolded protein response (Wang et al, 2018; Murugina et al, 2020; Xi et al, 2014; Kurtoglu et al, 2007).

Upon stimulation, the balance between glycogen synthesis and degradation shifts towards degradation, even if extracellular Glc is available. We show here that cytokine production by M1 and M2

cells and phagocytosis by M2 cells are dependent on glycogenolysis, even when extracellular Glc is non-limiting. The role of glycogenolysis seems to differ between human macrophages and DCs, as glycogenolysis in DCs has been implicated in cytokine production only in the absence of Glc and cell respiration (Thwe et al, 2017). Finally, in murine M1 cells, glycogenolysis-derived G6P is channeled to the PPP, leading to the generation of UDPG, a key regulator of cytokine production (Ma et al, 2020). We found that glycogenolysis also at least partially fuels PPP in human macrophages. These unexpected and intriguing results suggest that the PPP and glycolysis may be regulated differently. The PPP is known to be essential for macromolecular biosynthesis, the maintenance of cellular redox homeostasis, and cytokine synthesis (Nagy and Haschemi, 2015; Ham et al, 2013; Haschemi et al, 2012). Several hypotheses can be put forward to explain the regulation of the glycogen-PPP-cytokine synthesis axis independently of glycolysis: (i) a low availability of extracellular Glc may induce the production of citrate, thereby inhibiting phosphofructokinase (PFK1), the main regulator of glycolysis (Garland et al, 1963; Iacobazzi and Infantino, 2014); this would lead to an accumulation of glycogenolysis-derived G6P, with the flux diverted to the PPP; (ii) UDPG generated through glycogenolysis may activate G6PD, increasing the flux to the PPP, as reported in tumor cells (Rao et al, 2015); (iii) UDPG may fuel the PPP pathway (bypassing G6P) via the glucuronic acid pathway to form xylulose, an intermediate of the PPP pathway (Sommer et al, 2004; Mitchell et al, 2023). In conclusion, our results indicate that glycogen, rather than extracellular glucose, is essential for cytokine production and phagocytosis by human M1 and M2 cells.

These results reveal the metabolic plasticity of human macrophages, which depend on glycogenolysis for their function and can use non-glycogen substrates to synthesize glycogen under inflammatory conditions. They also shed light on the mechanisms by which macrophages meet tissue demands even in nutrient-poor environments.

# Methods

### Reagents and tools table

| Reagent/resource | Reference or source | Identifier/ catalog number |
|---|---|---|
| **Experimental models** | | |
| PBMC | French Blood Collection center | N/A |
| TAM | Angers University Hospital | N/A |
| **Antibodies** (Ab) | | |
| Anti-GYS1 Ab | Cell Signaling | pAb #3893 |
| Anti-PCK2 Ab | Cell Signaling | pAb #6924 |
| Anti-FBP1 Ab | Cell Signaling | mAb #59172 |
| Anti-PYGL Ab | Abcam | pAb #ab198268 |
| Anti-HSC70 Ab | Santa Cruz Biotechnology | mAb #sc-7298 |
| Anti-β-actin Ab | Proeintech | mAb #66009-1-Ig |
| IRdye 680 [RD] | Li-COR Biosciences | 926-68070 |
| IRdye 800 [CW] | Li-COR Biosciences | 826-32211 |

| Reagent/resource | Reference or source | Identifier/ catalog number |
|---|---|---|
| **Sequence-based reagents** | | |
| Control siRNA | Dharmacon | D-001910-10-05 |
| PCK2 siRNA | Dharmacon | E-006797-00-005 |
| FBP1 siRNA | Dharmacon | E-008725-00-0005 |
| PYGL siRNA | Dharmacon | E-009569-00-0010 |
| PYGB siRNA | Dharmacon | E-009587-00-0010 |
| G6PD siRNA | Dharmacon | E-008181-00-0010 |
| **Chemicals, enzymes, other reagents** | | |
| CD14+ magnetic beads | Miltenyi biotec | 5240404507 |
| Gln$^{low}$ RPMI 1640 medium | Corning | 10-043-CV |
| Glc$^{low}$ RPMI 1640 medium | Corning | 15-040-CV |
| Glc- and Gln-free medium | Sartorius | 01-101-1 A |
| Fetal calf serum | Eurobio | CVFSVF00-01 |
| HEPES | Lonza | 13-114E |
| Sodium pyruvate | Corning | 25-000-CI |
| Non-essential amino acids | Lonza | 13-114E |
| Penicillin/streptomycin | Lonza | DE17-602E |
| Glucose | Sigma-Aldrich | G8769 |
| ʟ-Glutamine | Lonza | 17-605E |
| Glycerol | Sigma-Aldrich | G5516 |
| Lactic acid | Sigma-Aldrich | L6402 |
| ʟ-Alanine | Alfa Aesar | J60279 |
| Recombinant human GM-CSF | Eurobio | 01-AR080 |
| Recombinant human M-CSF | Eurobio | 01-A0220 |
| Recombinant human IFNγ | Immunotools | 11343536 |
| Recombinant human IL-4 | R&D Systems | 6507 |
| Recombinant human IL-1β | Miltenyi Biotec | 130-093-895 |
| Recombinant human IL-10 | Immunotools | 11340105 |
| α-cyano-4-hydroxycinnamic acid (CHC) | Sigma-Aldrich | C2020 |
| GSK2837808A | Tocris | 5189 |
| CB-839 | MedChemExpress | 22038-5 |
| GOT1i | MedChemExpress | HY-122723 |
| NV-161 | MitoKit CII | 01-161-S2 |

| Reagent/resource | Reference or source | Identifier/ catalog number |
|---|---|---|
| PEPCKi | Santa Cruz Biotechnology | SC-496891 |
| MB05032 | MedChemExpress | HY-16307 |
| CP-91149 | Selleckchem | S271701 |
| 6-aminonicotinamide (6AN) | Cayman Chemical | 10009315 |
| LPS (E. coli serotype O111:B4) | Sigma-Aldrich | L2630 |
| Glycogen colorimetric assay kit | Biovision | AB65620 |
| Poly-L Lysine | Sigma-Aldrich | P2658 |
| 2-NBDG | Thermo Fischer Scientific | N13195 |
| DAPI | Sigma-Aldrich | D9542 |
| Hoescht 33342 | Sigma-Aldrich | H3570 |
| Schiff's reagent (PAS staining) | Thermo Fischer Scientific | J/7300/PB08 |
| Paraformaldéhyde | Thermo Fischer Scientific | 047377.9L |
| Sodium periodate | Sigma-Aldrich | S1878 |
| Ethanol | Thermo Fischer Scientific | E/0665DF/17 |
| Histolemon | Carlo Erba | 454911 |
| Entellan mounting medium | Sigma-Aldrich | 1.070960.0500 |
| FluoSpheres™ Carboxylate-Modified Microspheres | Invitrogen | F8826 |
| $^{13}C_6$-Glc | Sigma-Aldrich | 389374 |
| $^{13}C_3$-lactic acid | Sigma-Aldrich | 490040 |
| $^{13}C_5$-Gln | Cambridge Isotope Laboratories | CLM-1822-H-0.5 |
| Superscript II reverse transcriptase | Invitrogen | 100004925 |
| IL-6 (human) ELISA set | Diaclone | 851 520 020 |
| IL-1β (human) ELISA set | Diaclone | 851 610 020 |
| IL-10 (human) ELISA set | Diaclone | 851 540 020 |
| IL-12p70 (human) ELISA set | Diaclone | 851 570 020 |
| TNFα (human) ELISA set | Diaclone | 851 570 020 |
| VEGF (human) ELISA set | R&D Systems | DY293B-05 |
| G-CSF (human) ELISA set | R&D Systems | DY214 |
| Nucleospin RNA reagent kit | Macherey-Nagel | N/A |
| **Software** | | |
| ImageJ software | National Institutes of Health | N/A |
| Wave Pro 10.0.0 software | Agilent | N/A |
| PRISM (version 9.5.0) | www.graphpad.com | N/A |

| Reagent/resource | Reference or source | Identifier/ catalog number |
|---|---|---|
| **Other** | | |
| Nunc UpCell culture plates | Thermo Fischer Scientific | 174901/ 174900/ 174899/ 174898 |
| Lab-Tek chambered coverglass | Thermo Fischer Scientific | 155411 |
| Glucometer Accu-Chek | Roche Diagnostics | N/A |
| LightCycler 96 system | Roche Diagnostics | N/A |
| Leica DMR microscope | Leica | N/A |
| Leica SP8 confocal microscope | Leica | N/A |
| Leica Bond III immunostainer | Leica | N/A |
| Zeiss Axioskop 2 MOT microscope | Zeiss | N/A |
| MacsQuant cytometer | Myltenyi Biotech | N/A |
| Seahorse XF96 extracellular flux analyzer | Agilent technologies | N/A |
| Mass spectrometer Orbitap Q exactive | Thermo Fischer Scientific | N/A |
| Dionex UltiMate® 3000 UHPLC | Dionex | N/A |
| IsoCor module in Galaxy | *galaxy.workflow4metabolomics.org* | N/A |
| TraceFinder | Thermo Fischer Scientific | N/A |
| UHPLC-HRMS | Thermo Fischer Scientific | N/A |
| Hyperion 3000 infrared microscope | Bruker | N/A |
| Fc Odyssey imaging system | Li-COR Biosciences | N/A |
| NanoDrop 2000 spectrophotometer | Thermo Fisher Scientific | N/A |

## Monocyte isolation

Peripheral blood was collected from healthy, anonymous human volunteers who gave their consent to the use of their blood for research studies (Blood Collection Center (EFS), Nantes, France). The blood collection and experiments were carried out in accordance with the principles set out in the WMA Declaration of Helsinki and of the Department of Health and Human Services Belmont Report and were approved by the Ethics Committee of the University Hospital of Angers (agreement CPDL-PLER-2021 038). Peripheral blood mononuclear cells (PBMC) were obtained by standard density-gradient centrifugation on lymphocyte separation medium (Eurobio, Courtaboeuf, France). CD14$^+$ monocytes were then isolated by positive magnetic cell sorting (Miltenyi Biotec, Bergisch Gladbach, Germany; purity was routine >98%). Each experiment was performed with monocytes isolated from different donors.

## Macrophage generation

Monocytes ($1 \times 10^6$ cells/mL) were differentiated into macrophages by culture in Upcell plates (Nunc, Roskilde, Denmark) at a density of $1 \times 10^6$ cells/mL in classic culture medium (CM): Gln$^{low}$ RPMI 1640 medium (Corning, Corning, NY) supplemented with 10% fetal calf serum (FCS) (Eurobio), 2 mM L-Gln, 0.1 mM non-essential amino acids, 10 mM HEPES, 100 U/mL penicillin, 100 µg/mL streptomycin (all from Lonza) and 1 mM sodium pyruvate (Corning). For prototypic day 5 M1 cells, monocytes were differentiated in CM in the presence of 50 ng/mL GM-CSF (Eurobio) with the addition of 50 ng/mL IFNγ (Immunotools, Friesoythe, Germany) on day 2. Prototypic day 5 M2 cells were generated in CM in the presence of 50 ng/mL M-CSF (Eurobio), with the addition of 50 ng/mL IL-4 (R&D Systems) on day 2. GM-CSF-Mφ and M-CSF-Mφ were generated from monocytes by culture for 5 days with GM-CSF or M-CSF, respectively. M1 cells are highly glycolytic. We therefore replaced half the medium with fresh CM containing the appropriate cytokines on day 2, to ensure that glucose remained available until day 5. In some experiments, monocytes were differentiated in GM-CSF or M-CSF with the addition of 50 ng/mL IL-1β (Miltenyi Biotec) or 50 ng/mL IL-10 (Immunotools, Friesoythe, Germany) on day 2. For glycogenesis experiments, macrophages were generated from monocytes as described above, but were washed, counted, and transferred to Glc$^{low}$ RPMI 1640 medium (Glc$^{low}$ medium ( < 0.1 g/L Glc); Corning) supplemented with 10% FCS and antibiotics on day 2. For glyconeogenesis experiments, macrophages were generated as described above, but with transfer, on day 2, to Glc- and Gln-free RPMI 1640 medium (Sartorius, Dourdan, France) supplemented with 10% FCS and antibiotics (Glc$^{low}$ Gln$^{low}$ medium), with or without supplementation with 2 g/L Glc (Sigma-Aldrich), 20 mM L-Gln (Lonza), 10 mM lactic acid (Sigma-Aldrich), 70 mM glycerol (Sigma-Aldrich), 100 mM L-alanine (Alfa Aesar, Paris, France) or 10 mM pyruvate (Corning). For the inhibition of glyconeogenesis, inhibitors were added on day 2, 15 min before the addition of the non-glucose substrate to the medium. The inhibitors used were α-cyano-4-hydroxycinnamic acid (CHC) (125 µM; Sigma-Aldrich) and GSK2837808A (15 µM; Tocris, Minneapolis, MN) for the lactic acid-specific pathway, CB-839 (200 nM; MedChemExpress, Monmouth Junction, NJ), GOT1i (20 µM; MedChemExpress), and NV-161 (100 µM; Oroboros Instruments MitoKit CII, Innsbruck, Austria) for the glutamine-specific pathway, PEPCKi (10 µM; Santa Cruz Biotechnology, Dallas, TX) for PCK2 inhibition and MB05032 (50 µM; MedChemExpress) for FBP1 inhibition.

## Macrophage stimulation

Day 5 M1 and M2 cells generated in CM were counted and cultured in either CM or Glc$^{low}$ medium in Upcell plates ($1 \times 10^6$ cells/mL) and were stimulated with 100 ng/mL LPS (from *Escherichia coli* serotype O111:B4, $30 \times 10^6$ endotoxin unit/mg (1.4% contaminant protein); Sigma-Aldrich). The supernatants and cells were collected separately and stored at −80 °C until use. For glycogenolysis and PPP inhibition experiments, cells were left untreated or were treated with the inhibitors CP-91149 (50 µM; Selleckchem, Houston, TX) or 6-aminonicotinamide (6AN) (20 µM; Cayman Chemical, Ann Arbor, MI) for 15 min before LPS stimulation (see Table 1 for details).

## Tumor-associated macrophages

Ovarian cancer ascites were obtained with written informed consent from the patients, in accordance with the requirements of Angers University Hospital ethics committee and the Declaration of Helsinki (agreement 2013/38). Ascites were collected aseptically, and mononuclear cells were isolated by standard density-gradient centrifugation followed by CD14-positive magnetic cell sorting; purity was routinely >95%. Cells were cultured for three days in Glc$^{low}$ Gln$^{low}$ medium supplemented with 50 ng/mL GM-CSF and 50 ng/mL IFNγ, in the presence or absence of 2 g/L Glc or 10 mM lactic acid. For glycogenolysis experiments, day 3 TAM were treated or not with 50 µM CP-91149 for 15 min before a 6 h stimulation with 100 ng/mL LPS.

## Glycogen detection and quantification

Intracellular glycogen was quantified in monocytes and day 5 macrophages using a colorimetric assay kit according to the manufacturer's instructions (Biovision, Milpitas, CA). For the PAS staining of glycogen, monocytes and day 5 macrophages were seeded at a density of $1 \times 10^6$ cells/well on glass coverslips coated with 100 µg/mL poly-L-lysine (Sigma-Aldrich). Cells were fixed with 3.7% PFA (Thermo Fischer Scientific) and then treated with 0.5% sodium periodate (Sigma-Aldrich) and subjected to PAS staining for 15 min (Thermo Fisher Scientific). After three washes with ethanol (Thermo Fisher Scientific) and histolemon (Carlo Erba, Val-de-Reuil, France), slides were mounted in Entellan mounting medium (Sigma-Aldrich). Images were acquired on a Leica DMR microscope (100× magnification) and on a Zeiss Axioskop 2 MOT microscope (20× magnification) and images were analyzed using the ImageJ software. Glycogen levels in monocytes and day 5 macrophages were also assessed by detection of the fluorescent D-glucose analog 2-[N-[7-nitrobenzene-2-oxa-1,3-diazol-4-yl]amino]-2-deoxyglucose (2-NBDG; Thermo Fisher Scientific). Cells were seeded in eight-well Lab-Tek chambered coverglass (Nunc) at a density of $2 \times 10^5$ cells/well and were incubated for 2 h in Glc-free and Gln-free medium supplemented with 10% FCS, 2 g/L 2-NBDG and 20 mM L-Gln. Cells were fixed with 4% PFA and counterstained with 1 µg/mL DAPI (Sigma-Aldrich) or Hoescht 33342 (Sigma-Aldrich). Images were acquired on a Leica SP8 confocal microscope (Leica Microsystems, Nanterre, France) and images were analyzed using the ImageJ software. The glycogen content of TAM was analyzed on 5 µm FFPE (Formalin-Fixed Paraffin-Embedded) sections of lung adenocarcinoma (Department of Pathology, Angers University Hospital) by PAS staining, after amylase treatment or not (Benchmark special stains platform; Ventana Medical Systems, Tucson, AZ). Slides were counterstained with hematoxylin. Mouse anti-human CD163 monoclonal antibody (clone 10D6; Leica Microsystems, Newcastle, UK) was used to determine TAM distribution on consecutive slides. Slides were analyzed using the Leica Bond III immunostainer (Leica Microsystems, Newcastle, UK).

## Glucose consumption

Glucose concentrations in cell culture supernatants were determined with a blood glucometer (Accu-Chek; Roche Diagnostics,

Meylan, France) according to the manufacturer's instructions. Results are expressed in mg/dL, as the amount of Glc in the medium minus the amount of Glc present in the culture supernatants of day 2 or day 5 macrophages.

## Cytokine quantification

Cytokines were quantified in the cell culture supernatants of LPS-stimulated macrophages and TAM at the time points indicated, with ELISA kits (Diaclone or R&D Systems).

## Phagocytosis assay

For fluorescent polystyrene microspheres (FluoSpheres™ Carboxylate-Modified Microspheres, Invitrogen), day 5 Mφ were incubated at 37 °C for 30 min with 1.0 μm-diameter at a cell/bead ratio of 1/30. Cells were then washed three times with ice-cold PBS. The rate of phagocytosis was determined by monitoring fluorescence (MacsQuant cytometer; Miltenyi Biotech) and was defined as the percentage of cells internalizing beads on an XF96 extracellular flux analyzer (Agilent Technologies, Santa Clara, CA). Cells incubated without beads were used as the negative control.

## Oxygen consumption rate

Day 5 Mφ were used to seed XF96 plates ($3 \times 10^4$ and $6 \times 10^4$ cells/well for M1 and M2 cells, respectively) by overnight incubation at 37 °C. Cells were then washed twice with Seahorse XF basal medium supplemented with 1 mM pyruvate, 2 mM L-Gln, and 11 mM D-Glc adjusted to pH 7.4 before oxygen consumption rate (OCR) analysis. OCR was determined with an XF96 extracellular flux analyzer (Agilent Technologies) on cells in basal conditions and after the addition of 50 μM CP-91149 and 100 ng/mL LPS for 60 min in CM or Glc$^{low}$ medium at 37 °C and in the absence of $CO_2$.

## Glycogen extraction and hydrolysis, and glucose analysis by GC-MS

For glycogenesis and glyconeogenesis experiments, day 2 GM-CSF- and M-CSF-Mφ were recovered, counted, and cultured in Glc$^{low}$ medium supplemented with 2 g/L $^{13}C_6$-Glc (Sigma-Aldrich), or in Glc$^{low}$ and Gln$^{low}$ medium supplemented with 10 mM $^{13}C_3$-lactic acid (Sigma-Aldrich) or 20 mM $^{13}C_5$-Gln (Cambridge Isotope Laboratories), with addition of the appropriate cytokines on day 2. Cells were harvested and counted at day 5, and glycogen was extracted by adding 2 mL 30% KOH to the cell pellets and boiling for 15 min. We then added 2 mL 95% ethanol and incubated the tubes overnight at 4 °C for glycogen precipitation. The precipitated glycogen was then sedimented by centrifugation at 1000×g for 10 min at 4 °C. Pellets were washed twice in 60% ethanol before resuspension in 1 M $H_2SO_4$ and boiling for 3 h to hydrolyze glycogen to glucose. The pH was adjusted to neutral, and the final volume was adjusted to 5 mL with distilled water. The samples were then freeze-dried. The glucose content was evaluated by exact mass gas chromatography/mass spectrometry (GC-MS) by measuring $^{13}C$ enrichment using the mass spectrometer Orbitrap Q Exactive (Thermo Fisher Scientific). $^{13}C$ enrichment data represent the $^{13}C$-atom excess whereby the contribution of natural $^{13}C$ abundance (determined in non-labeled samples) has been subtracted. We

subjected 10 mg of the lyophilized sample to extraction in 250 μL methanol:water (80/20; v/v), with 10 μM ribitol as an internal standard. The mixture was centrifuged (10 min at 10,000 × g), and 10 μL was poured into a vial with an insert and spin-dried at 39 °C. Samples were derivatized (automatically with a preparative robot) with 20 μL methoxyamine (20 mg/mL in pyridine; 90 min at 37 °C) and 30 μL N-methyl-N(trimethylsilyl)trifluoroacetamide (MSTFA) for 30 min at 37 °C. Before injection, 5 μL of a mixture of alkanes (14 alkanes from $C_9$ to $C_{36}$, 3 μg/μL; Connecticut n-Hydrocarbon Mix, Supelco; Sigma-Aldrich) was added in each sample for calculation of the retention index. Analyses were performed by injecting 1 μL in split-less mode at 230 °C (injector temperature) onto a TG-5 SILMS column (30 m × 0.25 mm × 0.25 μm; Thermo Fisher Scientific) in a Trace 1300 Series GC (Thermo Fisher Scientific). Helium at a constant flow rate of 1 mL/min was used as the carrier gas. After one minute at the initial GC oven temperature (70 °C), the temperature was increased to 325 °C at a rate of 15 °C.min$^{-1}$ and was then maintained at 325 °C for 4 min. MS analyses were performed with positive polarity in full MS scan mode with the following source settings: mass scan range 50-750 $m/z$, resolution 60,000, AGC target 1E6, MS transfer line 300 °C and filament delay 4.12 min. Ionization by electron impact (70 eV) was performed with at an ion source temperature of 250 °C. Amino acids were identified automatically by TraceFinder (Thermo Fisher Scientific) on the basis of retention time, main characteristic fragment ($m/z$ ion) and a confirmation fragment, with a maximum tolerance of 0.00007 Da.

## Metabolite analysis by LC-MS

*Chemicals and reagents.* Methanol (MeOH), water and formic acid (Optima LC/MS grade) were purchased from Thermo Fisher Scientific. The hexose, alpha-ketoglutarate, aspartate, glutamate, citrate, fumarate, lactate, malate, pyruvate, succinate and glutamine were obtained from Sigma-Aldrich.

*Fluxomics analysis.* Samples were randomly prepared as follows: day 2 GM-CSF-Mφ were switched to Glc$^{low}$ Gln$^{low}$ medium containing GM-CSF and IFNγ. On day 3, cells were treated or not with 100 μM NV-161 (Oroboros Instruments MitoKit CII, Innsbruck, Austria) and supplemented with 2 mM $^{13}C_5$-Gln (Sigma-Aldrich) for 30 min before analysis by UHPLC-HRMS (Ultra High-Pressure Liquid Chromatography-High Resolution Mass Spectrometry; Thermo Fisher Scientific). Briefly, after removal of the medium, the cell pellets (estimated at $2 \times 10^6$ cells) were rinsed twice with a 0.22% NaCl solution and extracted with a MeOH/H2O (80/20) one. After centrifugation (15 min at 5 °C, 12,000 × g), supernatants were evaporated, and samples were then reconstituted in an aqueous solution (2% MeOH) before injection. Internal quality controls (QCs, a pool of all the samples) were injected regularly across the sequence in order to assess the stability of the signal for all the 11 metabolites (CV < 5%, mass precision <8 ppm). Analysis was performed in negative mode on a Thermo Scientific Q Exactive mass spectrometer (Thermo Fisher Scientific, Bremen, Germany), equipped with a heated electrospray ionization source and coupled with a Dionex UltiMate® 3000 UHPLC (Dionex, Sunnyvale, CA) equipped with a Phenomenex Luna Omega Polar C18 (150 mm × 2.10 mm, 100 Å) UHPLC column. Ionization conditions, MS parameters and chromatography conditions were identical to those previously described and validated

(Belal et al, 2022; Bocca et al, 2018). A TraceFinder 4.1 processing method was designed to target only 11 metabolites and their isotopic species. Raw MS data were corrected for naturally-occurring isotopes and the purity of the $^{13}$C-Gln using IsoCor module in Galaxy (galaxy.workflow4metabolomics.org) and normalized by the DNA concentration of each biological sample (NanoDrop 2000 spectrophotometer; Thermo Fisher Scientific) (Silva LP et al, 2013).

## FTIR spectroscopic imaging

Five-micrometer-thick section of paraffin-embedded tumor were deposited on BaF2 windows. Spectral analysis was performed with a Bruker Hyperion 3000 infrared microscope coupled to a Vertex 70 spectrometer using a $64 \times 64$ focal plane array (FPA) detectorA 15× Cassegrain objective (Numerical Aperture 0.4) was used for all acquisition. Mid-infrared spectra were recorded at a resolution of $4\,cm^{-1}$ (spectral region 900–2000 $cm^{-1}$), with 32 accumulations in transmission mode. Background spectra were also recorded with the same specifications. Post-processing was done with a lab-made script written in Matlab and included baseline correction, digital paraffin subtraction using signal intensity at 1465 $cm^{-1}$, vector normalization to the amide I band and denoising using the Savitzky–Golay algorithm with a degree of 2 and a span length of 9. A quality control of each spectrum was performed by calculating the signal-to-noise ratio (SNR) over the spectral range 1850–2000 $cm^{-1}$ free of biological signal. Second derivative were computed over the spectral range 1034–1185 $cm^{-1}$ and used as a loading vector for curve fitting. Spectral curve fitting quality was assessed by a root mean square error value set at 1% of the total spectral interval area. Intensities of sub-bands located at ~1127 $cm^{-1}$ and ~1152 $cm^{-1}$ were normalized to the intensity of the ~1655 $cm^{-1}$ amide I band to estimate the lactic acid and glycogen contents, respectively, as reported (Hackett et al, 2016).

## Gene silencing

*Inhibition of glyconeogenesis in M1 cells.* Day 2 GM-CSF-Mφ were cultured at a density of $1 \times 10^6$/ml in Glc$^{low}$ and Gln$^{low}$ medium supplemented with 2.5% FSC, 50 ng/mL GM-CSF and 50 ng/mL IFNγ (R&D Systems). After 30 min, 1 μM of Accell SMART Pool siRNA (Dharmacon, Lafayette, CO) targeting PCK2 or FBP1 or a control siRNA was added. On day 5, the medium was supplemented with 20 mM Gln and 7.5% FSC. After 48 h (day 7), glycogen was quantified, and RT-qPCR analysis was performed.

*Inhibition of the PPP and glycogenolysis by M1 cells.* Day 2 GM-CSF-Mφ, at a density of $1 \times 10^6$/ml in CM supplemented with 2.5% FSC and containing IFNγ were incubated for 30 min and 1 μM Accell SMART Pool siRNA (Dharmacon) targeting PYGL, PYGB, or G6PD, or a negative control siRNA was then added. On day 5, the cell culture supernatants were refreshed and supplemented with medium containing 10% FSC and the cells were then left unstimulated or were stimulated with 100 ng/mL LPS.

## mRNA quantification

Total RNA was extracted with the NucleoSpin RNA reagent kit (Macherey-Nagel, Allenton, PA), and reverse-transcribed with the Superscript II reverse transcriptase (Invitrogen). Quantitative PCR (qPCR) was then performed to determine the amounts of mRNA for the proteins indicated. Gene expression was calculated using the $2^{-\Delta\Delta CT}$ method, using RPS18, EF1α, and TBP as reference genes for normalization; results are expressed as relative mRNA levels. Primer sequences are available upon request.

## Western blotting

Total protein (20 μg) was separated by SDS-PAGE electrophoresis and transferred onto nitrocellulose membranes. Immunoblotting was performed by overnight incubation at 4 °C with antibodies directed against PCK2 (Cell Signaling Technologies, Danvers, MA) FBP1 (Cell Signaling Technologies), GYS1 (Cell Signaling Technologies), PYGL (Abcam, Cambridge, UK), HSC70 (Santa Cruz Biotechnology) and β-actin (Proteintech, Planegg-Martinsried, Germany). The membranes were washed and immunodetection was performed with appropriate IRdye 680 and 800 secondary antibodies. The signals were acquired with the Fc Odyssey imaging system (Li-COR Biosciences, Lincoln, NE). Data were analyzed with Wave Pro 10.1.0 software.

## Statistical analysis

Statistical analyses were performed with PRISM software version 9.5.0 (www.graphpad.com). Data were tested for normally distribution with the Shapiro–wilk normality test. Student's unpaired two-tailed $t$ tests were used for comparisons of two sets of conditions and Welch ANOVA followed by Dunnett's post hoc test was used for comparisons of more than two sets of conditions. We considered $P$ values $< 0.05$ to be statistically significant, and $n$ indicates the number of independent biological replicates.

# Data availability

This study does not include data deposited in external repositories.

The source data of this paper are collected in the following database record: biostudies:S-SCDT-10_1038-S44319-024-00278-4.

# Peer review information

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

## Acknowledgements

The authors acknowledge members of the qPCR (PACeM) and microscopy (SCIAM) facilities of the University of Angers for expert technical assistance. The authors thank members of the HiMolA facility (Inserm unit 1229, Angers, France) for assistance with FTIR imaging. This work was performed in the context of the LabEX IGO program (National Research Agency via the Investment for the Future program ANR-11-LABX-0016-01) and of the research program 3I-Impact (supported by the University of Angers and Angers University Hospital).

## Author contributions

**Najia Jeroundi**: Investigation; Methodology; Writing—original draft; Writing—review and editing. **Charlotte Roy**: Investigation; Methodology; Writing—original draft; Writing—review and editing. **Laetitia Basset**: Investigation; Methodology; Writing—original draft. **Pascale Pignon**: Investigation. **Laurence Preisser**: Investigation. **Simon Blanchard**: Formal analysis; Visualization. **Cinzia Bocca**: Formal analysis; Investigation. **Cyril Abadie**: Investigation. **Julie Lalande**: Investigation. **Naïg Gueguen**: Visualization; Methodology. **Guillaume Mabilleau**: Investigation; Methodology. **Guy Lenaers**: Methodology. **Aurélie Moreau**: Conceptualization; Funding acquisition. **Marie-Christine Copin**: Investigation. **Guillaume Tcherkez**: Formal analysis; Methodology. **Yves Delneste**: Supervision; Funding acquisition. **Dominique Couez**: Conceptualization; Supervision; Writing—original draft; Writing—review and editing. **Pascale Jeannin**: Conceptualization; Supervision; Funding acquisition; Visualization; Writing—original draft; Writing—review and editing.

Source data underlying figure panels in this paper may have individual authorship assigned. Where available, figure panel/source data authorship is listed in the following database record: biostudies:S-SCDT-10_1038-S44319-024-00278-4.

## Disclosure and competing interests statement

The authors declare no competing interests.

# Expanded View Figures

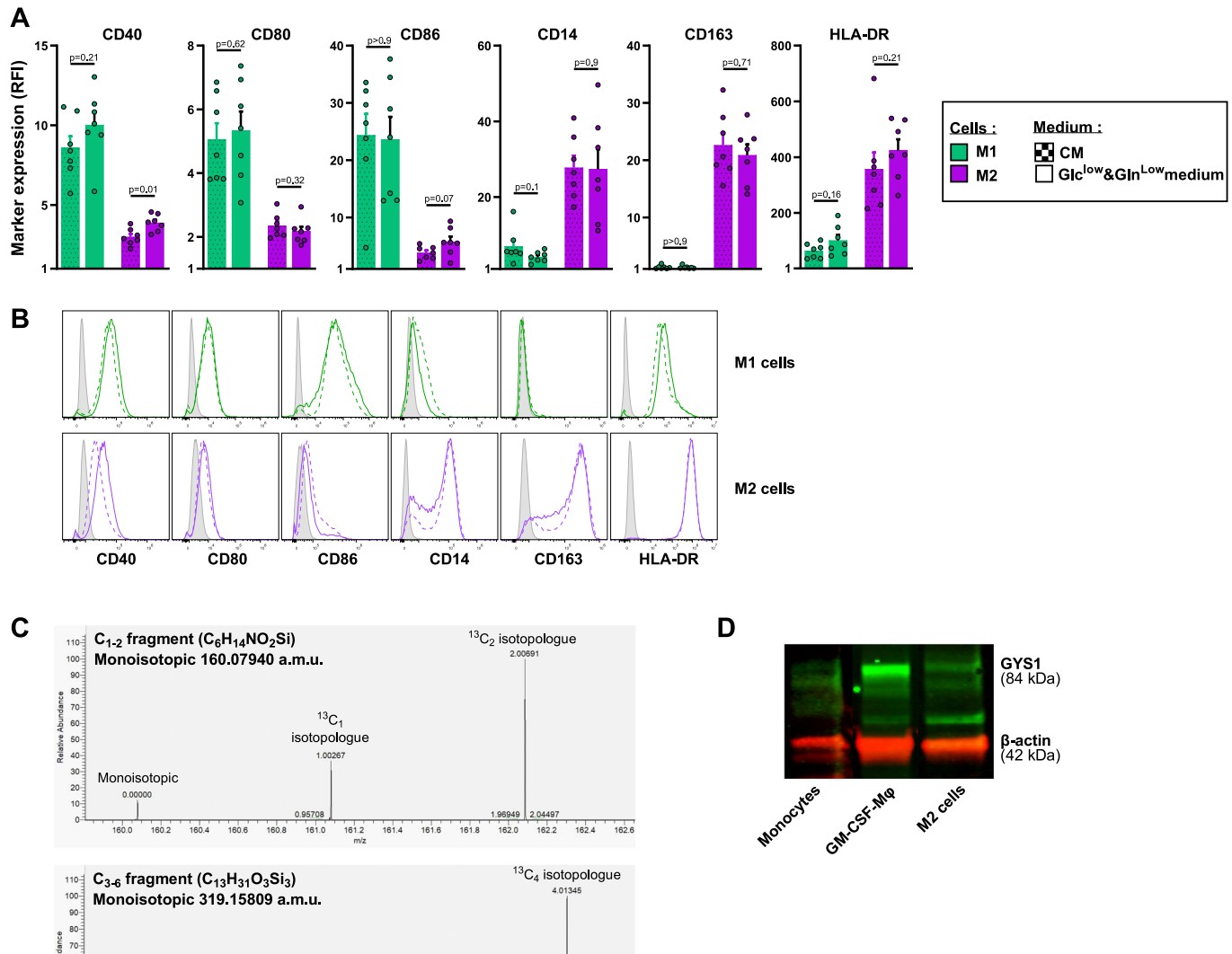

**Figure EV1. GM-CSF or M-CSF plus IL-4 triggers human macrophage glycogenesis.**

(A, B) Phenotypic characterization of M1 and M2 cells. M1 and M2 cells were generated either in conventional medium (CM) for 5 days or in CM for the first two days followed by 3 days culture in $Glc^{low}$ $Gln^{low}$ medium up to day 5. (A) The M1 versus M2 phenotype was analyzed by flow cytometry using BUV395-labeled anti-CD14, BV421-labeled anti-CD163, BV510-labeled anti-CD40, FITC-labeled anti-CD80, PE-cy7-labeled anti-CD86 and APC-labeled anti-HLA-DR mAbs (all from BD Biosciences). Results are expressed as relative fluorescence intensity (RFI) ($n = 7$). (B) Representative flow cytometry histograms. Gray histograms represent isotype controls, dashed lines and solid lines represent cells generated in CM and $Glc^{low}$ $Gln^{low}$ medium, respectively. (C) Isotopic patterns in Glc from glycogen show the prevalence of fully labeled isotopologues. Two typical fragments of 1-methoxy-penta-trimethylsilyl glucose (Glc 1MeOX, 5TMS) glucose are shown: (upper panel) isotopic pattern of the fragment at $m/z$ 160.0794 a.m.u. corresponding to C-atom positions C1-C2, (lower panel) isotopic pattern of the fragment at m/z 319.15809 a.m.u. corresponding to C-atom positions C3 to C6. Peaks are labeled with the mass difference with respect to the monoisotopic ($^{12}$C) form. Note that due to the $^{13}$C enrichment, Si isotopologues (29Si and 30Si at natural abundance) of the glucose derivative are small and thus not visible on this scale. (D) GYS1 expression was analyzed by western blotting in monocytes, day 5 GM-CSF-Mφ, and M2 cells; actin was used as loading control (representative of one out of 3). Values are represented as the mean ± SEM, each dot represents a different donor. Statistical significance was determined by two-tailed unpaired Welch t test (A). Source data are available online for this figure.

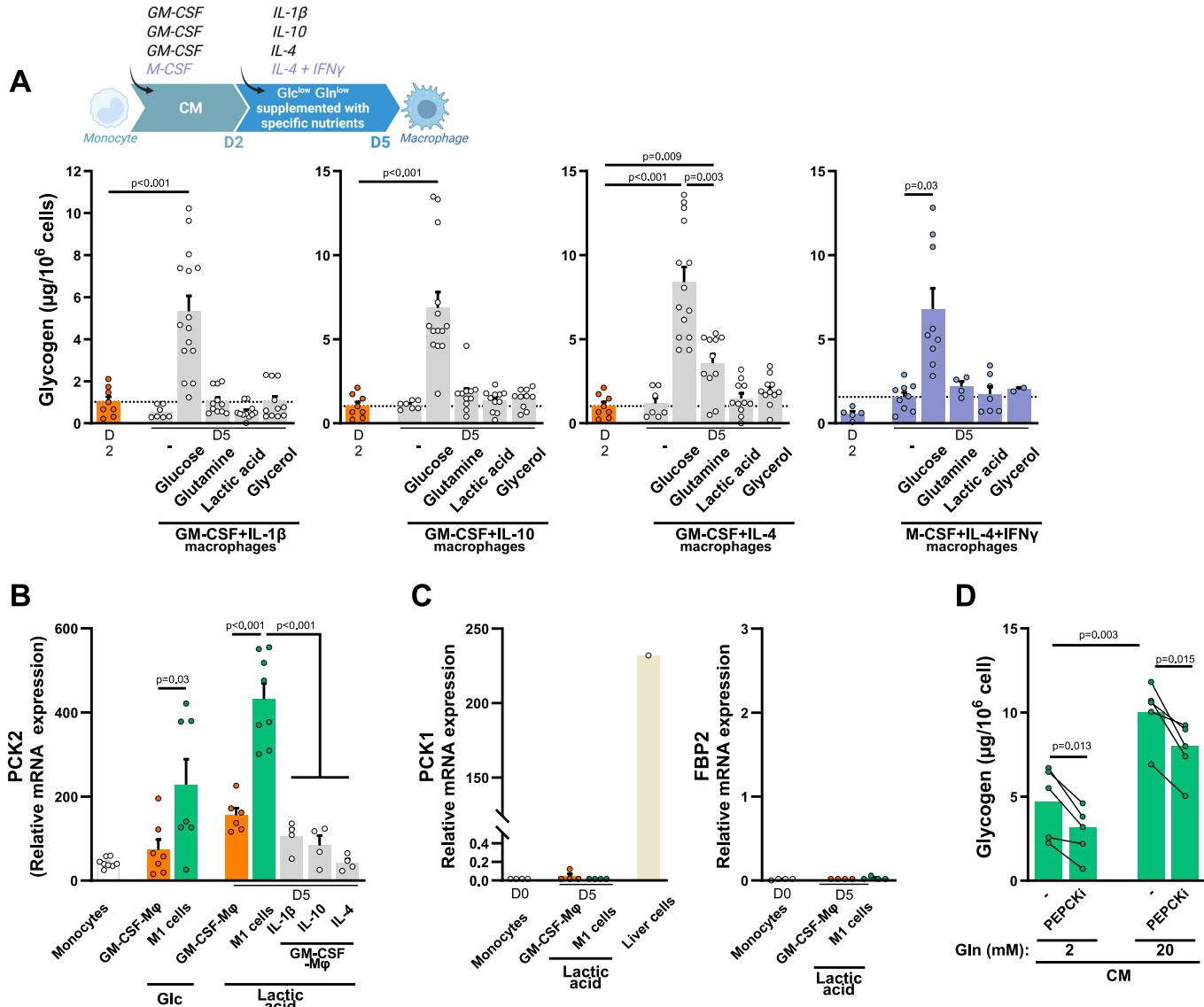

Figure EV2. The combination of GM-CSF plus IFNγ promotes human macrophage glyconeogenesis.

(A–C) Impact of cytokines on Mφ glyconeogenesis. GM-CSF-Mφ and M-CSF-Mφ were switched on day 2 to Glc^low Gln^low medium supplemented or not with the specific nutrients Glc, Gln, lactic acid or glycerol. The timelines summarize the experimental procedures for macrophage generation. (A) IL-1β, IL-10, IL-4, or IL-4 plus IFNγ were added or not during the differentiation and glycogen was quantified in day 5 Mφ (n = 4–15). (B) IL-1β, IL-10, IL-4 were added or not during the differentiation of GM-CSF-Mφ and PCK2 expression was determined by RT-qPCR (n = 4–8). (C) PCK1 and FBP2 expression was determined by RT-qPCR in monocytes, GM-CSF-Mφ and M1 cells. Liver cells were used as a positive control for PCK1 expression (n = 4). (D) M1 cells rely on glycogenesis and glyconeogenesis to synthesize glycogen. Glycogen content was determined in M1 cells cultured for 5 days in CM containing either 2 or 20 mM Gln, in the absence or presence of 10 µM PEPCKi (n = 5). Values are represented as the mean ± SEM, each dot represents a different donor. Statistical significance was determined by Welch's ANOVA test followed by Dunnett's multiple comparison post hoc test (A, B) or two-tailed unpaired Welch t test (C) or by paired t test (D). *P < 0.01, **P < 0.005, ***P < 0.001, ****P < 0.0001. Source data are available online for this figure.

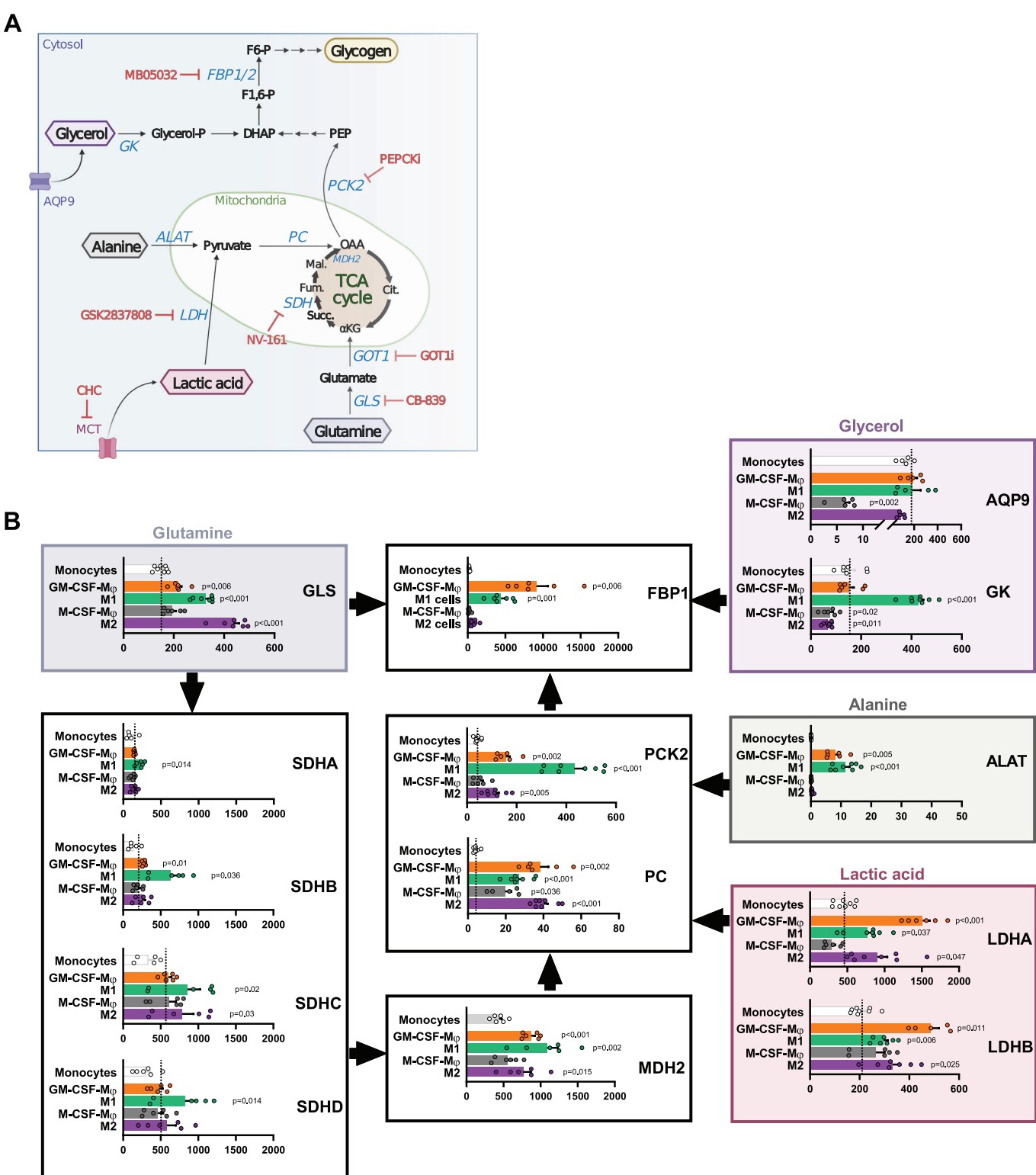

**Figure EV3. Glyconeogenesis pathways.**

(A) Main metabolites and enzymatic pathways involved in glyconeogenesis. Specific inhibitors used in this study are in red. (B) The expression of mRNA encoding enzymes involved in M1 glyconeogenesis (GLS, SDHA-D, FBP1, PCK2, PC, MDH2, AQP9, GK, ALAT and LDHA-B) was determined by RT-qPCR in monocytes and day 5 Mφ (n = 6–8). Values are represented as the mean ± SEM; each dot represents a different donor. Statistical significance was determined by Welch's ANOVA test followed by Dunnett's multiple comparison post hoc test. The P values represent the comparison between monocytes and macrophages. *P < 0.01, **P < 0.005, ***P < 0.001, ****P < 0.0001. Source data are available online for this figure.

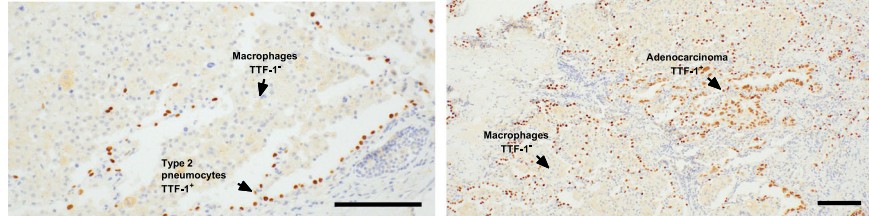

**Figure EV4.    Tumor-associated macrophages store glycogen and are capable of glyconeogenesis.**

Immunohistochemical staining of TTF-1 in lung adenocarcinoma. Scale bar, 100 μm. Source data are available online for this figure.

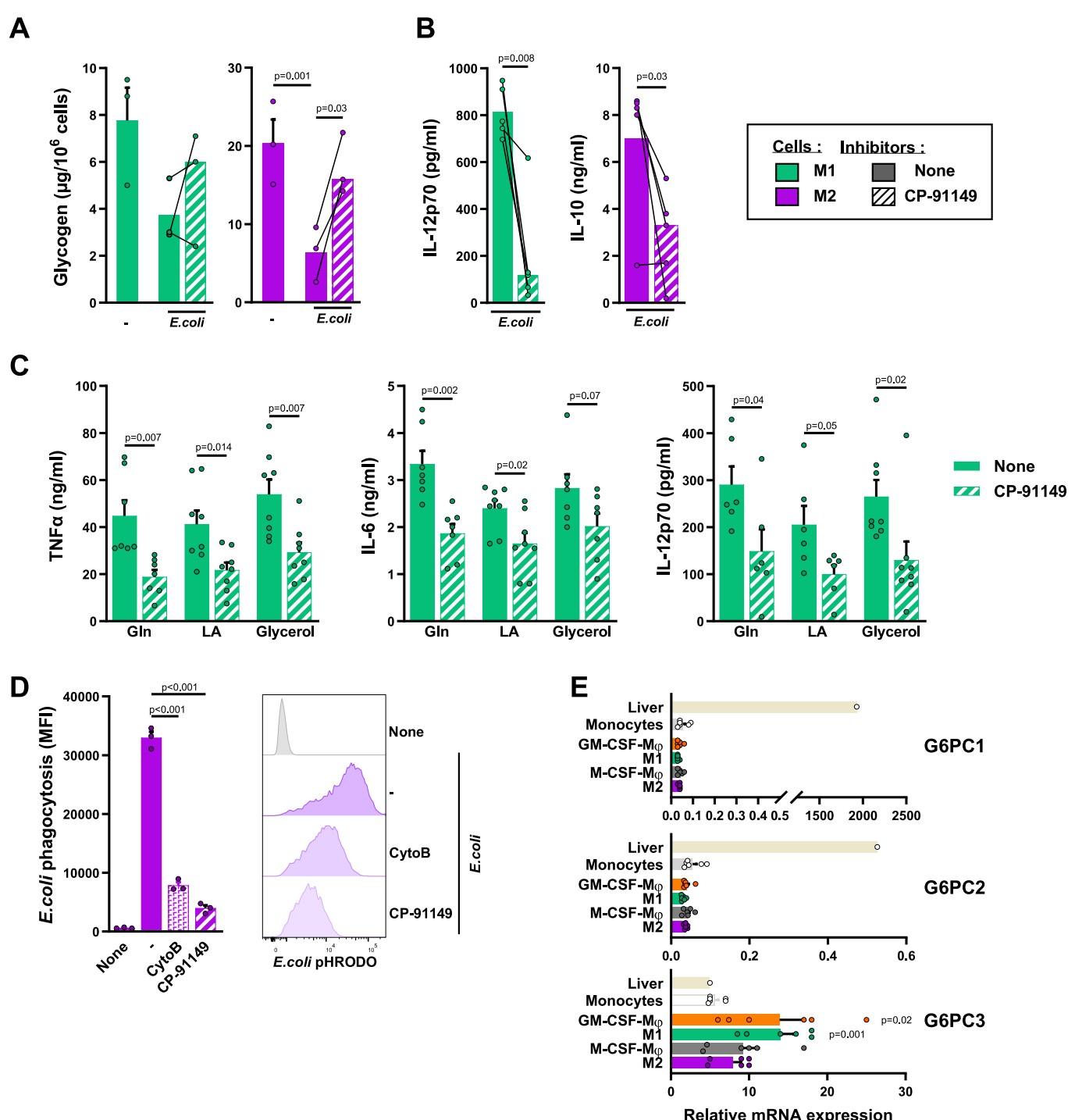

**Figure EV5.  Glycogen sustains M1 and M2 cell functions.**

(A, B) Day 5 M1 and M2 cells (generated in conventional medium or CM) were treated or not for 15 min with 50 μM CP-91149 before stimulation with 100 ng/mL LPS or with *E. coli* at a multiplicity of infection of 10. (A) Glycogen was quantified after 24 h activation ($n = 3$). (B) Cytokines were quantified in the supernatants by ELISA after 24 h (IL-12p70) or 6 h (IL-10) stimulation ($n = 5$). (C) GM-CSF-Mφ were switched on day 2 to Glc^low Gln^low medium supplemented or not with Gln, lactic acid or glycerol and treated or not with CP-91149 15 min before LPS stimulation. TNFα and IL-6 (2 h), or IL-12p70 (24 h) were quantified by ELISA in stimulated day 5 M1 cells culture supernatants ($n = 6$–8). (D) M2 cells were pretreated with 50 μM CP-91149 or 10 μM cytochalasin B (CytoB) for 15 min before addition of 0.2 mg/mL pHrodo-conjugated *E. coli* BioParticles for 2 h at 37 °C. Fluorescence was determined by flow cytometry and expressed in MFI values ($n = 3$). (E) G6PC1-3 mRNA expression was determined by RT-qPCR in monocytes and day 5 Mφ ($n = 6$). Liver cells were used as a positive control. Values are represented as the mean ± SEM, each dot represents a different donor. Statistical significance was determined by paired *t* test (A, B) or by two-tailed unpaired Welch *t* test (C, D). *$P < 0.01$, **$P < 0.005$, ***$P < 0.001$, ****$P < 0.0001$. Source data are available online for this figure.

