## [Peer Review File · EMBO Reports]

Glycogenesis and glyconeogenesis from glutamine, lactate and glycerol support human macrophage functions.

Najia JEROUNDI, Charlotte Roy, Laetitia Basset, Pascale Pignon, Laurence Preisser, Simon Blanchard, Cinzia Bocca, Cyril Abadie, Julie Lalande, Naig Gueguen, Guillaume Mabillean, guy lenaers, Aurélie Moreau, Marie-Christine Copin, Guillaume Tcherkez, Yves Delneste, Dominique Couez, and Pascale Jeannin

Corresponding author(s): Pascale Jeannin (pascale.jeannin@univ-angers.fr)

Review Timeline:

Submission Date:	17th Oct 23
Editorial Decision:	15th Dec 23
Revision Received:	28th Jun 24
Editorial Decision:	15th Aug 24
Revision Received:	3rd Sep 24
Accepted:	15th Sep 24

Editor: Deniz Senyilmaz Tiebe

Transaction Report:

Dear Prof. Jeannin,

Thank you for the submission of your research manuscript to our journal, which was now seen by three referees, whose reports are copied below.

My apologies for this unusual delay in getting back to you. It took longer than anticipated to receive the full set of referee reports.

Referees express interest in the proposed role of glycogen metabolism in regulation of macrophage functions. However, they also raise concerns that need to be addressed to consider publication here.

Given these positive recommendations, we would like to invite you to submit a revised manuscript. Please revise your manuscript with the understanding that the referee concerns (as in their reports) must be fully addressed and their suggestions taken on board. Please address all referee concerns in a complete point-by-point response. Acceptance of the manuscript will depend on a positive outcome of a second round of review. It is EMBO reports policy to allow a single round of major experimental revision only and acceptance or rejection of the manuscript will therefore depend on the completeness of your responses included in the next, final version of the manuscript.

We realize that it is difficult to revise to a specific deadline. In the interest of protecting the conceptual advance provided by the work, we recommend a revision within 3 months. Please discuss the revision progress ahead of this time with me if you require more time to complete the revisions, or if you have questions or comments regarding the revision (also by video chat).

1. A data availability section providing access to data deposited in public databases is missing (where applicable).
2. Your manuscript contains statistics and error bars based on $n=2$. Please use scatter plots in these cases.

You can submit the revision either as a Scientific Report or as a Research Article. For Scientific Reports, the revised manuscript can contain up to 5 main figures and 5 Expanded View figures, and it should not exceed 27000 characters. If the revision leads to a manuscript with more than 5 main figures it will be published as a Research Article. In this case the Results and Discussion section should be separate. If a Scientific Report is submitted, these sections have to be combined. This will help to shorten the manuscript text by eliminating some redundancy that is inevitable when discussing the same experiments twice. In either case, all materials and methods should be included in the main manuscript file.

4) a .docx formatted letter INCLUDING the reviewers' reports and your detailed point-by-point responses to their comments. As part of the EMBO publication's Transparent Editorial Process, EMBO reports publishes online a Review Process File (RPF) to accompany accepted manuscripts. This File will be published in conjunction with your paper and will include the referee reports, your point-by-point response and all pertinent correspondence relating to the manuscript. <https://www.embopress.org/page/journal/14693178/authorguide#transparentprocess>

5) a complete author checklist, which you can download from our author guidelines <https://www.embopress.org/page/journal/14693178/authorguide>. Please insert information in the checklist that is also reflected in the manuscript. The completed author checklist will also be part of the RPF.

6) Please note that all corresponding authors are required to supply an ORCID ID for their name upon submission of a revised manuscript (<<https://orcid.org/>>). Please find instructions on how to link your ORCID ID to your account in our manuscript tracking system in our Author guidelines <<https://www.embopress.org/page/journal/14693178/authorguide#authorshipguidelines>>

7) Before submitting your revision, primary datasets produced in this study need to be deposited in an appropriate public database (see <https://www.embopress.org/page/journal/14693178/authorguide#datadeposition>). Please remember to provide a reviewer password if the datasets are not yet public. The accession numbers and database should be listed in a formal "Data Availability" section placed after Materials & Method (see also <https://www.embopress.org/page/journal/14693178/authorguide#datadeposition>). Please note that the Data Availability Section is restricted to new primary data that are part of this study. * Note - All links should resolve to a page where the data can be accessed. *
If your study has not produced novel datasets, please mention this fact in the Data Availability Section.

Additional information on source data and instruction on how to label the files are available:
<https://www.embopress.org/page/journal/14693178/authorguide#sourcedata>

9) Our journal encourages inclusion of *data citations in the reference list* to directly cite datasets that were re-used and obtained from public databases. Data citations in the article text are distinct from normal bibliographical citations and should directly link to the database records from which the data can be accessed. In the main text, data citations are formatted as follows: "Data ref: Smith et al, 2001" or "Data ref: NCBI Sequence Read Archive PRJNA342805, 2017". In the Reference list, data citations must be labeled with "[DATASET]". A data reference must provide the database name, accession number/identifiers and a resolvable link to the landing page from which the data can be accessed at the end of the reference. Further instructions are available at <http://www.embopress.org/page/journal/14693178/authorguide#referencesformat>

- the name of the statistical test used to generate error bars and P values,
- the number (n) of independent experiments (please specify technical or biological replicates) underlying each data point,
- the nature of the bars and error bars (s.d., s.e.m.),
- If the data are obtained from n Program fragment delivered error `Can't locate object method "less" via package "than" (perhaps you forgot to load "than"?) at //ejpvfs23/sites23b/embor_www/letters/embor_decision_revise_and_review.txt line 56.' 2, use scatter blots showing the individual data points.

12) Please also note our reference format:

I look forward to seeing a revised version of your manuscript when it is ready. Please let me know if you have questions or comments regarding the revision.

Kind regards,

Deniz Senyilmaz Tiebe

Deniz Senyilmaz Tiebe, PhD
Scientific Editor
EMBO Reports

Referee #1:

Jeroundi and colleagues aim to shed light into the ability of human macrophages to metabolize glycogen. In their research, they differentiated macrophages into M1 and M2 subtypes by using well accepted treatments to recapitulate potential physiological stages of macrophages in tissues upon infection or inflammatory challenges. More interestingly, authors probed TAM cells from human samples. This work adds to what we know probing mouse cells, and therefore it is of interest. The experimental approach is sound and within the state-of-the-art. My comments mostly refer to data availability and power calculations. Additional experiments are suggested to strengthen the functional biology part of the study. In this section, LPS is used. However it does not truly represent an in vivo scenario in the case of an infection. Additionally, unless otherwise stated, the quality of the commercially available LPS is low and it is full of lipoproteins and other contaminants making difficult to claim that the results are indeed LPS-induced. Please note my comments for the authors' consideration. I hope authors will find them valuable to strengthen this elegant study.

1. Details on samples. It is not clear in the study (neither in the methods nor in the figure legends), the number of different human donors tested (and whether they were or not from the same sex or this was known). Also it is not apparent in the data set whether authors tested technical replicates or not. All this information should be specified clearly in the figure legends. A point open for discussion is how many different human donors should be tested. This reviewer may suggest that 5 is appropriate. It is absolutely essential to probe different donors considering the interindividual variability.
2. Microscopy analysis. In Fig 1c, authors need to indicate how many cells were considered (specify the number of cells analyzed per each of the experiments), and whether the phenotype was found in every single cell counted. This information should be added as supplementary figure. Fig 4c, again it is not clear how many different human subjects were analyzed or whether findings are representative of images from the same subject (this is not appropriate). As noted before, authors need to include some quantitative information beyond the images.
3. To control the type of cells authors have been testing, authors need to present in fig 1 (or as supplementary material), the analysis of some human macrophage M1 and M2 markers.
4. Functional biology experiments. The use of LPS has its value (though please refer to my comments of the purity of the LPS tested). I urge the authors to challenge cells with live bacteria. This will allow them to assess whether the initial interaction (attachment) is affected (or not), and the effect on phagocytosis. Alternatively, authors may want to test *E. coli* pHRODO and *Staph aureus* pHRODO. Latex beads are used as a model but there is literature showing that differences exist when probing live bacteria.

Referee #2:

In this study, Jeroundi et al investigated glycogen metabolism in human macrophages, and reported variations in glycogen level and synthesis between "M1" vs. "M2" macrophage subtypes. M1 cells are capable of generating glycogen using gluconeogenesis substrates under low glucose condition. LPS stimulation reduced glycogen content in IFN γ or IL4 stimulated macrophages and inhibition of glycogenolysis reduced cytokine release suggesting glycogen is a fuel source for these essential macrophage functions. Overall, the strengths of this study include that it addresses a topic of broad scientific interest, providing significant amount of experimental data pertaining to stimulation-dependent glycogen metabolism in human macrophages, and the findings have health relevance for metabolic flexibility in nutrient depleted environments.

Major comments:

1. In the experiments where macrophages are differentiated in low glucose media, measurements of markers are needed to make sure the cells are differentiated, and in the case of M1 or M2 cells, they are indeed "M1" or "M2". The difference in glycogen metabolism could be due to cells not differentiating or polarizing.
2. Figure 1g suggested in GM-CSF macrophages the glycogen synthesis rate was 0.4ug glycogen/million cells/day, so after 5 days that would be 2ug, but the level in 1a is ~8ug, what is the explanation?
3. More surprisingly, why is the glycogen synthesis flux from lactate and glutamine, particularly lactate, so small (Fig 2b), but glycogen storage level in glutamine or lactate supplemented cells almost as large as glucose supplemented?
4. Important statistical comparisons were missing in the following figures
 - a. Fig.1d Comparison between GM-CSF vs. GM-CSF+IL4
 - b. Fig.6e,f Comparison between without CP-91149 vs. without
5. Figure 3e is hard to interpret. It is better to present the data as labeling fraction. Also it is surprising that SDH inhibitor did not cause a significant decrease in 4-labeled fumarate
6. The results showed some evidence that TAMs have the capacity to synthesize glycogen when stimulated, however, it is unclear if they do so to a meaningful extent in tumor. The physiological relevance of this part is unclear.
7. The support of the argument that glycogenolysis "fuel PPP, enabling human M1 and M2 cells to perform their functions" is very correlative-It was demonstrated that glycogenolysis inhibition decreases macrophage functions and that PPP inhibition decreases macrophage functions, but did not show direct evidence that glycogenolysis fuel PPP.
8. Overall, the paper showed that macrophages can generate glycogen from gluconeogenesis in low glucose condition, and glycogen break down can be important for macrophage functions. However, the physiological situation is unclear. In the body, when will macrophage generate glycogen? When would they break it down? At the site of infection, where glucose levels may be low and cells are exposed to stimulation like LPS, would macrophages mainly synthesize or break down glycogen? Or the glycogen is synthesized from glucose in glucose replete conditions where macrophage differentiate, then broken down in low-glucose infection site?

Minor:

1. What timepoint was the data in Fig. 1c stained? This information should be included.
2. In Figure 1b the authors claim 'IL-4 triggered a time-dependent accumulation of glycogen in M-CSF-M ϕ , this effect peaking after three days of incubation.' The data suggests glycogen is still accumulating throughout day 5, not reaching a peak. At day 3 IL4 appears to be significantly different from M-CSF alone.
3. Figure citation at the end of TAM results "PAS staining abolished by α -amylase (which cleaves the α 1 \rightarrow 4 glucoside bonds in glycogen) was detected, confirming the presence of glycogen in CD163-positive cells (Fig. S4c.)" Should it be Fig.4c?

Referee #3:

Here Jeroundi et al. examine the use of glycogen as a fuel system in human macrophages. Using in vitro derived cells with GM-CSF or M-CSF, and in some cases IFN γ or IL4, they show that macrophages can accumulate glycogen and that this glycogen store can be used to support inflammatory cytokine production, even when glucose is present. Overall the in vitro data are convincing and well presented as concerning the blood derived macrophage populations. The paper uses inhibitors, and siRNA to test some of the targets suggested in the glycogen pathway etc. The data involving tumor-derived macrophages (TAM), are less convincing.

There are just two substantial concerns regarding the in vitro-derived cells. First, is the apparent lack of effect of IFN shown in figure 1a. Despite the fact that IFN does little to alter the glycogen accumulation several other figures use this idea of M1 cells. I understand the distinction here is that the IFN apparently permits the use of the alternative fuels for glycogen whereas the GM-CSF cells are limited to the use of glucose, but the way the authors describe and refer to M1 is confusing. Second are the tracing data meant to show incorporation of ^{13}C into glycogen. Controls are lacking here, leaving us with the idea that because it is detectable, this is a relevant process. These levels of ^{13}C incorporation need to be compared with some condition that helps us understand the level that would be considered "positive" by the authors.

The tumor-derived work is less convincing. The supplementation or treatment of cells collected from ascites is not particularly helpful without some integration of the metabolic niche of the peritoneal space. Is glucose limited there? Might the fuel sources be different there? In addition, the PAS and PAS + amylase staining is not convincing. There is no counter staining for CD163 so it is unclear what cells are PAS positive.

More minor comments include:

The authors suggest the lack of congruence with previous work comes from the shorter exposure to IL4 in the published studies. This could easily be tested here and should be.

Figure 1g is not very helpful as there is nothing to compare it to. Are these values greater than would be seen for glutamine or some other fuel? What are the values in control cells?

Similarly, figure 2b is difficult to interpret. It suggests that glutamine but not lactate is used but 2a shows both support glycogen accumulation. At what point is this considered positive?

Figure 4b- how are the CD163+ cells detected in these sections? There is no staining for this marker. Sort some TAM for glycogen?

The use of experimental graphics is excessive. These should be used only for particularly complex experiments or to show the sites of the action of the inhibitors.

The levels of gene expression and protein expression do not seem to match the levels of glycogen accumulation. Could the authors comment/explain.

Responses to Reviewer 1.

Jeroundi and colleagues aim to shed light into the ability of human macrophages to metabolize glycogen. In their research, they differentiated macrophages into M1 and M2 subtypes by using well accepted treatments to recapitulate potential physiological stages of macrophages in tissues upon infection or inflammatory challenges. More interestingly, authors probed TAM cells from human samples. This work adds to what we know probing mouse cells, and therefore it is of interest. The experimental approach is sounded and within the state-of-the-art. My comments mostly refer to data availability and power calculations. Additional experiments are suggested to strengthen the functional biology part of the study. In this section, LPS is used. However, it does not truly represent an in vivo scenario in the case of an infection. Additionally, unless otherwise stated, the quality of the commercially available LPS is low and it is full of lipoproteins and other contaminants making difficult to claim that the results are indeed LPS-induced. Please note my comments for the authors' consideration. I hope authors will find them valuable to strength this elegant study.

1. Details on samples. It is not clear in the study (neither in the methods nor in the figure legends), the number of different human donors tested (and whether they were or not from the same sex or this was known). Also, it is not apparent in the data set whether authors tested technical replicates or not. All this information should be specified clearly in the figure legends. A point open for discussion is how many different human donors should be tested. This reviewer may suggest that 5 is appropriate. It is absolutely essential to probe different donors considering the inter-individual variability.

The authors acknowledge Reviewer 1 for this important point. Using human cells, we are faced with inter-individual variability, and, in each experiment, each point corresponds to different donor (i.e. does not represent technical replicates). The n value refers to the number of different donors used in each experiment.

As requested, we performed additional experiments to get results obtained with cells isolated from at least 5 different donors per experiment. In the revised manuscript:

- the Figures 1B,1C, 1E, 1F, 1G, 1I, 2A, 3B,3E, 4B and 5C and in Expanded View Figures EV2B&C and EV5E have been modified to integrate results from these additional experiments;
- the number of healthy donors used in each experiment are now mentioned in the Figure legends;
- The Material and Methods section (paragraph "Monocyte isolation"; page 20 line 460) has been modified as follows: "*Each experiment was performed with monocytes isolated from different donors*".

The blood collection center supplies research laboratories with blood components from healthy donors, who have given their written consent for a non-medical use of their blood. In France, blood donation is free and anonymous. Consequently, the only information we have is the absence of infectious agents (such as HBV, HCV and HIV); no other data, such as gender, age or ethnic origin, is provided to the research laboratory.

2. Microscopy analysis. In Fig 1c, authors need to indicate how many cells were considered (specify the number of cells analyzed per each of the experiments), and whether the phenotype was found in every single cell counted. This information should be added as supplementary figure.

As requested, additional PAS and 2-NBDG staining experiments were performed to determine the number of positive cells.

PAS staining and 2-NBDG fluorescence were analyzed by light and confocal fluorescence microscopy, respectively. Cells from 5 (for PAS) or 3 (for 2-NBDG) different donors were used; ≈ 300 cells per donor were analyzed and images were processed using Image J software.

These results show that most if not all cells (monocytes, M1 cells and M2 cells) are positive for PAS and 2-NBDG staining ($> 95\%$ positive cells; see figure R1-1).

By normalizing signal intensity to cell number, PAS staining and 2-NBDG fluorescence confirm that the signal intensity is higher in M1 and M2 cells than in monocytes (3.47 and 3.32 times higher for M1 and M2 with PAS staining and 3.64 and 4.14 times higher for M1 and M2 with 2-NBDG fluorescence) (see figure R1-2). These results are consistent with the results obtained by the quantification of intracellular glycogen by colorimetric assay showing higher glycogen levels in M1 and M2 cells than in monocytes (Figure.1A)

Staining	Monocytes	M1 cells	M2 cells
PAS	100%	100%	95%
2-NBDG + DAPI	100%	100%	100%

Figure R1-1. Upper panels, Monocytes and day 5 M1 and M2 cells were stained with PAS (one out of 5 different donors) or 2-NBDG (one out of 3 different donors) for 2 h; 50-350 cells by field are

presented. *Lower panels*, Table summarises the frequency of positive cells after PAS staining (Scale bar: 20 μm) or 2-NBDG internalization (Scale bar: 10 μm).

Figure R1-2: PAS staining and 2-NBDG uptake in M1 and M2 cells. 1C. *Left panels*: monocytes and day 5 M1 and M2 cells were analyzed by light microscopy (Scale bar: 50 μm) after PAS staining or by confocal microscopy after 2-NBDG internalization (Scale bar: 40 and 2 μm). For fluorescent microscopy analysis, nuclei were counterstained with DAPI. Results are representative of 1 out of 3 (for 2-NBDG) or 5 (for PAS) different donors. *Right panels*: relative intensity of PAS staining and 2-NBDG fluorescence (determined using the Image J software) in monocytes and in M1 and M2 cells (detailed in the Material and Methods section). 300 cells were analyzed per experimental condition; results were obtained from cells isolated from 3-5 different donors.

In the revised version of the manuscript, these new results are presented in Figure 1C, right panels; the text has been modified as follows:

Results section, paragraph “GM-CSF or M-CSF plus IL-4 triggers glycogenesis in human macrophages”, page 7, line 129: “*Image analysis showed that >95% of monocytes and day 5 M1 and M2 cells were positive for PAS staining or 2-NBDG internalization (Fig. 1C, left panels and data not shown) and confirmed that M1 and M2 cells contain higher levels of glycogen than monocytes (Fig. 1C, right panels).*”

Material and Methods section, paragraph “Glycogen detection and quantification”:

- page 23 line 520: “*Images were acquired on a Leica DMR microscope (100 \times magnification) and on a Zeiss Axioskop 2 MOT microscope (20 \times magnification) and images were analyzed using the Image J software*”.

- page 24 line 528: “*Images were acquired on a Leica SP8 confocal microscope (Leica Microsystems, Nanterre, France) and images were analyzed using the Image J software*”.

- **The legend of Figure 1C**, page 38, line 924: “*Left panels, monocytes and day 5 M1 and M2 cells were analyzed by light microscopy after PAS staining (upper panels; scale bar: 50 μm) or by confocal microscopy after 2 h 2-NBDG internalization (middle and lower panels; scale bar: 40 and 2 μm). For fluorescent microscopy analysis, nuclei were counterstained with DAPI. Results are representative of 1 out of 3 (for 2-NBDG) or 5 (for PAS) different donors. Right panels, relative intensity of PAS*

staining and 2-NBDG fluorescence in monocytes and day 5 M1 and M2 cells. 300 cells/donor were analyzed (n=3-5). Values are represented as the mean \pm SEM, each dot represents a different donor. Statistical significance was determined by two-tailed unpaired Welch t-test (C).

3. Fig 4c, it is not clear how many different human subjects were analyzed or whether findings are representative of images from the same subject (this is not appropriate). As noted before, authors need to include some quantitative information beyond the images.

The aim of this experiment was to illustrate the presence of glycogen in macrophages in situ, not to quantitatively assess glycogen in resident macrophages.

For this purpose, we have chosen lung tumors as they are heavily infiltrated by macrophages (Welsh TJ et al. 2005; Al-Shibli K et al. 2009; Mei J et al. 2016; Jackute J et al. 2018; reviewed in Conway EM et al. 2016) and characterized by a harsh microenvironment (rendering necessary a metabolic adaptation of macrophages to survive and function).

A qualitative analysis was performed on tissue sections from 4 different patients. Tissue analysis revealed a diversity of results, particularly in terms of macrophage numbers and localization, making quantitative analysis difficult if not impossible. As the primary objective was to illustrate qualitatively the presence of glycogen in macrophages, we present results obtained with the tumor tissue containing the highest number of macrophages.

Additional experiments have been performed to determine, by immunohistochemistry, CD163-positive cells in adjacent lung adenocarcinoma tissue sections. Results revealed a massive infiltration of CD163-positive and of PAS-positive cells in the same area (see Figure R1-3a); as a control, we observed that PAS-positive cells are TTF-1-negative and distinct from type 2 pneumocytes and TTF-1+ adenocarcinoma cells (see Figure R1-3b).

In addition, Fourier-transform infrared spectroscopy (FITR) confirmed the colocalization of lactic acid and glycogen in tissue areas enriched with PAS-positive macrophages. A significant linear correlation between glycogen and lactic acid levels ($R^2=0.87$, $p<0.0001$) suggests that lactic acid may serve as a substrate for glycogen synthesis in vivo (see Figure R1-3c); results are representative 1 out of 2 independent experiments. Given that GM-CSF and IFN- γ , along with elevated lactic acid, are present in solid tumors, this finding supports the concept that TAMs can utilize lactic acid to generate glycogen.

Figure R1-3: Tumor-associated macrophages in lung adenocarcinoma store glycogen. **a.** Immunohistochemical analysis of CD163 expression in lung adenocarcinoma. On the same tissue section, glycogen was detected by PAS staining, after treatment or not with amylase (Scale bar: 100 μm). **b.** Immunohistochemical staining of TTF-1 in adenocarcinoma cells (Scale bar: 100 μm). **c.** Lactic acid and glycogen quantification by FTIR spectroscopic imaging in lung adenocarcinoma. *Left panel*, unstained brightfield image; *right panels*, glycogen and lactic acid maps (Scale bar: 200 μm); linear correlation between lactic acid and glycogen contents in infiltrating lung adenocarcinoma was calculated (results are representative 1 out of 2 independent experiments).

In the revised version of the manuscript, these results have been added in Figures 4 and EV4; the Results and Methods sections and the corresponding Legends have been modified as follows:

Results section, paragraph "Tumor-associated macrophages store glycogen and are capable of glyconeogenesis" page 12, line 259: "...As expected, lung adenocarcinoma was massively infiltrated by CD163-positive macrophages (large cells with abundant, vacuolated cytoplasm and a round, regular nucleus containing a small nucleolus) (Fig. 4C). In the same tissue sections, PAS staining, which was abolished by α -amylase, revealed the presence of glycogen in TAM (Fig. 4C). Moreover CD163-positive cells were not stained with TTF1, a marker of adenocarcinoma cells and type 2 pneumocytes that do not contain glycogen (Fig. EV4). The presence of glycogen in macrophages was confirmed by FTIR spectroscopic imaging in serial sections. FTIR spectroscopy also revealed a colocalization of glycogen with lactic acid ($R^2 = 0.87$, $p < 0.0001$) (Fig. 4D). Given that GM-CSF and IFN- γ , along with elevated lactic acid, are present in solid tumors, this finding suggest that TAMs can use lactic acid in vivo to generate glycogen (Fig. 4D).

Methods section paragraph "Glycogen detection and quantification", page 24, line 533: ".../... Slides were counterstained with hematoxylin. Mouse anti-human CD163 monoclonal antibody (clone 10D6; Leica Microsystems, Newcastle, UK) was used to determine TAM distribution on consecutive slides. Slides were analyzed using the Leica Bond III immunostainer (Leica Microsystems)."

A new paragraph has been added to describe the "FTIR spectroscopic imaging", page 27, line 621: "Five-micrometer-thick section of the paraffin-embedded tumor were deposited on BaF2 windows. Spectral analysis was performed with a Bruker Hyperion 3000 infrared microscope coupled to a Vertex 70 spectrometer using a 64×64 focal plane array (FPA) detector. A $15\times$ Cassegrain objective (Numerical Aperture 0.4) was used for all acquisition. Mid-infrared spectra were recorded at a resolution of 4 cm^{-1} (spectral region $900\text{--}2000\text{ cm}^{-1}$), with 32 accumulations in transmission mode. Background spectra were also recorded with the same specifications. Post-processing of spectra was done with a lab-made script written in Matlab and includes baseline correction, digital paraffin subtraction using signal intensity at 1465 cm^{-1} , vector normalization to the amide I band and denoising using the Savitzky-Golay algorithm with a degree of 2 and a span length of 9. A quality control of each spectrum was performed by calculating the signal-to-noise ratio (SNR) over the spectral range $1850\text{--}2000\text{ cm}^{-1}$ free of biological signal. Second derivative were computed over the spectral range $1034\text{ cm}^{-1} - 1185\text{ cm}^{-1}$ and used as a loading vector for curve fitting. Spectral curve fitting quality was assessed by a root mean square error value set at 1% of the total spectral interval area. Intensities of sub bands located at $\sim 1127\text{ cm}^{-1}$ and $\sim 1152\text{ cm}^{-1}$ were normalized to the intensity of the $\sim 1655\text{ cm}^{-1}$ amide I band to estimate the lactic acid and glycogen contents, respectively, as reported (Hackett et al, 2016).

Reference: Hackett MJ, Sylvain NJ, Hou H, Caine S, Alaverdashvili M, Pushie MJ, Kelly ME. Concurrent glycogen and lactate imaging with FTIR spectroscopy to spatially localize metabolic parameters of the glial response following brain ischemia. *Anal Chem.* 2016;88(22):10949-10956.

The legends of Figures 4C&D and EV4:

- **4C**, page 40, line 981: "Immunohistochemical analysis of CD163 expression in lung adenocarcinoma. On the same tissue section, glycogen was detected by PAS staining, after treatment or not with amylase; Scale bar: $100\ \mu\text{m}$ ".
- **4D**, page 40, line 983: ".../... Lactic acid and glycogen quantification by FTIR spectroscopic imaging in lung adenocarcinoma. Left panel, unstained bright field image; right panel, glycogen and lactic acid maps; Scale bar, $200\ \mu\text{m}$; linear correlation between lactic acid and glycogen contents in infiltrating lung adenocarcinoma was calculated (results are representative of 1 out 2 biologically independent experiments).

- EV4, page 47, line 1101: *Immunohistochemical staining of TTF-1 in lung adenocarcinoma. Scale bar: 100 μm.*

References cited in the response to Reviewer:

Al-Shibli K, Al-Saad S, Donnem T, Persson M, Bremnes RM, and Busund LT (2009). The prognostic value of intraepithelial and stromal innate immune system cells in non-small cell lung carcinoma. *Histopathology*. 2009;55:301-12.

Conway EM, Pikor LA, Kung SH, Hamilton MJ, Lam S, Lam WL, Bennewith KL. Macrophages, Inflammation, and Lung Cancer. *Am J Respir Crit Care Med*. 2016;193:116-30.

Jackute J, Zemaitis M, Pranys D, Sitkauskiene B, Miliauskas S, Vaitkiene S, Sakalauskas R. Distribution of M1 and M2 macrophages in tumor islets and stroma in relation to prognosis of non-small cell lung cancer. *BMC Immunol*. 2018;19:3.

Mei J, Xiao Z, Guo C, Pu Q, Ma L, Liu C, Lin F, Liao H, You Z, Liu L. Prognostic impact of tumor-associated macrophage infiltration in non-small cell lung cancer: A systemic review and meta-analysis. *Oncotarget*. 2016;7:34217-28.

Welsh TJ, Green RH, Richardson D, Waller DA, O'Byrne KJ, and Bradding P (2005). Macrophage and mast-cell invasion of tumor cell islets confers a marked survival advantage in non-small-cell lung cancer. *J Clin Oncol*. 2005;23:8959-67.

4. To control the type of cells authors have been testing, authors need to present in fig 1 (or as supplementary material), the analysis of some human macrophages M1 and M2 markers.

As suggested, additional experiments have been performed to characterize the phenotype of M1 and M2 cells, using a set of membrane markers of macrophage polarization. It is known that human M1 cells express high levels of CD40, CD80 and CD86, while human M2 cells express elevated levels of CD14, CD163 and HLA-DR (Beyer et al., 2012; Hickman E et al., 2023; Bertani FR et al., 2017). Additional experiments were performed with cells from 7 different donors.

As expected, we observed that human M1 cells express high levels of CD40, CD80 and CD86, while human M2 cells express high levels of CD14, CD163 and HLA-DR (see Figure R1-4).

In other experiments, we have analyzed the phenotype of day 5 M1 and M2 cells (generated by culturing monocytes in CM and then in Glc^{low} Gln^{low} medium for the next 3 days; Figure. 2 of the submitted manuscript). Results revealed that the expression of the M1 and M2 markers is similar to the one of M1 and M2 cells generated in CM (see Figure. R1-4), showing that the culture medium switch during the differentiation process does not impact the polarization of macrophages.

Fig. R1-4: Phenotypic characterization of human M1 and M2 cells. M1 and M2 cells were generated either in conventional medium (CM) for 5 days or in CM for the first two days followed by 3 days culture in Glc^{low} Gln^{low} medium up to day 5. **A.** The expression of CD40, CD80, CD86, CD14, CD163, and HLA-DR was analyzed by flow cytometry. Results are expressed as relative fluorescence intensity (RFI) (n=7). Statistical significance was determined by two-tailed unpaired Welch t-test. **B.** Representative flow cytometry histograms; grey histograms represent isotype controls, dashed lines and solid lines represent cells generated in CM and in low medium, respectively. Values are represented as the mean ± SEM, each dot represents a different donor. Statistical significance was determined by two-tailed unpaired Welch t-test (A).

These additional results have been inserted in the revised version of the manuscript, in a new Expanded View Figure 1 and the Results section has been modified as follows:

Results section: paragraphs

- “GM-CSF or M-CSF plus IL-4 triggers glyconeogenesis in human macrophages” page 6, line 111: “... and exhibited conventional M1 (CD40^{high}, CD80^{high}, CD86^{high}, CD14^{low}, CD163^{low}, and HLA-DR^{low}) and M2 (CD40^{low}, CD80^{low}, CD86^{low}, CD14^{high}, CD163^{high}, and HLA-DR^{high}) profiles (Fig. EV1A&B).”

- “A combination of GM-CSF plus IFN γ promotes human macrophages glyconeogenesis” page 8, line 171: “...; the culture of cells in Glc^{low} Gln^{low} medium did not impact the acquisition of M1 and M2 specific markers (Fig. EV1A&B).”

Legend of Figure EV1A&B, page 45, line 1053: “**A-B.** Phenotypic characterization of M1 and M2 cells. M1 and M2 cells were generated either in conventional medium (CM) for 5 days or in CM for the first two days followed by 3 days culture in Glc^{low} Gln^{low} medium up to day 5. **A.** The M1 versus M2 phenotype was analyzed by flow cytometry using BUV395-labelled anti-CD14, BV421-labelled anti-CD163, BV510-labelled anti-CD40, FITC-labelled anti-CD80, PE-cy7-labelled anti-CD86 and APC-labelled anti-HLA-DR mAbs (all from BD Biosciences). Results are expressed as relative fluorescence intensity (RFI) (n=7). **B.** Representative flow cytometry histograms; grey histograms represent isotype controls, dashed lines and solid lines represent cells generated in CM and in low medium,

respectively... Values are represented as the mean \pm SEM, each dot represents a different donor. Statistical significance was determined by two-tailed unpaired Welch t-test (A).”

References cited in the response:

- Beyer M, et al. High-resolution transcriptome of human macrophages. PLoS One. 2012;7(9):e45466. Epub 2012/10/03.

- Hickman E, et al. Expanded characterization of in vitro polarized M0, M1, and M2 human monocyte-derived macrophages: Bioenergetic and secreted mediator profiles. PLoS One. 2023;18(3):e0279037.

- Bertani FR, et al. Classification of M1/M2-polarized human macrophages by label-free hyperspectral reflectance confocal microscopy and multivariate analysis. Sci Rep. 2017 Aug 21;7(1):8965.

5. Functional biology experiments. The use of LPS has its value (though please refer to my comments of the purity of the LPS tested). I urge the authors to challenge cells with live bacteria. This will allow them to assess whether the initial interaction (attachment) is affected (or not), and the effect on phagocytosis. Alternatively, authors may want to test *E. coli* pHRODO and *Staph aureus* pHrodo. Latex beads are used as a model but there is literature showing that differences exist when probing live bacteria.

In the submitted version of the manuscript, to assess the impact of glycogenolysis on human macrophage functions (cytokine secretion and phagocytosis), macrophages were stimulated with LPS purified by phenol extraction from *E. coli* serotype O111:B4 (Figure 5&6). The certificate of analysis of the supplier (Sigma-Aldrich) specifies 30×10^6 endotoxin unit/mg with 1.4% contaminant protein content for the batch used.

As suggested, in additional experiments, M1 and M2 cells were also stimulated with:

- Ultrapure LPS (LPS-EB UltraPure from *E. coli* O111:B4) purchased from InvivoGen. The certificate of analysis specifies bacterial endotoxin level $>2.5 \times 10^6$ EU/mg and the absence of activity on hTLR2 reporter cells (HEK-Blue hTLR2 cells; InvivoGen) (excluding the presence of peptidoglycan); as a positive control, LPS-EB induces a strong activation of HEK-Blue hTLR4 cells.
- Live Gram-negative *E. coli*.

Results showed that LPS from Sigma Aldrich (used in the submitted version of the manuscript), UltraPure LPS-EB and live *E. coli* induced IL-12 production by M1 cells and IL-10 by M2 cells (Figure R1-5A). Moreover, and whatever the stimulus, cytokine secretion by M1 and M2 cells was significantly reduced upon inhibition of glycogenolysis with the PYG inhibitor CP-91149 (Figure R1-5A).

We previously observed that the stimulation of M1 and M2 cells by LPS induced a reduction of glycogen stores, and that this reduction was prevented in the presence of a glycogenolysis inhibitor (Figure. 5A, in the revised version of the manuscript). In additional experiments, we have confirmed that the stimulation of M1 and M2 cells with UltraPure LPS-EB and live *E. coli* also induced a reduction in glycogen stores consecutive to an increased glycogenolysis; this reduction of glycogen stores was prevented by inhibiting glycogenolysis (Figure. R1-5B).

In the submitted version of the manuscript, we showed that the phagocytic capacity of M2 cells (evaluated using polystyrene microspheres) was reduced in the presence of CP-91149 (Figure 6B, right-hand panel). As suggested by Reviewer 1, we extended this observation using *E.coli*. Results showed that the PYG inhibitor reduced the phagocytosis of *E.coli* at a similar level to the one induced with cytochalasin B, an inhibitor of actin polymerization, used as a positive control (Figure R1-5C).

Figure R1-5. Role of glycogen in M1 and M2 cell functions. **a, b** Day 5 M1 and M2 cells (generated in CM) were treated or not for 15 min with CP-91149 before stimulation with 100 ng/mL LPS (*E. coli* serotype O111:B4; Sigma-Aldrich), 100 ng/mL LPS-EB Ultrapure (LPS UP; from *E. coli* serotype O111:B4; Invivogen) or with *E. coli* at a multiplicity of infection of 10. **a**. Cytokines were quantified by ELISA in the 24 h (IL-12p70) or 6 h (IL-10) culture supernatants (n=5). **b**. Glycogen was quantified after 24 h activation (n=3). **c**. M2 cells were pretreated with 50 µM CP-91149 or cytochalasin B (CytoB) (Sigma-Aldrich) for 15 min before addition of 0.2 mg/mL pHrodo-conjugated *E. coli* BioParticles (Invitrogen) for 2 h at 37°C. Fluorescence was determined by flow cytometry using a Canto II (BD Biosciences) and results were analyzed using FlowJo software and expressed in MFI values. Values are represented as the mean ± SEM, each dot represents a different donor. Statistical significance was determined by paired t-test (A, B) or by two-tailed unpaired Welch t-test (C).

In the revised version of the manuscript, and as requested, the results obtained with *E. coli* have been added in a new Expanded View Figure (EV5). The manuscript has been modified as follows:

Result section, paragraphs

- “Activated M1 and M2 cells catabolize glycogen”, page 13, line 289: “Similarly, live *E. coli* also induced a decrease of glycogen stores in M1 and M2 cells that was prevented by PYG inhibition (Fig. EV5A).”

- “Glycogen sustains M1 and M2 cell functions”, page 14

- line 302: "... and also IL-12 and IL-10 production by *E. coli*-stimulated M1 and M2 cells, respectively (Fig. EV5B)."

- line 305: "...and finally reduced polystyrene microspheres and of pHrodo *E. coli* phagocytosis by M2 cells (Fig. 6D and Fig. EV5D).

Results have been added in the revised **Figure EV5 A, B and D** and the legends have been modified as follows:

Page 47, line 1102, **Figure EV5. Glycogen sustains M1 and M2 cell functions.** A-B. Day 5 M1 and M2 cells (generated in conventional medium) were treated or not for 15 min with 50 μM CP-91149 before stimulation with 100 ng/mL LPS or with *E. coli* at a multiplicity of infection of 10. A. Glycogen was quantified after 24 h activation (n=3). B. Cytokines were quantified in the supernatants by ELISA after 24 h (IL-12p70) or 6 h (IL-10) (n=5). ... (line 1087) D. M2 cells were pretreated with 50 μM CP-91149 or 10 μM cytochalasin B (CytoB) for 15 min before addition of 0.2 mg/mL pHrodo-

conjugated *E. coli* BioParticles for 2 h at 37°C. Fluorescence was determined by flow cytometry and expressed in MFI values...Values are represented as the mean \pm SEM, each dot represents a different donor. Statistical significance was determined by paired t-test (A, B) or by two-tailed unpaired Welch t-test (C, D).

Responses to Reviewer 2.

In this study, Jeroundi et al investigated glycogen metabolism in human macrophages, and reported variations in glycogen level and synthesis between "M1" vs "M2" macrophage subtypes. M1 cells are capable of generating glycogen using gluconeogenesis substrates under low glucose condition. LPS stimulation reduced glycogen content in IFN γ or IL4 stimulated macrophages and inhibition of glycogenolysis reduced cytokine release suggesting glycogen is a fuel source for these essential macrophage functions. Overall, the strengths of this study include that it addresses a topic of broad scientific interest, providing significant amount of experimental data pertaining to stimulation-dependent glycogen metabolism in human macrophages, and the findings have health relevance for metabolic flexibility in nutrient depleted environments.

Major comments.

1. In the experiments where macrophages are differentiated in low glucose media, measurements of markers are needed to make sure the cells are differentiated, and in the case of M1 or M2 cells, they are indeed "M1" or "M2". The difference in glycogen metabolism could be due to cells not differentiating or polarizing.

We acknowledge Reviewer 2 for this comment. As pointed out, it was important to verify that culturing cells in a nutrient-poor medium had no impact on their polarization.

Additional experiments have thus been performed to characterize the phenotype of M1 and M2 cells by using a set of membrane markers of macrophage polarization. It was reported that human M1 cells express high levels of CD40, CD80 and CD86, while human M2 cells express elevated levels of CD14, CD163 and HLA-DR (Beyer et al, 2012; Bertani FR et al, 2017; Hickman E et al, 2023). The expression of these molecules was analyzed by flow cytometry on cells (i) cultured for 2 days in conventional medium followed by 3 days culture in Glc^{low} Gln^{low} medium or (ii) cultured for 5 days in conventional medium; experiments were performed using cells from 7 different donors.

As expected, we observed that human M1 cells express high levels of CD40, CD80 and CD86, while human M2 cells express high levels of CD14, CD163 and HLA-DR (see Figure R2-1).

As requested by Reviewer 2, results showed that the expression of M1 and M2 markers is similar on M1 and M2 cells whatever the culture conditions (see Figure R2-1), showing that the culture medium switch during the differentiation process does not affect macrophage polarization.

Figure R2-1: Phenotypic characterization of human M1 and M2 cells. M1 and M2 cells were generated either in conventional medium (CM) for 5 days or in CM for the first two days followed by 3 days culture in Glc^{low} Gln^{low} medium up to day 5. **A.** The expression of CD40, CD80, CD86, CD14, CD163, and HLA-DR was analyzed by flow cytometry. Results are expressed as relative fluorescence intensity (RFI) (n=7). Statistical significance was determined by two-tailed unpaired Welch t-test. **B.** Representative flow cytometry histograms; grey histograms represent isotype controls, dashed lines and solid lines represent cells generated in CM and in low medium, respectively. Values are represented as the mean ± SEM, each dot represents a different donor. Statistical significance was determined by two-tailed unpaired Welch t-test.

These additional results have been inserted in the revised version of the manuscript, in a new Expanded View Figure 1 and the Results section has been modified as follows:

Results section: paragraphs

- “GM-CSF or M-CSF plus IL-4 triggers in human macrophage glyconeogenesis” page 6, line 111: “... and exhibited conventional M1 (CD40^{high}, CD80^{high}, CD86^{high}, CD14^{low}, CD163^{low}, and HLA-DR^{low}) and M2 (CD40^{low}, CD80^{low}, CD86^{low}, CD14^{high}, CD163^{high}, and HLA-DR^{high}) profiles (Fig. EV1A&B).”

- “A combination of GM-CSF plus IFN γ promotes human macrophages glyconeogenesis” page 8, line 171: “...; the culture of cells in Glc^{low} Gln^{low} medium did not impact the acquisition of M1 and M2 specific markers (Fig. EV1A&B).”

Legend of Figure EV1A&B, page 45, line 1053: “**A-B.** Phenotypic characterization of M1 and M2 cells. M1 and M2 cells were generated either in conventional medium (CM) for 5 days or in CM for the first two days followed by 3 days culture in Glc^{low} Gln^{low} medium up to day 5. **A.** The M1 versus M2 phenotype was analyzed by flow cytometry using BUV395-labelled anti-CD14, BV421-labelled anti-CD163, BV510-labelled anti-CD40, FITC-labelled anti-CD80, PE-cy7-labelled anti-CD86 and APC-labelled anti-HLA-DR mAbs (all from BD Biosciences). Results are expressed as relative fluorescence intensity (RFI) (n=7). **B.** Representative flow cytometry histograms: grey histograms represent isotype controls, dashed lines and solid lines represent cells generated in CM and in low medium, respectively... Values are represented as the mean ± SEM, each dot represents a different donor. Statistical significance was determined by two-tailed unpaired Welch t-test (A).”

References cited in the response:

- Beyer M, et al. High-resolution transcriptome of human macrophages. PLoS One. 2012;7(9):e45466. Epub 2012/10/03.
- Hickman E, et al. Expanded characterization of in vitro polarized M0, M1, and M2 human monocyte-derived macrophages: Bioenergetic and secreted mediator profiles. PLoS One. 2023;18(3):e0279037.
- Bertani FR, et al. Classification of M1/M2-polarized human macrophages by label-free hyperspectral reflectance confocal microscopy and multivariate analysis. Sci Rep. 2017 Aug 21;7(1):8965.

2. Figure 1g suggested in GM-CSF macrophages the glycogen synthesis rate was 0.4 µg glycogen/million cells/day, so after 5 days that would be 2 µg, but the level in 1a is ~8 µg, what is the explanation?

We thank Reviewer 2 for this remark, that gives us an opportunity to clarify (i) that Fig. 1A and 1G do not represent the same thing, (ii) and that this difference may result from the different methods used to quantify intracellular glycogen:

Figure 1A illustrates total glycogen content after 5 days culture in CM (containing 2 g/L Glc and 2 mM Gln); the assay to quantify glycogen is based on the hydrolysis of glycogen to glucose (with glucoamylase) which is then oxidized into a product that reacts with OxiRed probe (colorimetric assay kit, # K646-100; BioVision, Milpitas, CA); results are expressed in µg glycogen per 10⁶ cells. In Figure 1G, cells were cultured in CM for 2 days and then in Glc^{low} medium (0.1 g/L Glc) supplemented with ¹³C₆-Glc for 3 days. The levels of intracellular glycogen were determined at day 5 by quantifying ¹³C-Glc by GC-MS, as described by Ma et al. (2020); contrary to the colorimetric assay, this protocol requires glycogen purification and hydrolysis before GC-MS analysis of glucose monomers; results are expressed in µg of ¹³C-Glc per 10⁶ cells per 24 h after subtraction of the levels of ¹²C-Glc present in glycogen synthesized by the cells between day 0 and day 2.

-The glycogen quantitation method was different between Figs. 1a and 1g and this might have introduced a small bias. This is nevertheless unlikely because glycogen extraction and hydrolysis are quantitative methods.

-Fig. 1g and Fig. 1a represent different things:

Fig. 1a represents the total glycogen pool observed after 5 days cultivation. That is, in Fig. 1a, the total glycogen pool can include glycogen that was preexisting in cells before cultivation and not only glycogen synthesized during the 5 days of cultivation.

By contrast, Fig. 1g represents the net flux of glycogen synthesis rate from ¹³C₆-glucose in the simplified medium (Glc^{low} medium), from days 3 to 5. As such: (1) it is a net flux (thus accounts for potential reutilization of ¹³C-glycogen), (2) it does not include preexisting, non-labelled glycogen (there is a small, but quantifiable, amount of glycogen at day 0 and 1-1.5 µg glycogen/10⁶ cells at day 2), and (3) the glycogen synthesis rate could vary with time, as is often observed in cultivated cells, with a progressive decrease in synthetic rate with cultivation time. Also, the glycogen synthesis rate might have been a bit slower in the simplified ¹³C-medium compared to the full unlabeled medium.

The results cannot thus be compared since, in one hand, they are expressed in "absolute" glycogen content (Fig. 1A), while, in the other hand, they are given as a ^{13}C net flux-enriched glycogen (Fig. 1G).

In the revised version of the manuscript, the technique used to purify and hydrolyze glycogen has been detailed in the Methods section and the Results section has been modified as follows:

Results section, paragraph "GM-CSF or M-CSF plus IL-4 triggers human macrophage glycogenesis", page 7, line 144: "...To confirm that extracellular glucose was used to synthesize glycogen, GM-CSF-M ϕ and M2 cells were generated in the presence of $^{13}\text{C}_6$ -Glc. At day 5, the intracellular glycogen pool was purified and then hydrolyzed to generate Glc monomers before quantification by GC-MS, as described²⁶. Results revealed the presence of ^{13}C -labelled Glc in both cell types (Fig. 1G and EV1C)." **Reference** 26. Gudewicz PW, Filkins JP. Glycogen metabolism in inflammatory macrophages (1976). J Reticuloendothel Soc. Aug;20(2):147-57

Methods section, paragraph "Glycogen extraction and hydrolysis, and glucose analysis by GC-MS", page 25, line 571: "The glucose content was evaluated by exact mass gas chromatography/mass spectrometry (GC-MS) by measuring ^{13}C enrichment using the mass spectrometer Orbitrap Q Exactive (Thermo Fisher Scientific). ^{13}C -enrichment data represent the ^{13}C atom excess whereby the contribution of natural ^{13}C abundance (determined in non-labelled samples) has been subtracted".

References cited in the response:

Ma J et al. Glycogen metabolism regulates macrophage-mediated acute inflammatory responses. Nat. Commun. 2020; 11:1-16.

3. More surprisingly, why is the glycogen synthesis flux from lactate and glutamine, particularly lactate, so small (Fig 2b), but glycogen storage level in glutamine or lactate supplemented cells almost as large as glucose supplemented?

We agree with this observation. We show in Figure 2A that M1 cells generated in a poor medium do not accumulate glycogen but store similar levels of glycogen after supplementation with Glc, LA or Gln; GM-M ϕ and M2 cells also accumulate similar amounts of glycogen, but only after Glc supplementation (Figure 2A). In these experiments, the total amount of glycogen is expressed in μg per 10^6 cells

To confirm glyconeogenesis, we show that $^{13}\text{C}_5$ -Gln and $^{13}\text{C}_3$ -lactic acid are metabolized by M1 cells to generate ^{13}C -glycogen (Figure 2B). To obtain this result, the same strategy was used with $^{13}\text{C}_5$ -Gln or $^{13}\text{C}_3$ -lactic acid to trace M1 cell glyconeogenesis from Gln and lactic acid (LA): First, monocytes were cultivated in a nutrient-rich medium with GM-CSF. Then at day 2, cells have been collected, counted and cultivation was reinitiated in a nutrient-poor medium supplemented with $^{13}\text{C}_5$ -Gln or $^{13}\text{C}_3$ -LA, with IFN γ . At day 5, total glycogen was quantified, purified and hydrolysed for GC-MS analysis of glucose monomers. The same experiment was performed with ^{13}C -glucose to confirm glycogenesis by GM-M ϕ and M2 cells (Figure 1G).

The order of magnitude of net glycogen synthesis flux from ^{13}C -Gln and ^{13}C -LA (Figure. 2B) is in the order of nanograms per million cells per day, i.e. nearly 2 orders of magnitude lower than with ^{13}C -glucose (Figure 1G).

These effects, were, however, expected and reflect classical effects observed during isotopic labelling:

1) Isotopic dilution: Gln and LA metabolization through the gluconeogenic pathway must be accompanied by an isotopic dilution along the path, due to the contribution of non-labelled, preexisting metabolic pools of intermediates, and thus the synthesis of only partially ^{13}C -labelled glucose to form glycogen.

2) Cellular Gln and LA production: Gln and LA produced by cells themselves via protein turn-over and organic acids metabolism (i.e. the participation of non-labelled Gln and LA) was also likely involved, thereby leading to an extra isotopic dilution.

3) Loss of ^{13}C : In addition, a fraction of Gln and LA can be used also by catabolism, leading to a loss of ^{13}C (in the form of $^{13}\text{CO}_2$). Additional experiments were carried out with $^{13}\text{C}_5$ -Gln, with or without a succinate dehydrogenase (SDH) inhibitor (Figure 3E). Since the TCA cycle is disrupted in murine M1 cells at SDH, we wondered whether the same phenomenon occurs in human macrophages (i.e. may also exhibit a truncated TCA cycle). Our results show that Gln can be metabolized via the classical TCA cycle (via succinate and malate) (Fig. 3E). These observations confirm that the metabolism of human and murine macrophages is different and that effectively, there is a metabolic partitioning of Gln between competing pathways: gluconeogenesis and catabolism. That is, only a fraction of supplied Gln is used by gluconeogenesis. For lactic acid, our previous data showed that lactic acid is internalized and metabolized by catabolism in inflammatory macrophages to generate malate and OAA, which can be in turn used as a substrate by gluconeogenesis via PEP (Paolini et al., 2020).

Taken as a whole, the observed net flux of ^{13}C -glycogen synthesis from ^{13}C -labelled Gln and LA is small while the total amount of glycogen is similar to that in Fig. 1 due to isotopic dilution (and thus the use of ^{12}C) and ^{13}C loss in metabolism, such that gluconeogenic glucose is much less ^{13}C -labelled than source Gln and LA.

Reference cited in the response:

Paolini, L. et al. Lactic acidosis together with GM-CSF and M-CSF induces human macrophages toward an inflammatory protumor phenotype. *Cancer Immunol Res.* 2020;8:383-395.

4. Important statistical comparisons were missing in the following figures: (i) Fig 1d: comparison between GM-CSF and GM-CSF + IL-4; (ii) Fig. 6e&f: comparison with or without CP.

We thank Reviewer 2 for this comment and apologize for the omission. Statistical analysis has been added in the revised version of the manuscript:

- **Figure 1D:** comparison of glycogen levels in macrophages generated in the presence of GM-CSF or GM-CSF plus IL-4. Figure 1D shows the glycogen levels in macrophages differentiated with GM-CSF (over 5 days), in the absence or presence of $\text{IFN}\gamma$, IL-1 β , IL-10, or IL-4 during the last three days. Statistical analysis indicates that only IL-4 (but not $\text{IFN}\gamma$, IL-1 β , and IL-10) significantly enhances glycogen levels in GM-CSF-M ϕ .

The revised version of the manuscript has been modified as follows:

- **legend of Figure 1D**, page 38, line 931. *GM-CSF or M-CSF plus IL-4 triggers human macrophage glycogenesis ...D. was quantified at day 5 (n=6-10) ... Values are represented as the mean ± SEM, each dot represents a different donor. Statistical significance was determined by Welch's ANOVA test followed by Dunnett's multiple comparison post hoc test.*

- **Results section**, paragraph "GM-CSF or M-CSF plus IL-4 triggers glycogenesis in human macrophages", page 7, line 135: "None of them affected glycogen content by GM-CSF-Mφ or M-CSF-Mφ, except IL-4 that significantly enhanced glycogen levels in GM-CSF-Mφ (Fig. 1D)".

- **Figure 6 E&F**: comparison of lactate release and oxygen consumption rate (OCR), in the absence or presence of the glycogenolysis inhibitor CP-91149. Results showed that the inhibition of glycogenolysis did not impact glycolysis and mitochondrial respiration, as assessed by quantifying extracellular lactate and OCR, respectively.

The revised version of the manuscript has been modified as follows:

- **legend of Figure 6**, page 42, line 1017, "Glycogen sustains M1 and M2 cell functions: ... was monitored after 2h (n=4-6) ... Values are represented as the mean ± SEM, each dot represents a different donor. Statistical significance was determined by Welch's ANOVA test followed by Dunnett's multiple comparison post hoc test.

- Results section, paragraph “Glycogen sustains M1 and M2 cell functions”, page 15, line 333: *"Finally, the inhibition of glycogenolysis had no effect on glycolysis (as assessed by quantifying lactate production) and mitochondrial respiration (as assessed by measuring oxygen consumption) in LPS-stimulated M1 (Fig. 6E) and M2 cells (Fig. 6F), suggesting that macrophage respiratory metabolism is powered by extracellular Glc rather than glycogen degradation products"*.

5. Figure 3e is hard to interpret. It is better to present the data as labeling fraction. Also it is surprising that SDH inhibitor did not cause a significant decrease in 4-labeled fumarate.

As pointed out by Reviewer 2, the results presented in the submitted version of Figures 3E&F were difficult to interpret.

In fact, the abundance of the metabolites (Figure 3F in the submitted version) were presented independently from the schematic representation of Gln metabolism (Figure. 3E, submitted version). Furthermore, results on the impact of SDH inhibition were obtained using cells from only 3 different donors.

In order to facilitate the reading of the results, we have combined Figures 3E and 3F into a new single Figure 3E which includes the labelling fraction of each metabolite generated after $^{13}\text{C}_5\text{-Gln}$ metabolism by M1 cells.

Additional experiments of $^{13}\text{C}_5\text{-Gln}$ metabolism by M1 cells isolated from 8 different donors have been performed. Results show a significant decrease in 4-labeled fumarate after SDH inhibition (see Figure R2-2); these results are now presented in a new Figure 3E in the revised manuscript.

Figure R2-2. Glyconeogenesis pathways in M1 cells. Day 2 GM-CSF-M ϕ were switched to Glc^{low} Gln^{low} medium containing GM-CSF and IFN γ . At day 3, cells were treated or not with the SDH inhibitor NV-161 and supplemented with ^{13}C -Gln for 30 min. Relative abundance of ^{13}C -metabolites was determined by LC-MS; results are expressed as relative abundance of each metabolite generated after Gln metabolism. Results are expressed in arbitrary units (AU; mean \pm SEM, n=8). Statistical significance was determined by 2-way ANOVA test.

In the revised version of the manuscript, the Results section and the legend of Figure 3 have been modified as follows:

Results section, paragraph “A combination of GM-CSF plus IFN γ promote human macrophage glyconeogenesis”, page 10, line 223: “ ^{13}C -succinate content was significantly increased in the presence of the succinate dehydrogenase (SDH, complex II) inhibitor NV-161, whereas the levels of fumarate and malate were significantly decreased (Fig. 3E), indicating .../...”

Legend of the revised Figure 3E: page 40, line 966, “M1 cells glyconeogenesis pathways” .../... *E.* Day 2 GM-CSF-M ϕ were switched to Glc^{low} Gln^{low} medium containing GM-CSF and IFN γ . At day 3, cells were treated or not with the SDH inhibitor NV-161 and supplemented with ¹³C₅-Gln for 30 min. Relative abundance of ¹³C-metabolites was determined by LC-MS and results are expressed as relative abundance of each metabolite generated after Gln metabolism. Results are presented as arbitrary unit (AU; n=8). Values are represented as the mean \pm SEM, each dot represents a different donor. Statistical significance was determined by two-tailed paired t-test (E).”

6. The results showed some evidence that TAMs have the capacity to synthesize glycogen when stimulated, however, it is unclear if they do so to a meaningful extent in tumor. The physiological relevance of this part is unclear.

The objective of these experiments was to evaluate whether ex vivo isolated established macrophages, as observed with monocytes-derived macrophages, contain glycogen stores and/or are capable of glycogenesis and glyconeogenesis. We thus used macrophages isolated from ascites of patients with ovarian cancer as a model of established tumor-associated macrophages (TAMs) (Nowak M & Klink M. 2020).

We observed that freshly isolated TAM do not contain glycogen but synthesize it using Glc or lactic acid after culture in the presence of GM-CSF and IFN γ (Figure 4A). Results also evidenced the presence of glycogen in the cytoplasm of TAM infiltrating lung adeno carcinoma (as assessed by PAS staining, Figure. 4C).

That TAMs exhibited low glycogen stores after purification (Figure 4A) probably results from the low concentrations of cytokines enabling glycogenesis and glyconeogenesis in ovarian tumor ascites (GM-CSF: <100 pg/ml, IFN γ and IL-4: <10 pg/ml) rather than from an absence of fuel sources, as Glc and lactic acid are detectable in these ascites (4.1 ± 1.2 mM and 4.8 ± 3 mM, respectively; n=10).

This point is now mentioned in the Results section, paragraph “Tumor associated macrophages store glycogen and are capable of glyconeogenesis”, page 11, line 251, as follows: “Consistent with this observation, ascites fluids had detectable levels of Glc (< 5 mM), and low levels of IL-4 and IFN γ (<10 pg/ml), and GM-CSF (<100 pg/ml), which are required for glycogenesis and glyconeogenesis by human macrophages”.

In additional experiments, we verified that glycogen is necessary for TAM to function (as observed for in vitro generated macrophages). TAMs were cultured in Glc^{low} Gln^{low} medium containing GM-CSF and IFN γ and supplemented or not with Glc or lactic acid. At day 3, cells were stimulated with LPS to monitor cytokine secretion, in the presence or absence of the glycogenolysis inhibitor CP-91149. Results showed that, as observed for monocyte-derived macrophages, CP-9114 significantly reduced the production of TNF α , VEGF and G-CSF by LPS-stimulated TAM (see Figure R3-2). These results suggest that, similar to in vitro generated macrophages, glycogen metabolism support TAM function.

Figure R2-3. Glycogen metabolism is necessary for TAM function. TAM isolated from ovarian cancer ascites were cultured with GM-CSF and IFN γ in Glc^{low} Gln^{low} medium supplemented with Glc or lactic acid. At day 3, cells were treated or not with CP-91149 for 15 min before a 6 h stimulation with LPS. TNF α , VEGF and G-CSF were quantified by ELISA (n=3). Values are represented as the mean \pm SEM, each dot represents a different donor. Statistical significance was determined by paired t-test.

These new results have been inserted in the revised Figure 4 and the Results and Methods sections have been modified as follows:

Results section, paragraph “Glycogen sustains M1 cell and M2 cell functions”, page 14, line 303: “The inhibition of glycogenolysis also reduced cytokine production by .../... and by TAM with glycogen stores generated from Glc or lactic acid (Fig. 4E).”

Methods section, paragraph “Tumor associated macrophages”, page 23, line 509: “.../... For glycogenolysis experiments, day 3 TAM were treated or not with 50 μ M CP-91149 for 15 min before a 6 h stimulation with 100 ng/mL LPS”.

Legend of the new Figure 4E, page 40, line 987: “.../... Day 3 TAM were treated or not with CP-91149 for 15 min before a 6 h stimulation with LPS. TNF α , VEGF and G-CSF were quantified by ELISA (n=3). Values are represented as the mean \pm SEM, each dot represents a different donor. Statistical significance was determined... by paired t-test (E).”

In solid tumors, M-CSF and inflammatory cytokines, such as GM-CSF and IFN γ , are present along with lactic acid. This led us to investigate the presence of glycogen in TAM infiltrating lung adenocarcinoma (and not in ovarian cancer tissues since the abundance of mucins in these tumors interfered with glycogen detection). PAS staining revealed the presence of glycogen in TAMs (Figure 4C).

In additional experiments, Fourier-transform infrared spectroscopy (FTR) confirmed the colocalization of lactic acid and glycogen in tissue areas enriched with PAS-positive macrophages. A significant linear correlation between glycogen and lactic acid levels ($R^2=0.87$, $p<0.0001$) suggests that lactic acid may serve as a substrate for glycogen synthesis in vivo (see Figure R2-4). Given that GM-CSF and IFN- γ , along with elevated lactic acid, are present in solid tumors, this finding supports the concept that TAMs can utilize lactic acid to generate glycogen.

Figure R2-4. Lactic acid and glycogen quantification by FTIR spectroscopic imaging in lung adenocarcinoma. *Left panel*, unstained brightfield image; *right panels*, glycogen and lactic acid maps (Scale bar: 200 μm); linear correlation between lactic acid and glycogen contents in infiltrating lung adenocarcinoma was calculated (results are representative 1 out of 2 independent experiments).

In the revised version of the manuscript, these additional results have been added in Figures 4 and the Results and Methods sections and the corresponding Legends have been modified as follows:

Results section, paragraph "Tumor-associated macrophages store glycogen and are capable of glyconeogenesis" page 12, line 264: "... *The presence of glycogen in macrophages was confirmed by FTIR spectroscopic imaging in serial sections. FTIR spectroscopy also revealed a colocalization of glycogen with lactic acid ($R^2 = 0.87$, $p < 0.0001$) (Fig. 4D). Given that GM-CSF and IFN- γ , along with elevated lactic acid, are present in solid tumors, this finding suggest that TAMs can use lactic acid in vivo to generate glycogen (Fig. 4D).*

Methods section, a new paragraph has been added to describe the "FTIR spectroscopic imaging", page 27, line 621: "*Five-micrometer-thick section of the paraffin-embedded tumor were deposited on BaF2 windows. Spectral analysis was performed with a Bruker Hyperion 3000 infrared microscope coupled to a Vertex 70 spectrometer using a 64×64 focal plane array (FPA) detector. A $15\times$ Cassegrain objective (Numerical Aperture 0.4) was used for all acquisition. Mid-infrared spectra were recorded at a resolution of 4 cm^{-1} (spectral region $900\text{--}2000\text{ cm}^{-1}$), with 32 accumulations in transmission mode. Background spectra were also recorded with the same specifications. Post-processing of spectra was done with a lab-made script written in Matlab and includes baseline correction, digital paraffin subtraction using signal intensity at 1465 cm^{-1} , vector normalization to the amide I band and denoising using the Savitzky-Golay algorithm with a degree of 2 and a span length of 9. A quality control of each spectrum was performed by calculating the signal-to-noise ratio (SNR) over the spectral range $1850\text{--}2000\text{ cm}^{-1}$ free of biological signal. Second derivative were computed over the spectral range $1034\text{--}1185\text{ cm}^{-1}$ and used as a loading vector for curve fitting. Spectral curve fitting quality was assessed by a root mean square error value set at 1% of the total spectral interval area. Intensities of sub bands located at $\sim 1127\text{ cm}^{-1}$ and $\sim 1152\text{ cm}^{-1}$ were normalized to the intensity of the $\sim 1655\text{ cm}^{-1}$ amide I band to estimate the lactic acid and glycogen contents, respectively, as reported (Hackett et al, 2016)*

Reference: Hackett MJ, Sylvain NJ, Hou H, Caine S, Alaverdashvili M, Pushie MJ, Kelly ME. Concurrent Glycogen and Lactate Imaging with FTIR Spectroscopy To Spatially Localize Metabolic Parameters of the Glial Response Following Brain Ischemia. Anal Chem. 2016 Nov 15;88(22):10949-10956.

The legends of Figures 4D, page 40, line 983: ".../... Lactic acid and glycogen quantification by FTIR spectroscopic imaging in lung adenocarcinoma. Left panel, unstained bright field image; right panel, glycogen and lactic acid maps; Scale bar, 200 μ m; linear correlation between lactic acid and glycogen contents in infiltrating lung adenocarcinoma was calculated (results are representative of 1 out of 2 biologically independent experiments).

References cited in the response:

Nowak M, Klink M. The role of Tumor-Associated Macrophages in the progression and chemoresistance of ovarian cancer. *Cells*. 2020;9(5):1299.

Hackett MJ, Sylvain NJ, Hou H, Caine S, Alaverdashvili M, Pushie MJ, Kelly ME. Concurrent glycogen and lactate imaging with FTIR spectroscopy to spatially localize metabolic parameters of the glial response following brain ischemia. *Anal Chem*. 2016;88(22):10949-10956.

7. The support of the argument that glycogenolysis "fuel PPP, enabling human M1 and M2 cells to perform their functions" is very correlative - It was demonstrated that glycogenolysis inhibition decreases macrophage functions and that PPP inhibition decreases macrophage functions, but did not show direct evidence that glycogenolysis fuel PPP.

We agree with this comment. Results showed that LPS-induced cytokine production was associated with a decrease of glycogenolysis and that the inhibition of glycogenolysis and of PPP reduced the levels of cytokine secretion and prevented glycogen depletion (Figures 5 and 6). More precisely, glycogen levels in M1 and M2 cells decreased after LPS stimulation, and this decrease was partially prevented by the PPP inhibitor 6AN (Figure 5A). PPP inhibition (with chemical drug or siRNA) also decreased cytokine production by LPS-stimulated M1 and M2 cells (Figure 6 A-D).

Our conclusion (i.e. glycogenolysis feeds the PPP pathway) was drawn based on this last observation and was supported by results from Ma et al who reported similar results in murine M1 macrophages (Ma et al., 2022). As pointed out by Reviewer 2, these results only suggest, but do not prove, that glycogenolysis fuels PPP.

Accordingly, the introduction and the Results section, have been modified as follows:

Introduction section (page 5, line 97) "...We found that glycogenolysis fueled at least in part the pentose phosphate pathway (PPP)".

Results section, paragraph "Glycogen sustains M1 and M2 cell functions" (page 15, line 324): "These results suggest that glycogenolysis enables human M1 and M2 cells to fulfill their functions, at least in part through PPP."

8. Overall, the paper showed that macrophages can generate glycogen from gluconeogenesis in low glucose condition, and glycogen break down can be important for macrophage functions. However, the physiological situation is unclear. In the body, when will macrophage generate glycogen? When would they break it down? At the site of infection, where glucose levels may be

low and cells are exposed to stimulation like LPS, would macrophages mainly synthesize or break down glycogen? Or the glycogen is synthesized from glucose in glucose replete conditions where macrophage differentiate, then broken down in low-glucose infection site?

8-1. In the body, when will macrophage generate glycogen?

According to our observations, M-CSF plus IL-4 and GM-CSF allow non-stimulated macrophages to synthesize and accumulate glycogen from Glc (Figure 1A).

M-CSF is constitutively present in tissues but only triggers glycogenesis in the presence of IL-4 (which is mainly expressed during Th2-polarized immune responses). GM-CSF and IFN γ , secreted in inflamed tissues, enable macrophages to use other glycogenic substrates, such as lactic acid, Gln or glycerol (Figure 2A). We hypothesize that, in inflammatory conditions (presence of GM-CSF and IFN γ), in the absence of stimulation, macrophages store glycogen synthesized using different substrates, including lactic acid. The capacity of TAM to store glycogen tend to confirm this hypothesis (Figure 4A,C&D).

8-2. When would they break it down? At the site of infection, where glucose levels may be low and cells are exposed to stimulation like LPS, would macrophages mainly synthesize or break down glycogen?

Our results suggest that:

- in the absence of stimulation, macrophages synthesize and store glycogen using Glc or, depending on its levels, lactic acid, glutamine or glycerol as substrates. In these conditions, glycogenesis and glyconeogenesis are predominant over glycogenolysis, leading to the accumulation of glycogen.
- Upon stimulation, the balance between glycogen synthesis and degradation shifts towards degradation, even if extracellular glucose is available (Figure 5A).

Others have shown that glycogenesis and glycogenolysis by dendritic cells occur concomitantly in response to stimulation (Thwe et al. 2017). Nevertheless, as pointed out by Reviewer 2, our results did not allow to determine whether inflammatory macrophages could concomitantly use Glc and non-carbohydrate substrates to synthesize glycogen. This would enable to determine the substrates used by M1 cells under physiological conditions, when glucose and other substrates are available. Additional experiments have thus been performed to determine whether M1 cells depend on glyconeogenesis, even in the presence of glucose.

Monocytes from 5 different donors were differentiated in M1 cells by culture with GM-CSF plus IFN γ in CM medium containing 2 g/L Glc and either 2 mM Glutamine (Gln) (physiological condition) or 20 mM Gln (over-supplementation). To ensure Glc availability throughout the differentiation process, the medium was replaced with fresh medium at day 2. We observed that glycogen stores were higher in cells cultured with 20 mM Gln than in cells cultured with 2 mM Gln, suggesting that M1 cells may use Gln to synthesize glycogen even in the presence of Glc (see Figure R2-5). To confirm this result, we determined glycogen stores after blocking PCK2, a key enzyme of glyconeogenesis. M1 cells treated with the PCK2 inhibitor, PEPCKi, contained lower levels of glycogen than untreated cells, regardless of extracellular Gln availability (see Figure R2-5). These findings indicate that, under inflammatory conditions (presence of GM-CSF plus IFN γ), macrophages can use both Glc and non-carbohydrate substrates to synthesize glycogen.

Figure R2-5: M1 cells rely on glycogenesis and glyconeogenesis to synthesize glycogen. Glycogen content was determined in M1 cells cultured for 5 days in CM containing either 2 or 20 mM Gln, in the absence or presence of 10 µM PEPCKi (n=5). Values are represented as the mean ± SEM, each dot represents a different donor. Statistical significance was determined by paired t-test.

In the revised manuscript, these results are presented in a new Figure EV2D and the Results and Discussion sections have been modified as follows:

Results section, paragraph “A combination of GM-CSF plus IFN γ promotes human macrophage glyconeogenesis”, page 11, line 228: “We then evaluated whether M1 cells can synthesize glycogen using concomitantly glycogenesis and glyconeogenesis. Day 5 M1 cells were generated in CM containing 2 g/L Glc and either 2 or 20 mM Gln. Glycogen stores were higher in cells cultured with 20 mM Gln compared to cells cultured with 2 mM Gln, suggesting that they can use Gln to synthesize glycogen even when Glc is available (Fig. EV2D). In addition, glycogen levels in M1 cells were lower in the presence of a PCK2 inhibitor, regardless of Gln concentration (Fig. EV2D), suggesting that M1 cells can use simultaneously Glc and non-carbohydrate substrates to synthesize glycogen.”

Legend of the figure EV2D, page 46, line 1085: “M1 cells rely on glycogenesis and glyconeogenesis to synthesize glycogen. Glycogen content was determined in M1 cells cultured for 5 days in CM containing either 2 or 20 mM Gln, in the absence or presence of 10 µM PEPCKi (n=5). Values are represented as the mean ± SEM, each dot represents a different donor. Statistical significance was determined... by paired t-test (D).

Discussion section:

- page 18, line 410: “...To conclude, our results suggest that in the absence of stimulation, macrophages synthesize and store glycogen from glucose. Under inflammatory conditions, macrophages acquire the ability to use additional substrates to synthesize glycogen.”
- page 19, line 424: “.../... Upon stimulation, the balance between glycogen synthesis and degradation shifts towards degradation, even if extracellular glucose is available.”

Minor comments.

1. What timepoint was the data in Fig. 1c stained? This information should be included.

Monocytes (D0) and day 5 M1 and M2 cells were stained with PAS for 15 min or internalized with 2-NBDG during 2 h. The time point for 2-NBDG internalization or PAS staining is now included in the method section of the manuscript and the legend of the Figure 1C.

Furthermore, as requested by Reviewer 1, additional PAS and 2-NBDG staining experiments were performed to determine the number of positive cells.

PAS staining and 2-NBDG fluorescence were analyzed by light and confocal fluorescence microscopy, respectively. Cells from 5 (for PAS) or 3 (for 2-NBDG) different donors were used; \approx 300 cells per donor were analyzed and images were processed using Image J software.

These results show that most if not all cells (monocytes, M1 cells and M2 cells) are positive for PAS and 2-NBDG staining (> 95% positive cells; see figure R2-6).

By normalizing signal intensity to cell number, PAS staining and 2-NBDG fluorescence confirm that the signal intensity is higher in M1 and M2 cells than in monocytes (3.47 and 3.32 times higher for M1 and M2 with PAS staining and 3.64 and 4.14 times higher for M1 and M2 with 2-NBDG fluorescence) (see figure R2-6). These results are consistent with the results obtained by the quantification of intracellular glycogen by colorimetric assay showing higher glycogen levels in M1 and M2 cells than in monocytes (R2.7 ; Figure.1A)

Staining	Monocytes	M1 cells	M2 cells
PAS	100%	100%	95%
2-NBDG + DAPI	100%	100%	100%

Figure R2-6. *Upper panels*, Monocytes and day 5 M1 and M2 cells were stained with PAS (one out of 5 different donors) or 2-NBDG (one out of 3 different donors) for 2 h; 50-350 cells by field are presented. *Lower panels*, Table summarizes the frequency of positive cells after PAS staining (Scale bar: 20 μ m) or 2-NBDG internalization (Scale bar: 10 μ m).

Figure R2-7: PAS staining and 2-NBDG uptake in M1 and M2 cells. 1C. *Left panels:* monocytes and day 5 M1 and M2 cells were analyzed by light microscopy (Scale bar: 50 μ m) after PAS staining or by confocal microscopy after 2-NBDG internalization (Scale bar: 40 and 2 μ m). For fluorescent microscopy analysis, nuclei were counterstained with DAPI. Results are representative of 1 out of 3 (for 2-NBDG) or 5 (for PAS) different donors. *Right panels:* relative intensity of PAS staining and 2-NBDG fluorescence (determined using the Image J software) in monocytes and in M1 and M2 cells (detailed in the Material and Methods section). 300 cells were analyzed per experimental condition; results were obtained from cells isolated from 3-5 different donors.

In the revised version of the manuscript, these new results are presented in Figure 1C, right panels; the text has been modified as follows:

Results section, paragraph “GM-CSF or M-CSF plus IL-4 triggers glycogenesis in human macrophages”, page 7, line 129: “*Image analysis showed that >95% of monocytes and day 5 M1 and M2 cells were positive for PAS staining or 2-NBDG internalization (Fig. 1C, left panels and data not shown) and confirmed that M1 and M2 cells contain higher levels of glycogen than monocytes (Fig. 1C, right panels).*”

Material and Methods section, paragraph “Glycogen detection and quantification”:

- page 23 line 513: “*monocytes and day 5 macrophages...PAS staining for 15 minutes...Images were acquired on a Leica DMR microscope (100 \times magnification) and on a Zeiss Axioskop 2 MOT microscope (20 \times magnification) and images were analyzed using the Image J software...2h in Glc-free and Gln-free*”.

- page 24 line 528: “*Images were acquired on a Leica SP8 confocal microscope (Leica Microsystems, Nanterre, France) and images were analyzed using the Image J software*”.

- **The legend of Figure 1C**, page 38, line 924: “*Left panels, monocytes and day 5 M1 and M2 cells were analyzed by light microscopy after PAS staining (upper panels; scale bar: 50 μ m) or by confocal microscopy after 2 h 2-NBDG internalization (middle and lower panels; scale bar: 40 and 2 μ m). For fluorescent microscopy analysis, nuclei were counterstained with DAPI. Results are representative of 1 out of 3 (for 2-NBDG) or 5 (for PAS) different donors. Right panels, relative intensity of PAS staining and 2-NBDG fluorescence in monocytes and day 5 M1 and M2 cells. 300 cells/donor were analyzed (n=3-5). Values are represented as the mean \pm SEM, each dot represents a different donor. Statistical significance was determined by two-tailed unpaired Welch t-test (C).*”

2. In Figure 1b the authors claim 'IL-4 triggered a time-dependent accumulation of glycogen in M-CSF-M ϕ , this effect peaking after three days of incubation.' The data suggests glycogen is still accumulating throughout day 5, not reaching a peak. At day 3 IL4 appears to be significantly different from M-CSF alone.

This information was mentioned in the Figure 1B in the submitted version of the manuscript. Figure 1B shows the time-dependent effect of IL-4 on glycogen synthesis by M-CSF-M ϕ . A 24 h incubation with IL-4 induced much lower glycogen levels than a 3 day incubation (see Figure R2-8). We agree with the remark of the Reviewer: in the initial version, the incubation time with IL-4 may have been incorrectly explained, as IL-4 was added at day 2 and analysis was performed at day 5.

Consequently, the revised manuscript has been modified as follows:

Results section, the paragraph “GM-CSF or M-CSF plus IL-4 triggers human macrophage glycogenesis”, page 6, line 121: “*The addition of IL-4 at day 2 to differentiating M-CSF-M ϕ triggered a small but significant accumulation of glycogen after 24 h (D3) and a substantial accumulation after 3 days (D5) (Fig. 1B).*”

Discussion section, page 16, line 360: “*Based on our observations, we suspect that the lack of glycogen reserves reported by others in murine M2 cells generated with M-CSF and IL-4 could be due to an insufficient incubation time with IL-4, which was limited to only 24 hours¹⁸.*”

References: 18. Ma, J. et al. Glycogen metabolism regulates macrophage-mediated acute inflammatory responses. Nat. Commun.11, 1–16 (2020).

Moreover, in the initial Figure 1B, the results were obtained with cells from 3 donors. In the revised version, additional experiments with monocytes from two additional subjects were performed (see Figure R2-8).

Figure R2-8. GM-CSF or M-CSF plus IL-4 triggers human macrophage glycogenesis.... B. Day 2 M-CSF-M ϕ were exposed to IL-4 (purple) or not (Grey) for the last 3 days. Glycogen was quantified in monocytes (D0) and after 24 h (D3), 48 h (D4) and 72 h (D5) incubation with or without IL-4 (n=5-8). Values are represented as the mean \pm SEM, each dot represents a different donor. Statistical significance was determined by two-tailed unpaired Welch paired t-test.

3. Figure citation at the end of TAM results "PAS staining abolished by α -amylase (which cleaves the α 1 \rightarrow 4 glucoside bonds in glycogen) was detected, confirming the presence of glycogen in CD163-positive cells (Fig. S4c.)" Should it be Fig.4c?

We thank Reviewer 2 for this comment. In the revised version of the manuscript, the presence of CD163-positive macrophages has been added next to the PAS staining (Fig. 4C), showing the presence of glycogen in TAM infiltrating lung adenocarcinoma. Of note, the presence of glycogen in macrophages was confirmed by FTIR spectroscopic imaging (see response to comment 6).

Responses to Reviewer 3.

Here Jeroundi et al. examine the use of glycogen as a fuel system in human macrophages. Using in vitro derived cells with GM-CSF or M-CSF, and in some cases IFN γ or IL4, they show that macrophages can accumulate glycogen and that this glycogen store can be used to support inflammatory cytokine production, even when glucose is present. Overall, the in vitro data are convincing and well presented as concerning the blood derived macrophage populations. The paper uses inhibitors, and siRNA to test some of the targets suggested in the glycogen pathway etc. The data involving tumor-derived macrophages (TAM), are less convincing. There are just two substantial concerns regarding the in vitro-derived cells.

1-1. First, is the apparent lack of effect of IFN γ shown in figure 1a. Despite the fact that IFN γ does little to alter the glycogen accumulation several other figures use this idea of M1 cells. I understand the distinction here is that the IFN γ apparently permits the use of the alternative fuels for glycogen whereas the GM-CSF cells are limited to the use of glucose, but the way the authors describe and refer to M1 is confusing.

We agree with this comment. Figure 1A shows that IFN γ does not modulate glycogen levels in GM-CSF-M ϕ and Figure 1F shows that GM-CSF-M ϕ and M2 cells mainly use Glucose (Glc) to synthesize glycogen (as no glycogen accumulation was observed in Glucose-free medium). The capacity of M1 cells to perform glyconeogenesis is shown in Figure 2A (and not in Figure 1F). To simplify the message, we have chosen to present first the cytokine combinations that trigger glyconeogenesis (Figure 1) and then (ii) the cytokine combination that allows glyconeogenesis (Figure 2).

As requested, and to clarify the message concerning glycogen synthesis by M1 cells, the Results section has been modified as follows: paragraph “A combination of GM-CSF plus IFN γ promotes human macrophage glyconeogenesis”, page 8 , line 168: *“Although GM-CSF and M-CSF plus IL-4 induce macrophage glyconeogenesis (Fig. 1F), we hypothesized that some macrophage subtypes, and especially M1 cells (generated in the presence of GM-CSF plus IFN γ), may perform glyconeogenesis in a nutrient-poor environment.”*

The comments of the Reviewer helped us to realize that we did not evaluate whether inflammatory macrophages (M1 cells) can use simultaneously glucose and non-carbohydrate substrates to synthesize glycogen. This would enable to determine the substrates used by M1 cells under physiological conditions, when glucose and other substrates are available. Additional experiments have thus been performed to determine whether M1 cells depend on glyconeogenesis, even in the presence of glucose.

Monocytes from 5 different donors were differentiated in M1 cells by culture with GM-CSF plus IFN γ in CM medium containing 2 g/L Glc and either 2 mM Glutamine (Gln) (physiological condition) or 20 mM Gln (over-supplementation). To ensure Glc availability throughout the differentiation process, the medium was replaced with fresh medium at day 2. We observed that glycogen stores were higher in cells cultured with 20 mM Gln than in cells cultured with 2 mM Gln, suggesting that M1 cells may use Gln to synthesise glycogen even in the presence of Glc (see Figure R3-1). To confirm this result, we determined glycogen stores after blocking PCK2, a key enzyme of glyconeogenesis. M1 cells treated with the PCK2 inhibitor, PEPCKi, contained lower levels of glycogen than untreated cells, regardless of extracellular Gln availability (see Figure R3-1). These findings indicate that, under inflammatory conditions (presence of GM-CSF plus IFN γ), macrophages can use both Glc and non-carbohydrate substrates to synthesize glycogen.

Figure R3-1: M1 cells rely on glycogenesis and glyconeogenesis to synthesize glycogen. Glycogen content was determined in M1 cells cultured for 5 days in CM containing either 2 or 20 mM Gln, in the absence or presence of 10 µM PEPCKi (n=5). Values are represented as the mean ± SEM, each dot represents a different donor. Statistical significance was determined by paired t-test.

In the revised manuscript, these results are presented in a new Figure EV2D and the Results and Discussion sections have been modified as follows:

Results section, paragraph “A combination of GM-CSF plus IFN γ promotes human macrophage glyconeogenesis”, page 11, line 228: “We then evaluated whether M1 cells can synthesize glycogen using concomitantly glycogenesis and glyconeogenesis. Day 5 M1 cells were generated in CM containing 2 g/L Glc and either 2 or 20 mM Gln. Glycogen stores were higher in cells cultured with 20 mM Gln compared to cells cultured with 2 mM Gln, suggesting that they can use Gln to synthesize glycogen even when Glc is available (Fig. EV2D). In addition, glycogen levels in M1 cells were lower in the presence of a PCK2 inhibitor, regardless of Gln concentration (Fig. EV2D), suggesting that M1 cells can use simultaneously Glc and non-carbohydrate substrates to synthesize glycogen.”

Legend of the figure EV2D, page 46, line 1085: “M1 cells rely on glycogenesis and glyconeogenesis to synthesize glycogen. Glycogen content was determined in M1 cells cultured for 5 days in CM containing either 2 or 20 mM Gln, in the absence or presence of 10 µM PEPCKi (n=5). Values are represented as the mean ± SEM, each dot represents a different donor. Statistical significance was determined by paired t-test.

Discussion section page 18, line 410: “...To conclude, our results suggest that in the absence of stimulation, macrophages synthesize and store glycogen from glucose. Under inflammatory conditions, macrophages acquire the ability to use additional substrates to synthesize glycogen”.

1-2 Second are the tracing data meant to show incorporation of ^{13}C into glycogen. Controls are lacking here, leaving us with the idea that because it is detectable, this is a relevant process. These levels of ^{13}C incorporation need to be compared with some condition that helps us understand the level that would be considered "positive" by the authors.

In ^{13}C tracing experiments, results are expressed as ^{13}C -atom excess in glycogen after subtracting isotope (^{13}C) abundance in control experiments (no ^{13}C -Glc added); the later corresponds to the natural abundance of ^{13}C (1.1%). In other words, experiments under “isotopic control conditions” (natural abundance) have been performed, and natural ^{13}C abundance has been subtracted, determining the net flux of ^{13}C -glycogen synthesis. The results presented therefore correspond to the ^{13}C supplied to the cells and used to generate glycogen (and cannot originate from any other carbon source).

The Results and the Methods sections have been modified as follows:

Results section, paragraph “GM-CSF or M-CSF plus IL-4 triggers human macrophage glycogenesis”, page 7, line 144: “...*To confirm that extracellular glucose was used to synthesize glycogen, GM-CSF-M ϕ and M2 cells were generated in the presence of $^{13}\text{C}_6$ -Glc. At day 5, the intracellular glycogen pool was purified and then hydrolyzed to generate Glc monomers before quantification by GC-MS, as described (Gudewicz PW et al, 1976). Results revealed the presence of ^{13}C -labelled Glc in both cell types (Fig. 1G and EV1C).*”

Reference Gudewicz PW, Filkins JP. Glycogen metabolism in inflammatory macrophages (1976). J Reticuloendothel Soc. Aug;20(2):147-57

Methods section, paragraph “Glycogen extraction and hydrolysis, and glucose analysis by GC-MS”, page 25, line 571: “*The glucose content was evaluated by exact mass gas chromatography/mass spectrometry (GC-MS) by measuring ^{13}C enrichment using the mass spectrometer Orbitrap Q Exactive (Thermo Fisher Scientific). ^{13}C -enrichment data represent the ^{13}C atom excess whereby the contribution of natural ^{13}C abundance (determined in non-labelled samples) has been subtracted*”.

The same strategy was used with $^{13}\text{C}_5$ -Gln or $^{13}\text{C}_3$ -lactic acid to confirm that M1 cells are capable of glyconeogenesis with Gln and LA as substrates.

The Results section has been modified as follows: paragraph “The combination of GM-CSF plus IFN γ promotes human macrophage glyconeogenesis”, page 9, line 184: “*We confirmed the occurrence of glyconeogenesis in M1 cells by generating these cells in Glc^{low} Gln^{low} medium in the presence of $^{13}\text{C}_5$ -Gln or $^{13}\text{C}_3$ -lactic acid before monitoring ^{13}C -glucose content in glycogen by GC-MS. ^{13}C -enriched glycogen was detected in cells cultured with Gln and LA, evidencing the use of both molecules as alternative substrates for glyconeogenesis by M1 cells (Fig. 2B).*”

2-1. The data involving tumor-derived macrophages (TAM), are less convincing. The supplementation or treatment of cells collected from ascites is not particularly helpful without some integration of the metabolic niche of the peritoneal space. Is glucose limited there? Might the fuel sources be different there?

The objective of these experiments was to evaluate whether ex vivo isolated established macrophages, as observed with monocytes-derived macrophages, contain glycogen stores and/or are capable of glycogenesis and glyconeogenesis. We thus used macrophages isolated from ascites of patients with

ovarian cancer as a model of established tumor-associated macrophages (TAMs) (Nowak M & Klink M. 2020).

We observed that freshly isolated TAM do not contain glycogen but synthesize it using Glc or lactic acid after culture in the presence of GM-CSF and IFN γ (Figure 4A). Results also evidenced the presence of glycogen in the cytoplasm of TAM infiltrating lung adeno carcinoma (as assessed by PAS staining, Figure. 4C).

That TAMs exhibited low glycogen stores after purification (Figure 4A) probably results from the low concentrations of cytokines enabling glycogenesis and glyconeogenesis in ovarian tumor ascites (GM-CSF: <100 pg/ml, IFN γ and IL-4: <10 pg/ml) rather than from an absence of fuel sources, as Glc and lactic acid are detectable in these ascites (4.1 ± 1.2 mM and 4.8 ± 3 mM, respectively; n=10).

This point is now mentioned in the Results section, paragraph “Tumor associated macrophages store glycogen and are capable of glyconeogenesis”, page 11, line 250, as follows: “*Consistent with this observation, ascites fluids had detectable levels of Glc (< 5 mM), and low levels of IL-4 and IFN γ (<10 pg/ml), and GM-CSF (<100 pg/ml), which are required for glycogenesis and glyconeogenesis by human macrophages*”.

In additional experiments, we verified that glycogen is necessary for TAM to function (as observed for in vitro generated macrophages). TAMs were cultured in Glc^{low} Gln^{low} medium containing GM-CSF and IFN γ and supplemented or not with Glc or lactic acid. At day 3, cells were stimulated with LPS to monitor cytokine secretion, in the presence or absence of the glycogenolysis inhibitor CP-91149. Results showed that, as observed for monocyte-derived macrophages, CP-9114 significantly reduced the production of TNF α , VEGF and G-CSF by LPS-stimulated TAM (see Figure R3-2). These results suggest that, similar to in vitro generated macrophages, glycogen metabolism support TAM function.

Figure R3-2. Glycogen metabolism is necessary for TAM function. TAM isolated from ovarian cancer ascites were cultured with GM-CSF and IFN γ in Glc^{low} Gln^{low} medium supplemented with Glc or lactic acid. At day 3, cells were treated or not with CP-91149 for 15 min before a 6 h stimulation with LPS. TNF α , VEGF and G-CSF were quantified by ELISA (n=3). Values are represented as the mean \pm SEM, each dot represents a different donor. Statistical significance was determined by paired t-test.

These new results have been inserted in the revised Figure 4 and the Results and Methods sections have been modified as follows:

Results section, paragraph “Glycogen sustains M1 cell and M2 cell functions”, page 14, line 303: *“The inhibition of glycogenolysis also reduced cytokine production by .../... and by TAM with glycogen stores generated from Glc or lactic acid (Fig. 4E).”*

Methods section, paragraph “Tumor associated macrophages”, page 23, line 509: *“.../... For glycogenolysis experiments, day 3 TAM were treated or not with 50 μ M CP-91149 for 15 min before a 6 h stimulation with 100 ng/mL LPS”.*

Legend of the new Figure 4E, page 40, line 987: *“.../... Day 3 TAM were treated or not with CP-91149 for 15 min before a 6 h stimulation with LPS. TNF α , VEGF and G-CSF were quantified by ELISA (n=3). Values are represented as the mean \pm SEM, each dot represents a different donor. Statistical significance was determined by paired t-test.*

References cited in the response to Reviewer:

Nowak M, Klink M. The role of Tumor-Associated Macrophages in the progression and chemoresistance of ovarian cancer. *Cells*. 2020;9(5):1299.

Hackett MJ, Sylvain NJ, Hou H, Caine S, Alaverdashvili M, Pushie MJ, Kelly ME. Concurrent glycogen and lactate imaging with FTIR spectroscopy to spatially localize metabolic parameters of the glial response following brain ischemia. *Anal Chem*. 2016;88(22):10949-10956.

2-2 In addition, the PAS and PAS + amylase staining is not convincing. There is no counter staining for CD163, so it is unclear what cells are PAS positive (+ minor comment: Figure 4b- how are the CD163+ cells detected in these sections? There is no staining for this marker. Sort some TAM for glycogen?).

As suggested, additional experiments have been performed to determine, by immunohistochemistry, the expression of CD163 in lung adenocarcinoma tissue sections. Lung adenocarcinomas are heavily infiltrated by macrophages (Welsh TJ et al. 2005; Al-Shibli K et al. 2009; Mei J et al. 2016; Jackute J et al. 2018; reviewed in Conway EM et al. 2016) and characterized by a harsh microenvironment (rendering necessary a metabolic adaptation of macrophages to survive and function).

Results revealed a massive infiltration of CD163-positive and of PAS-positive cells in the same area (see Figure R3-3a); as a control, type 2 pneumocytes and adenocarcinoma cells are TTF-1-positive whereas PAS-positive cells are TTF-1-negative (see Figure R3-3b).

In addition, Fourier-transform infrared spectroscopy (FITR) confirmed the colocalization of lactic acid and glycogen in tissue areas enriched with PAS-positive macrophages. A significant linear correlation between glycogen and lactic acid levels ($R^2=0.87$, $p<0.0001$) suggests that lactic acid may serve as a substrate for glycogen synthesis in vivo (see Figure R3-3c). Given that GM-CSF and IFN- γ , along with elevated lactic acid, are present in solid tumors, this finding supports the concept that TAMs can utilize lactic acid to generate glycogen.

Figure R3-3: Tumor-associated macrophages in lung adenocarcinoma store glycogen. **a.** Immunohistochemical analysis of CD163 expression in lung adenocarcinoma. On the same tissue section, glycogen was detected by PAS staining, after treatment or not with amylase (Scale bar: 100 μm). **b.** Immunohistochemical staining of TTF-1 in adenocarcinoma cells (Scale bar: 100 μm). **c.** Lactic acid and glycogen quantification by FTIR spectroscopic imaging in lung adenocarcinoma. *Left panel*, unstained brightfield image; *right panels*, glycogen and lactic acid maps (Scale bar: 200 μm); linear correlation between lactic acid and glycogen contents in infiltrating lung adenocarcinoma was calculated (results are representative 1 out of 2 independent experiments).

In the revised version of the manuscript, these results have been added in Figures 4 and EV4; the Results and Methods sections and the corresponding Legends have been modified as follows:

Results section, paragraph "Tumor-associated macrophages store glycogen and are capable of glyconeogenesis" page 12, line 259: "...As expected, lung adenocarcinoma was massively infiltrated

by CD163-positive macrophages (large cells with abundant, vacuolated cytoplasm and a round, regular nucleus containing a small nucleolus) (Fig. 4C). In the same tissue sections, PAS staining, which was abolished by α -amylase, revealed the presence of glycogen in TAM (Fig. 4C). Moreover CD163-positive cells were not stained with TTF1, a marker of adenocarcinoma cells and type 2 pneumocytes that do not contain glycogen (Fig. EV4). The presence of glycogen in macrophages was confirmed by FTIR spectroscopic imaging in serial sections. FTIR spectroscopy also revealed a colocalization of glycogen with lactic acid ($R^2 = 0.87$, $p < 0.0001$) (Fig. 4D). Given that GM-CSF and IFN- γ , along with elevated lactic acid, are present in solid tumors, this finding suggests that TAMs can use lactic acid *in vivo* to generate glycogen (Fig. 4D).

Methods section paragraph “Glycogen detection and quantification”, page 24, line 533: “.../... Slides were counterstained with hematoxylin. Mouse anti-human CD163 monoclonal antibody (clone 10D6; Leica Microsystems, Newcastle, UK) was used to determine TAM distribution on consecutive slides. Slides were analyzed using the Leica Bond III immunostainer (Leica Microsystems).”

A new paragraph has been added to describe the “FTIR spectroscopic imaging”, page 27, line 621: “Five-micrometer-thick section of the paraffin-embedded tumor were deposited on BaF2 windows. Spectral analysis was performed with a Bruker Hyperion 3000 infrared microscope coupled to a Vertex 70 spectrometer using a 64×64 focal plane array (FPA) detector. A $15\times$ Cassegrain objective (Numerical Aperture 0.4) was used for all acquisition. Mid-infrared spectra were recorded at a resolution of 4 cm^{-1} (spectral region $900\text{--}2000\text{ cm}^{-1}$), with 32 accumulations in transmission mode. Background spectra were also recorded with the same specifications. Post-processing of spectra was done with a lab-made script written in Matlab and includes baseline correction, digital paraffin subtraction using signal intensity at 1465 cm^{-1} , vector normalization to the amide I band and denoising using the Savitzky-Golay algorithm with a degree of 2 and a span length of 9. A quality control of each spectrum was performed by calculating the signal-to-noise ratio (SNR) over the spectral range $1850\text{--}2000\text{ cm}^{-1}$ free of biological signal. Second derivative were computed over the spectral range $1034\text{ cm}^{-1} - 1185\text{ cm}^{-1}$ and used as a loading vector for curve fitting. Spectral curve fitting quality was assessed by a root mean square error value set at 1% of the total spectral interval area. Intensities of sub bands located at $\sim 1127\text{ cm}^{-1}$ and $\sim 1152\text{ cm}^{-1}$ were normalized to the intensity of the $\sim 1655\text{ cm}^{-1}$ amide I band to estimate the lactic acid and glycogen contents, respectively, as reported (Hackett et al, 2016).

Reference: Hackett MJ, Sylvain NJ, Hou H, Caine S, Alaverdashvili M, Pushie MJ, Kelly ME. Concurrent glycogen and lactate imaging with FTIR spectroscopy to spatially localize metabolic parameters of the glial response following brain ischemia. *Anal Chem.* 2016;88(22):10949-10956.

The legends of Figures 4C&D and EV4:

- **4C**, page 40, line 981: “Immunohistochemical analysis of CD163 expression in lung adenocarcinoma. On the same tissue section, glycogen was detected by PAS staining, after treatment or not with amylase; Scale bar: $100\ \mu\text{m}$ ”.
- **4D**, page 40, line 983: “.../... Lactic acid and glycogen quantification by FTIR spectroscopic imaging in lung adenocarcinoma. Left panel, unstained bright field image; right panel, glycogen and lactic acid maps; Scale bar, $200\ \mu\text{m}$; linear correlation between lactic acid and glycogen contents in infiltrating lung adenocarcinoma was calculated (results are representative of 1 out 2 biologically independent experiments).
- **EV4**, page 47, line 1101: Immunohistochemical staining of TTF-1 in lung adenocarcinoma. Scale bar: $100\ \mu\text{m}$.

References cited in the response to Reviewer:

Al-Shibli K, Al-Saad S, Donnem T, Persson M, Bremnes RM, and Busund LT (2009). The prognostic value of intraepithelial and stromal innate immune system cells in non-small cell lung carcinoma. *Histopathology*. 2009;55:301-12.

Conway EM, Pikor LA, Kung SH, Hamilton MJ, Lam S, Lam WL, Bennewith KL. Macrophages, Inflammation, and Lung Cancer. *Am J Respir Crit Care Med*. 2016;193:116-30.

Jackute J, Zemaitis M, Pranys D, Sitkauskiene B, Miliauskas S, Vaitkiene S, Sakalauskas R. Distribution of M1 and M2 macrophages in tumor islets and stroma in relation to prognosis of non-small cell lung cancer. *BMC Immunol*. 2018;19:3.

Mei J, Xiao Z, Guo C, Pu Q, Ma L, Liu C, Lin F, Liao H, You Z, Liu L. Prognostic impact of tumor-associated macrophage infiltration in non-small cell lung cancer: A systemic review and meta-analysis. *Oncotarget*. 2016;7:34217-28.

Welsh TJ, Green RH, Richardson D, Waller DA, O'Byrne KJ, and Bradding P (2005). Macrophage and mast-cell invasion of tumor cell islets confers a marked survival advantage in non-small-cell lung cancer. *J Clin Oncol*. 2005;23:8959-67.

Minor comments

The authors suggest the lack of congruence with previous work comes from the shorter exposure to IL4 in the published studies. This could easily be tested here and should be.

This information was mentioned in the Figure 1B in the submitted version of the manuscript. Figure 1B shows the time-dependent effect of IL-4 on glycogen synthesis by M-CSF-M ϕ . A 24 h incubation with IL-4 induced much lower glycogen levels than a 3 day incubation (see Figure R3-4). We agree with the remark of the Reviewer: in the initial version, the incubation time with IL-4 may have been incorrectly explained, as IL-4 was added at day 2 and analysis was performed at day 5.

Consequently, the revised manuscript has been modified as follows:

Results section, the paragraph “GM-CSF or M-CSF plus IL-4 triggers human macrophage glycogenesis”, page 6, line 121: “*The addition of IL-4 at day 2 to differentiating M-CSF-M ϕ triggered a small but significant accumulation of glycogen after 24 h (D3) and a substantial accumulation after 3 days (D5) (Fig. 1B).*”

Discussion section, page 16, line 360: “*Based on our observations, we suspect that the lack of glycogen reserves reported by others in murine M2 cells generated with M-CSF and IL-4 could be due to an insufficient incubation time with IL-4, which was limited to only 24 hours¹⁸.*”

References: 18. Ma, J. et al. Glycogen metabolism regulates macrophage-mediated acute inflammatory responses. *Nat. Commun.* 11, 1–16 (2020).

Moreover, in the initial Figure 1B, the results were obtained with cells from 3 donors. In the revised version, additional experiments with monocytes from two additional subjects were performed (see Figure R3-4).

Figure R3-4. GM-CSF or M-CSF plus IL-4 triggers human macrophage glycosynthesis.... B. Day 2 M-CSF-Mφ were exposed to IL-4 (purple) or not (Grey) for the last 3 days. Glycogen was quantified in monocytes (D0) and after 24 h (D3), 48 h (D4) and 72 h (D5) incubation with or without IL-4 (n=5-8). Values are represented as the mean ± SEM, each dot represents a different donor. Statistical significance was determined by two-tailed unpaired Welch paired t-test.

Figure 1g is not very helpful as there is nothing to compare it to. Are these values greater than would be seen for glutamine or some other fuel? What are the values in control cells? And similarly, figure 2b is difficult to interpret. It suggests that glutamine but not lactate is used but 2a shows both support glycogen accumulation. At what point is this considered positive?

We thank Reviewer 3 for pointing out this difference.

In Figure 1G, cells were cultured in CM for 2 days and then in Glc^{low} medium (0.1 g/L Glc) supplemented with ¹³C₆-Glc for 3 days. The levels of intracellular glycogen were determined at day 5 by quantifying ¹³C-Glc by GC-MS, as described by Ma et al. (2020); this protocol requires glycogen purification and hydrolysis before analysis. Figure 1G therefore presents the net flux of glycogen synthesis rate in cells cultured for 3 days in Glc^{low} medium.

Figure 2B shows that that ¹³C₅-Gln and ¹³C₃-lactic acid are metabolized by M1 cells to generate ¹³C-glycogen (Figure 2B). To obtain this result, the same strategy was used with ¹³C₅-Gln or ¹³C₃-lactic acid to trace M1 cell glyconeogenesis from Gln and lactic acid (LA) (instead of ¹³C-Glc in GM-Mφ and M2 cells). More precisely, monocytes were cultured in CM with GM-CSF for 2 days followed by 3 days culture in a Glc^{low} Gln^{low} medium supplemented with ¹³C₅-Gln and ¹³C₃-LA, in the presence of IFN γ . At day 5, total glycogen was quantified using the total glycogen assay kit from BioVision (Fig. 2A; results expressed in µg glycogen per 10⁶ cells) or glycogen was purified and hydrolyzed for GC-MS analysis (Fig. 2B; results expressed as a glycogen synthesis flux; ng ¹³C per 10⁶ cells per 24 h).

It is important to note that the net glycogen synthesis flux presented in both figures is expressed per 10⁶ cells per 24 h after subtraction of the levels of ¹²C present in glycogen synthesized by the cells between day 0 and day 2.

The order of magnitude of net glycogen synthesis flux from ¹³C-Gln and ¹³C-LA (Figure. 2B) is in the order of nanograms per million cells per day, i.e. nearly 2 orders of magnitude lower than with ¹³C-

glucose (Figure 1G). These effects, were, however, expected and reflect classical effects observed during isotopic labelling:

1) Isotopic dilution: Gln and LA metabolization through the gluconeogenic pathway must be accompanied by an isotopic dilution along the path, due to the contribution of non-labelled, preexisting metabolic pools of intermediates, and thus the synthesis of only partially ^{13}C -labelled glucose to form glycogen.

2) Cellular Gln and LA production: Gln and LA produced by cells themselves via protein turn-over and organic acids metabolism (i.e. the participation of non-labelled Gln and LA) was also likely involved, thereby leading to an extra isotopic dilution.

3) Loss of ^{13}C : In addition, a fraction of Gln and LA can be used also by catabolism, leading to a loss of ^{13}C (in the form of $^{13}\text{CO}_2$). Additional experiments were carried out with $^{13}\text{C}_5$ -Gln, with or without a succinate dehydrogenase (SDH) inhibitor (Figure 3E). Since the TCA cycle is disrupted in murine M1 cells at SDH, we wondered whether the same phenomenon occurs in human macrophages (i.e. may also exhibit a truncated TCA cycle). Our results show that Gln can be metabolized via the classical TCA cycle (via succinate and malate) (Fig. 3E). These observations confirm that the metabolism of human and murine macrophages is different and that effectively, there is a metabolic partitioning of Gln between competing pathways: gluconeogenesis and catabolism. That is, only a fraction of supplied Gln is used by gluconeogenesis. For lactic acid, our previous data showed that lactic acid is internalized and metabolized by catabolism in inflammatory macrophages to generate malate and OAA, which can be in turn used as a substrate by gluconeogenesis via PEP (Paolini et al., 2020).

Taken as a whole, the observed net flux of ^{13}C -glycogen synthesis from ^{13}C -labelled Gln and LA is small while the total amount of glycogen is similar to that in Fig. 1 due to isotopic dilution (and thus the use of ^{12}C) and ^{13}C loss in metabolism, such that gluconeogenic glucose is much less ^{13}C -labelled than source Gln and LA.

Reference cited in the response to Reviewer 3:

Ma, J. et al. Glycogen metabolism regulates macrophage-mediated acute inflammatory responses. *Nat. Commun.*

11, 1–16 (2020).

Paolini, L. et al. Lactic acidosis together with GM-CSF and M-CSF induces human macrophages toward an inflammatory protumor phenotype. *Cancer Immunol Res.* 2020;8:383-395.

The use of experimental graphics is excessive. These should be used only for particularly complex experiments or to show the sites of the action of the inhibitors.

We thank Reviewer 3 for this comment. In the revised manuscript, the following experimental graphics have been removed from figures 1F, 2D, 4A, 6A-C, EV3B and EV5C; only those that we considered essential for illustrating the experimental protocol were kept.

The levels of gene expression and protein expression do not seem to match the levels of glycogen accumulation. Could the authors comment/explain.

Our results show that, for a given experimental condition, the expression levels of mRNAs and proteins involved in glycogenesis (GYS1), gluconeogenesis (PCK2/FBP1) are related to glycogen levels.

More precisely, we observed that (i) the expression of GYS1 (at the mRNA and protein levels) is higher in GM-CSF-M ϕ and M2 compared with monocytes (Figures 1I and EV1D), consistent with the fact that these cells contain more glycogen than monocytes, and (ii) that the expression of the enzymes

FBP1 and PCK2, at both transcriptional and translational levels, is increased in M1 cells compared with GM-CSF-M ϕ and M2 cells (Figures 2 D&E), consistent with the ability of these cells to synthesize glycogen using the non-carbohydrate substrates Gln, LA and glycerol. In this way, our results show that the FBP1 enzyme is expressed by both cell types (GM-CSF-M ϕ and M1), while the PCK2 enzyme is twice as highly expressed in M1 cells than GM-CSF-M ϕ .

GM-CSF and IFN γ are expressed at inflammatory sites. Our results therefore suggest that IFN γ is a key player in triggering glyconeogenesis in M1 cells, at least via the induction of PCK2 expression which activation leads to the generation of fructose 1,6-biphosphate, the substrate of the FBP1 enzyme (see Figure 2C) that is involved in glycogen synthesis by M1 cells.

Altogether, the expression levels of these enzymes and their involvement in glycogen accumulation were validated through inhibition experiments (using chemical drugs) and further confirmed by using specific siRNAs (Figures 3 B-D). These findings reinforce their pivotal role in glycogen metabolism by macrophages.

Dear Prof. Jeannin,

Thank you for submitting your revised manuscript. It has now been seen by all of the original referees.

As you can see, the referees find that the study is significantly improved during revision and recommend publication. However, I need you to address the points below before I can accept the manuscript.

- Please address the minor concerns of referee #2.
- Please provide 3-5 keywords for your study. These will be visible in the html version of the paper and on PubMed and will help increase the discoverability of your work.
- Disclosure and competing interests statement needs to be moved after Acknowledgements.
- We note an author name discrepancy - i.e. Laetitia Basset in the manuscript file vs. Leatitia Basset in the manuscript tracking system.
- Author contributions section needs to be removed from the manuscript text file.
- We note the phrase 'data not shown' on page 7, which is not allowed as per journal policy. Please either show the data or remove the statement.
- We note that a Table S1 is called out in the text, but we cannot locate such a table.
- All research articles submitted as revised versions must include a structured methods section that includes a Reagents and Tools Table followed by a Methods and Protocols section. Please see <https://www.embopress.org/page/journal/14693178/authorguide#structuredmethods> for further information.
- Please provide a filled in source data checklist, which was sent by our source data coordinator Dr. Hannah Sonntag (I also attached it to this email).
- Please make sure that the source data matches the revised figure panels - e.g. I note a source data file labeled as Fig 4C, which is an image based panel.
- The title 'Summary' should be corrected as 'Abstract'.
- In the reference list, some references are marked as [PREPRINT] although they seem to have been published, which need to be corrected.
- Please include a statement that the experiments conformed to the principles set out in the WMA Declaration of Helsinki and the Department of Health and Human Services Belmont Report at the monocyte isolation as well. Currently, it is only mentioned in the Tumor-associated macrophages section.
- Our production/data editors have asked you to clarify several points in the figure legends:
 - o Please note that the exact p values are not provided in the legends of figures 1 a-b, d-f, i; 2a, d; 3b, e; 4a, d-e; 5a; 6a, c-f; EV 2a-b; EV 3b; EV 5d.

Thank you again for giving us to consider your manuscript for EMBO Reports, I look forward to your minor revision.

Kind regards,

Deniz Senyilmaz Tiebe

--

Deniz Senyilmaz Tiebe, PhD
Senior Scientific Editor
EMBO Reports

Referee #1:

Jeroundi and coworkers have provided a comprehensive revision of their manuscript. Authors have met the concerns of the reviewers and in doing so new experimental data is provided. All in all, authors have solidified the conclusions of their study.

Referee #2:

I appreciate the great effort the authors made to address the comments from each reviewer. Overall, the quality of this work is good and has been further improved by the revisions. There are some minor comments related to my original question #2 and #3.

Regarding #2, the authors explained the discrepancy could be due to a large portion of the 8ug glycogen store at day 5 is pre-existing. If this is the case, it would argue that the glycogen synthesis during the time in CM only have minor net contribution in increasing the glycogen store in cells. If this is the case, the conclusion and text should be adjusted to accurately reflect that. Regarding #3, the authors suggested that isotopic dilution/ production of Gln and LA is a major reason behind the orders of magnitude difference. Should this be the case, support of this should be seen in precursor labeling (e.g. intracellular Gln, LA, or

PEP labeling). Also, what other metabolic sources "dilute" the labeling and contribute to the production of glycolysis/gluconeogenesis precursors is an important issue. It could suggest that they are the important fuel (potentially more than Gln) under poor nutrient condition

Referee #3:

My concerns have been sufficiently addressed by the authors

Dear Editor,

On behalf of the authors, I sincerely acknowledge Referees and Editors for their positive feedback on our study. Please find hereafter responses to the minor concerns of referee #2 and to the points listed.

1. Responses to minor comments of referee #2.

1.1. Regarding question #2, the authors explained the discrepancy could be due to a large portion of the 8 µg glycogen store at day 5 is pre-existing. If this is the case, it would argue that the glycogen synthesis during the time in CM only have minor net contribution in increasing the glycogen store in cells. If this is the case, the conclusion and text should be adjusted to accurately reflect that.

We acknowledge Reviewer 2 for pointing this out.

In the response to the Reviewer 2, we have written that "the total glycogen pool (Fig. 1a) may include glycogen that pre-existed in the cells prior to culture and not just glycogen synthesized during the 5 days of culture". In fact, the data shows that monocytes contain low levels of glycogen, indicating that the glycogen detected on day 5 was synthesized during culture. We apologize for the error in writing the reply and for any confusion this may have caused. The text in the manuscript is indeed correct and has therefore not been amended.

1.2. Regarding question #3, the authors suggested that isotopic dilution/ production of Gln and LA is a major reason behind the orders of magnitude difference. Should this be the case, support of this should be seen in precursor labeling (e.g. intracellular Gln, LA, or PEP labeling). Also, what other metabolic sources "dilute" the labeling and contribute to the production of glycolysis/gluconeogenesis precursors is an important issue. It could suggest that they are the important fuel (potentially more than Gln) under poor nutrient condition.

The isotopic dilution means that metabolic intermediates from the ^{13}C -labelled source and the final product (glycogen or UDPG) should be less ^{13}C -enriched than the ^{13}C -source (Gln or lactic acid) itself. As the Referee says, it can be due to the contribution of alternative carbon sources, but not only. As mentioned in our response, there are three possible explanations for this result

- The use of the ^{13}C -compound leading to ^{13}C loss (we provided an example associated with the Krebs cycle, generating $^{13}\text{CO}_2$ which is lost as respired carbon dioxide),
- A simple dilution in pre-existing pools of intermediates (when labelling starts, all pools are non-labelled and it takes time to renew them and thus reach the isotopic maximum),
- Utilization of alternative carbon sources which can isotopically dilute ^{13}C in intermediates along the path from the carbon source to UDPG.

The last point can be easily illustrated. For example, in our previous study dealing with lactic acidosis where macrophages were fed with ^{13}C -lactic acid, intracellular lactate was at about 50% ^{13}C and succinate was at about 6% ^{13}C , demonstrating the isotopic dilution caused by internal carbon sources (pyruvate and acetyl-CoA from catabolism). Here, as suggested by the Referee, we did a tracing trial with ^{13}C -Gln and analyzed some metabolites by high resolution mass spectrometry. The results are shown below in **Figure R1**.

Figure R1. Trial experiment showing the isotopic enrichment in intracellular metabolites in macrophages after feeding with $^{13}\text{C}_5$ -Gln. The isotopic enrichment is shown in molecular $\%^{13}\text{C}$. The horizontal dashed line recalls the expected $\%^{13}\text{C}$ at natural ^{13}C abundance (1.1%). “control”, “nat ab” and “labelled” stand for control medium without Gln, culture medium with non-labelled Gln, and culture medium with $^{13}\text{C}_5$ -Gln, respectively. Experiments were carried out without or with the mitochondrial permease inhibitor NV161. Metabolites are ordered by decreasing $\%^{13}\text{C}$ under labelled conditions.

In this example, one can see that intracellular glutamate (Glu) and 2-oxoglutarate (2OG), the direct products of glutamine utilization, are not at 100% ^{13}C but rather at about 60% ^{13}C (without NV161), demonstrating the isotopic dilution by anaplerosis (pyruvate carboxylase) and the Krebs cycle reactions from non-labelled sources (culture medium with Glc). The same applies to other intermediates. For example, aspartate (Asp) is at about 15% ^{13}C due to the contribution of pyruvate carboxylase (anaplerosis) generating non-labelled oxaloacetate from bicarbonate and pyruvate. Accordingly, such a contribution from anaplerosis is decreased when the exchange of organic acids between the cytosol and the mitochondrion is inhibited with NV161, leading to an increase in the percentage of ^{13}C in succinate, glutamate and 2-OG and a decline in aspartate turn-over (and thus a decrease in $\%^{13}\text{C}$). Lactate and pyruvate are at about 2% ^{13}C only (i.e. twice the ^{13}C natural abundance) demonstrating the isotopic dilution by metabolism (e.g. alanine/pyruvate interconversion, pyruvate production by pyruvate kinase).

As outlined by Referee 2, although lactate, glycerol and glutamine are the most important direct contributors to glyconeogenesis, other metabolites and intermediates of the TCA constitute glyconeogenic precursors (AM Shah & FE Wondisford. 2020) that may have impacted the flux of ^{13}C -lactic acid and ^{13}C -Gln in the synthesis of glycogen. For example, eighteen out of 20 amino acids as well as triglycerides can contribute to glyconeogenesis (AM Shah & FE Wondisford. 2020 ; MH Green. 2020). It is important to note that (i) the net flux of glycerol into glyconeogenesis was not assessed in our study and (ii) that fluxes were only measured at day 5 and we can't exclude that fluxes could have been stronger at earlier times.

Overall, this supports our previous explanations that isotopic dilution by pre-existing metabolic pools and simultaneous metabolic pathways has occurred, leading to a relatively low percentage of ^{13}C in intermediates and thus explaining the apparent low allocation of ^{13}C to glycogen.

These points are now mentioned in the Results section (paragraph « A combination of GM-CSF plus IFN γ promotes human macrophage glyconeogenesis.

References cited.

- Green MH. Are Fatty Acids Gluconeogenic Precursors? J Nutr. 2020;150(9):2235-2238.
- Shah AM, Wondisford FE. Tracking the carbons supplying gluconeogenesis. J Biol Chem. 2020;295(42):14419-14429.

2. Responses to Editor comments.

- Please provide 3-5 keywords for your study.

Keywords: Macrophages, Glyconeogenesis, Glycogenolysis, Cytokine secretion, Phagocytosis.

- Disclosure and competing interests statement needs to be moved after Acknowledgements.

Done.

- We note an author name discrepancy - i.e. Laetitia Basset in the manuscript file vs. Leatitia Basset in the manuscript tracking system.

The exact first name writing is Laetitia – corrected in the tracking system.

- Author contributions section needs to be removed from the manuscript text file.

Done.

- 'Data not shown' is not allowed as per journal policy.

The phrase “data not shown” has been removed from the text.

- We note that a Table S1 is called out in the text, but we cannot locate such a table.

Table 1 was erroneously named Table S1 in the submitted manuscript. This error has been corrected in the text file of the updated manuscript.

- All research articles submitted as revised versions must include a structured methods section that includes a Reagents and Tools Table followed by a Methods and Protocols section.

As requested, a Reagents and Tools Table has been inserted in the Methods and Protocols section.

- Please provide a filled in source data checklist, which was sent by our source data coordinator Dr. Hannah Sonntag (I also attached it to this email).

A checklist of completed source data has been added.

- Please make sure that the source data matches the revised figure panels - e.g. I note a source data file labeled as Fig. 4C, which is an image based panel.

Done (in attachment).

- The title 'Summary' should be corrected as 'Abstract'.

Done.

- In the reference list, some references are marked as [PREPRINT] although they seem to have been published, which need to be corrected.

The reference list has been checked and corrected.

- Please include a statement that the experiments conformed to the principles set out in the WMA Declaration of Helsinki and the Department of Health and Human Services Belmont Report at the monocyte isolation as well. Currently, it is only mentioned in the Tumor-associated macrophages section.

Blood from healthy volunteers was collected and used according to the Declaration of Helsinki and the Department of Health and Human Services Belmont Report guidelines. As mentioned, the use of human blood for research and the protocols described in this manuscript have been validated the ethics committee of the University Hospital of Angers. This point has been correction in the Methods and protocols section (paragraph Monocyte isolation).

- Our production/data editors have asked you to clarify several points in the figure legends:
« Please note that the exact p values are not provided in the legends of figures 1a-b, d-f, i; 2a, d; 3b, e; 4a, d-e; 5a; 6a, c-f; EV 2a-b; EV 3b; EV 5d »

As requested, p-value ranges have been included in the legends of the listed figures.

Prof. Pascale Jeannin
Université of Angers
France

Dear Prof. Jeannin,

Thank you for submitting your revised manuscript. I have now looked at everything and all is fine. Therefore, I am very pleased to accept your manuscript for publication in EMBO Reports.

Congratulations on a nice work!

Kind regards,

Deniz Senyilmaz Tiebe

--

Deniz Senyilmaz Tiebe, PhD
Senior Scientific Editor
EMBO Reports

--
